# B cells orchestrate tolerance to the neuromyelitis optica autoantigen AQP4

Ali Maisam Afzali[1,2,3], Lucy Nirschl[1], Christopher Sie[1], Monika Pfaller[1], Oleksii Ulianov[1], Tobias Hassler[4], Christine Federle[4], Elisabetta Petrozziello[4], Sudhakar Reddy Kalluri[2], Hsin Hsiang Chen[1], Sofia Tyystjärvi[1], Andreas Muschaweckh[1], Katja Lammens[5], Claire Delbridge[6,7], Andreas Büttner[8], Katja Steiger[6], Gönül Seyhan[9], Ole Petter Ottersen[10], Rupert Öllinger[11], Roland Rad[11], Sebastian Jarosch[12], Adrian Straub[12], Anton Mühlbauer[12], Simon Grassmann[13], Bernhard Hemmer[2,3], Jan P. Böttcher[14], Ingrid Wagner[15], Mario Kreutzfeldt[15], Doron Merkler[15], Irene Bonafonte Pardàs[16], Marc Schmidt Supprian[9], Veit R. Buchholz[12], Sylvia Heink[1], Dirk H. Busch[12,17], Ludger Klein[4] & Thomas Korn[1,2,3 ✉]

Neuromyelitis optica is a paradigmatic autoimmune disease of the central nervous system, in which the water-channel protein AQP4 is the target antigen[1]. The immunopathology in neuromyelitis optica is largely driven by autoantibodies to AQP4[2]. However, the T cell response that is required for the generation of these anti-AQP4 antibodies is not well understood. Here we show that B cells endogenously express AQP4 in response to activation with anti-CD40 and IL-21 and are able to present their endogenous AQP4 to T cells with an AQP4-specific T cell receptor (TCR). A population of thymic B cells emulates a CD40-stimulated B cell transcriptome, including AQP4 (in mice and humans), and efficiently purges the thymic TCR repertoire of AQP4-reactive clones. Genetic ablation of *Aqp4* in B cells rescues AQP4-specific TCRs despite sufficient expression of AQP4 in medullary thymic epithelial cells, and B-cell-conditional AQP4-deficient mice are fully competent to raise AQP4-specific antibodies in productive germinal-centre responses. Thus, the negative selection of AQP4-specific thymocytes is dependent on the expression and presentation of AQP4 by thymic B cells. As AQP4 is expressed in B cells in a CD40-dependent (but not AIRE-dependent) manner, we propose that thymic B cells might tolerize against a group of germinal-centre-associated antigens, including disease-relevant autoantigens such as AQP4.

Neuromyelitis optica (NMO) is an autoimmune disease of the central nervous system (CNS) that is mediated by autoantibodies against the water-channel protein AQP4. AQP4 is widely expressed in the body, including the kidneys, stomach and muscle, but its M23 isoform occurs in orthogonal arrays of particles in astrocytes, which has been proposed to be one of the reasons why astrocytes appear to be a prime target of complement-mediated lysis when anti-AQP4 antibodies (NMO-IgG) bind to these structures. The effector function of NMO-IgG has been investigated in great detail[2]. However, less is known about the conditions that lead to a breach of tolerance against AQP4. NMO-IgG is class-switched (IgG1 and IgG3 in humans) and hypermutated, indicating that these autoantibodies are generated as the result of a germinal centre (GC) reaction[3]. NMO is also associated with a specific HLA haplotype (DRB1*0301)[4], and an immunodominant T cell epitope of AQP4 was reported in humans[5], again suggesting that an antigen-specific T cell response is required for the generation of NMO-IgG.

In a homologous mouse system, we and others have identified the MHC class II (I-A[b])-restricted epitope of AQP4[6,7]. In wild-type (WT) mice, the natural T cell repertoire is essentially devoid of AQP4-specific T cells[6]. Elimination of AQP4-specific TCRs in the thymus would be an appropriate strategy to establish tolerance against AQP4. As AQP4 is relatively broadly expressed, immunological ignorance is precluded. Classically, medullary thymic epithelial cells (mTECs) express and present tissue-restricted antigens and inducible autoantigens to engage

[1]Institute for Experimental Neuroimmunology, Technical University of Munich School of Medicine and Health, Munich, Germany. [2]Department of Neurology, Technical University of Munich School of Medicine and Health, Munich, Germany. [3]Munich Cluster for Systems Neurology, Munich, Germany. [4]Biomedical Center (BMC), Institute for Immunology, Faculty of Medicine, Ludwig-Maximilians-University Munich, Planegg-Martinsried, Germany. [5]Department of Biochemistry at the Gene Center, Ludwig-Maximilians-University, Munich, Germany. [6]Institute of Pathology, Technical University of Munich School of Medicine and Health, Munich, Germany. [7]Department of Neuropathology, Institute of Pathology, Technical University of Munich School of Medicine and Health, Munich, Germany. [8]Institute of Forensic Medicine, Rostock University Medical Center, Rostock, Germany. [9]Institute for Experimental Hematology, TranslaTUM Cancer Center, Technical University of Munich School of Medicine and Health, Munich, Germany. [10]Division of Anatomy, Institute of Basic Medical Sciences, University of Oslo, Oslo, Norway. [11]Institute of Molecular Oncology and Functional Genomics, TranslaTUM Cancer Center, Technical University of Munich School of Medicine and Health, Munich, Germany. [12]Institute for Medical Microbiology, Immunology and Hygiene, Technical University of Munich School of Medicine and Health, Munich, Germany. [13]Immunology Program, Memorial Sloan Kettering Cancer Center, New York, NY, USA. [14]Institute of Molecular Immunology, Technical University of Munich School of Medicine and Health, Munich, Germany. [15]Department of Pathology and Immunology, Division of Clinical Pathology, Geneva Faculty of Medicine, Centre Médical Universitaire, Geneva, Switzerland. [16]Institute for Computational Biology, Helmholtz Munich, Neuherberg, Germany. [17]German Center for Infection Research (DZIF), Partner Site Munich, Munich, Germany. ✉e-mail: thomas.korn@tum.de

and ablate autoreactive thymocytes[8]. Other thymic antigen-presenting cells (APCs) contribute to the shaping of the antigen-specific thymocyte repertoire[9]. For example, conventional dendritic cells (DCs) develop intrathymically, ingest and cross-present thymic antigens, or ingest blood-borne autoantigens before they migrate to the thymus and present their cargo to thymocytes[10]. Plasmacytoid DCs are also able to pick up serum antigens, migrate to the thymus in a CCR9-dependent manner and present antigens to thymocytes[11]. More recently, B cells were identified as a population of APCs in the thymus[12,13]. After licensing through a CD40 signal, thymic B cells were shown to be able to delete thymocytes specific for a model antigen[14]. However, the non-redundant or overlapping function of thymic B cells in the deletion of autoreactive thymocytes or their diversion into the FOXP3[+] regulatory T ($T_{reg}$) cell lineage in a physiological setting is largely unclear.

Here we report that B cells present their endogenous AQP4 in the context of MHC-II to delete AQP4-specific thymocytes. In fact, AQP4 is a disease-relevant antigen that is upregulated and presented in B cells after engagement of CD40. Thymic B cells (and not mTECs) are key in tolerizing the T cell repertoire against AQP4. We propose that thymic tolerance against AQP4 and perhaps other CD40-activated B cell antigens is an efficient means to withhold T cell help from unfavourable T cell–B cell interactions and prevent their maturation into GC reactions. The failure of this tolerance mechanism might—as in the case of NMO—result in efficient T-cell-dependent autoantibody production and overt autoimmune disease.

## Negative selection of AQP4-specific T cells

The generation of pathogenic anti-AQP4 IgG (NMO-IgG) requires an antigen-specific T follicular helper ($T_{FH}$) cell response. $Aqp4^{-/-}$ mice raise a robust T cell response against the I-A[b]-restricted AQP4 epitope AQP4(201–220) (hereafter, P41), whereas WT mice do not respond to immunization with full-length AQP4 or its immunogenic peptide. This observation indicates that the T cell repertoire is tightly tolerized against AQP4[6,7]. To address the mechanism of AQP4-specific T cell tolerance, we created an AQP4(205–215) (hereafter, P41-10)–I-A[b] tetramer to directly test the frequency of AQP4-specific T cells both in the conventional T cell and FOXP3[+] $T_{reg}$ cell compartments. To assess whether radioresistant or haematopoietic cells were responsible for purging AQP4-specific T cells, we generated bone marrow chimeras in which WT or AQP4-deficient bone marrow was grafted into lethally irradiated WT or AQP4-deficient host mice in a criss-cross design. After immunization with AQP4, a P41-10–I-A[b+] T cell population was detected in $Aqp4^{-/-}$ to $Aqp4^{-/-}$ bone marrow chimeras while WT to WT and WT to $Aqp4^{-/-}$ bone marrow chimeras did not respond. Notably, $Aqp4^{-/-}$ to WT bone marrow chimeras raised a sizeable fraction of P41-10–I-A[b+] T cells (Fig. 1a). While this fraction was not as large as in $Aqp4^{-/-}$ to $Aqp4^{-/-}$ bone marrow chimeras, these data suggested that, besides non-haematopoietic cells, haematopoietic cells contributed to the negative selection of AQP4-specific T cells.

## B cells purge AQP4-specific T cell clones

Tissue-restricted antigens are promiscuously expressed and presented in an MHC-II-dependent manner in mTECs to negatively select autoreactive thymocytes. Thymic DCs usually cross-present antigens that they have previously ingested. Moreover, a subset of thymic B cells has been reported to be able to present certain model antigens to facilitate the negative selection of thymocytes specific for these model antigens[14]. Here we detected *Aqp4* mRNA in purified TECs and thymic B cells but not in thymic DCs of mice (Fig. 1b,c). Thymic B cells were strategically positioned at the corticomedullary boundary and expressed AQP4 both in mice and humans (Fig. 1d–i and Extended Data Fig. 1a–c). Although TECs expressed AQP4, and AQP4 expression was different in distinct

TEC subsets[15,16] (Extended Data Fig. 2a,b), MHC-II expression per cell was one order of magnitude higher in thymic B cells compared with any TEC subset (Extended Data Fig. 2c–e).

To narrow down the relevant source of endogenous AQP4 expression that would contribute to the purging of AQP4-specific T cells, we genetically ablated *Aqp4* in mTECs and B cells (Extended Data Fig. 2f). Using immunohistochemistry, the ablation of *Aqp4* in B cells did not disrupt the architecture of AQP4 expression in TECs (Fig. 1e). At steady state, P41-10–I-A[b+] T cells were extremely rare in WT mice (Fig. 2a). However, significantly higher absolute numbers of AQP4-specific T cells were present in the naive repertoire of global $Aqp4^{-/-}$ mice. Similarly, elevated numbers of AQP4-specific T cells were detected in B-cell-conditional *Aqp4*-deficient mice (*Mb1-cre^{KI/WT}Aqp4^{flox/flox}*; hereafter, $Aqp4^{ΔB}$) but not to the same extent in *Foxn1-cre^+Aqp4^{flox/flox}* mice (hereafter, $Aqp4^{ΔTEC}$; Fig. 2a). After immunization with P41, AQP4-specific T cells were expanded both in $Aqp4^{-/-}$ and in $Aqp4^{ΔB}$ mice but did not expand to the same degree in $Aqp4^{ΔTEC}$ mice (Fig. 2b). Together, these data indicated that B cells contributed significantly to the purging of AQP4-specific T cells by expression of endogenous AQP4. Differences in the absolute numbers of AQP4-specific T cells in $Aqp4^{ΔTEC}$ and $Aqp4^{ΔB}$ mice after immunization with P41 could be due to different phenotypes of sensitized AQP4-specific T cells in these genotypes. Indeed, the fraction of FOXP3[+] $T_{reg}$ cells was higher among P41-10–I-A[b+] T cells in $Aqp4^{ΔTEC}$ mice compared with $Aqp4^{ΔB}$ mice and resembled the fraction of FOXP3[+] cells in WT AQP4-specific T cells (Fig. 2c and Extended Data Fig. 3a). These data suggested that B cell expression of AQP4 favoured the development of self-antigen-specific $T_{reg}$ cells. Furthermore, about 50% of AQP4-specific T cells in immunized $Aqp4^{ΔB}$ mice expressed RORγt and also about 50% expressed BCL6—very similar to the situation in global $Aqp4^{-/-}$ mice and significantly more than in WT or $Aqp4^{ΔTEC}$ mice (Fig. 2c and Extended Data Fig. 3a). Thus, through expression of AQP4, B cells appeared to eliminate conventional AQP4-specific T cells that had the potential to develop into T helper 17 cells or $T_{FH}$ cells in the peripheral immune compartment after antigen-specific sensitization. By contrast, AQP4-expressing mTECs had only a small contribution to the elimination of AQP4-specific T cells and did not modulate the balance of P41-10–I-A[b+] conventional T cells versus P41-10–I-A[b+]FOXP3[+] $T_{reg}$ cells (Fig. 2c).

Given the rescue from depletion of P41-10–I-A[b+] T cells in $Aqp4^{ΔB}$ mice, we examined whether AQP4 expression in B cells alone was sufficient to eliminate AQP4-specific T cells. We therefore constructed mixed bone marrow chimeras, in which only B cells were sufficient in AQP4 in an otherwise AQP4-deficient background (Fig. 2d and Extended Data Fig. 3b,c). After immunization with AQP4(201–220), P41-10–I-A[b+] T cells were expanded, as expected, in a globally AQP4-deficient environment but were absent when only B cells were able to express AQP4 (Fig. 2e). These data indicated that B-cell-intrinsic AQP4 alone was sufficient to purge AQP4-specific T cells from the T cell repertoire.

## Thymic B cells express and present AQP4

Thymic B cells express AQP4 (Fig. 1c). It has been suggested that thymic B cells get licensed by receiving a CD40 signal[14]. To establish spatiotemporal relationships between B cell subsets in primary and secondary lymphoid tissue, we sorted CD19[+] B cells from the bone marrow, lymph nodes, spleen, blood and thymus of adult WT mice and performed single-cell RNA-sequencing (scRNA-seq) analysis. Thymic B cells constituted a cluster distinct from bone marrow B cells or secondary lymphoid tissue B cells (Fig. 3a and Extended Data Fig. 4a). A CD40 signature was present in a subset of cluster 4 cells, composed of both thymic B cells and secondary lymphoid tissue B cells exhibiting a GC signature (Fig. 3a). Trajectory inference was compatible with a putative peripheral origin of thymic B cells (Fig. 3b).

To link the trajectory of B cells with AQP4 expression, we sorted distinct B cell subsets (naive, memory, marginal zone and GC) from

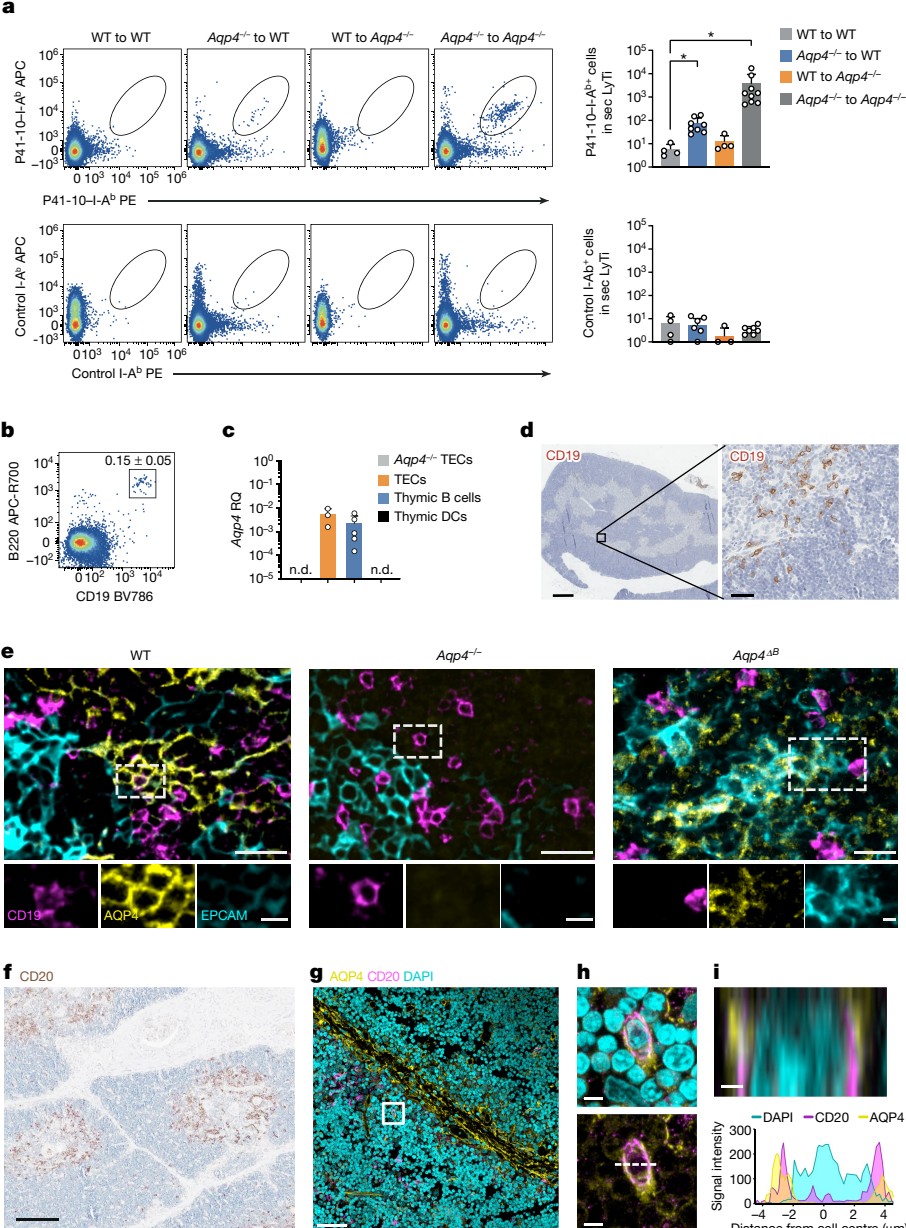

**Fig. 1 | AQP4-competent haematopoietic cells contribute to the negative selection of AQP4-specific T cells. a,** Criss-cross bone marrow (BM) chimeras of WT and *Aqp4*[−/−] mice were immunized with the I-A[b]-restricted epitope of AQP4 (P41) and tested for the frequency of AQP4-specific T cells with an AQP4(205–215)–I-A[b] tetramer (P41-10-I-A[b]) compared to a control I-A[b] tetramer (PLP(9–20)–I-A[b]; control I-A[b]). Representative cytograms and quantification of absolute P41-10-I-A[b+] T cell counts determined in the spleen and draining lymph nodes (secondary lymphoid tissue (sec LyTi)). Data are mean ± s.d. Statistical analysis was performed using two-tailed *t*-tests, with WT into WT chimeras used as the reference; *P < 0.05. The symbols indicate biological replicates. Zero values are not depicted in the bar graph due to logarithmic scaling. **b,** The fraction of thymic B cells. The mean ± s.d. thymic B cell fraction (%) is shown at the top right. *n* = 7 biological replicates. **c,** *Aqp4* expression in fluorescence-activated cell sorting (FACS)-sorted TECs (live CD45[−]EPCAM[+]), thymic B cells (live CD45[+]EPCAM[−]CD19[+]) and thymic dendritic cells (live CD45[+]EPCAM[−]CD19[−]CD11c[+]MHC-II[high]) normalized to *Aqp4* expression in

the spleen and compared their AQP4 expression with that of thymic B cell subsets defined as IgM[+]IgD[+], IgM[+]IgD[−] and IgM[−]IgD[−]. AQP4 was detectable in splenic GC B cells but was one order of magnitude more abundantly expressed in thymic B cells, in particular in thymic IgM[+]IgD[−]

astrocytes. Data are mean ± s.d. relative gene expression (RQ). The symbols indicate biological replicates; zero values are not depicted in the graph due to logarithmic scaling. n.d., not detected in three biological replicates. **d,** Representative CD19 immunostaining in WT thymus. *n* = 2 independent experiments. Scale bars, 500 μm (left) and 50 μm (right). **e,** Triple immunofluorescence staining of CD19, AQP4 and EPCAM in mouse thymus from WT, *Aqp4*[−/−] and *Aqp4*[ΔB] mice. *n* = 2 independent experiments. Individual channels aligned below a larger merged microphotograph. Scale bars, 5 μm (bottom) and 20 μm (top). **f,** B cell staining (CD20) in newborn human thymus. Scale bar, 200 μm. **g–i,** Double immunofluorescence staining for CD20 and AQP4 in the human thymus. Scale bars, 50 μm (**g**), 5 μm (**h**) and 1 μm (**i**). **g,** Overview. **h,** Magnification of the area marked by the rectangle in **g**. **i,** *z*-axis cross-section along the dashed line indicated in **h** (top). Bottom, the corresponding signal intensity profile of the immunofluorescence for CD20, AQP4 and DAPI in relation to the distance from the cell centre in μm is shown. *n* = 2 independent experiments.

B cells (Fig. 3c and Extended Data Fig. 4b). This pattern was not universal for autoantigens as myelin oligodendrocyte glycoprotein (MOG), a well-characterized CNS autoantigen, was not at all expressed in B cells including thymic B cells (Extended Data Fig. 4c).

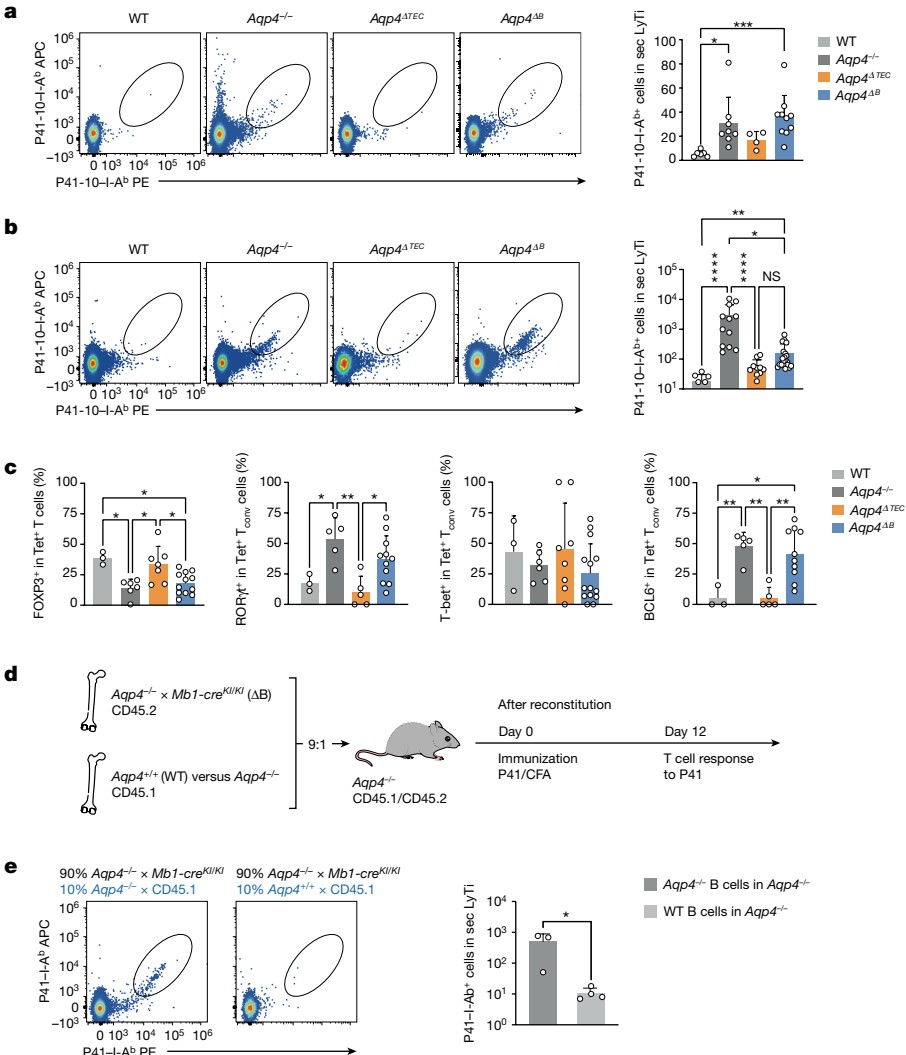

**Fig. 2 | B cells are essential in purging the T cell repertoire of AQP4-specific clones.** *Aqp4* was genetically ablated in mTECs (*Aqp4ΔTEC*) and in B cells (*Aqp4ΔB*) to identify the relevant cellular source of endogenous AQP4 expression for the negative selection of AQP4-specific T cell clones. **a,b**, Representative cytograms and quantification of absolute P41-10–I-Ab+ T cell counts determined in pooled single-cell suspensions from WT, *Aqp4−/−*, *Aqp4ΔTEC* and *Aqp4ΔB* spleen and draining lymph nodes (secondary lymphoid tissue, sec LyTi). The symbols indicate pools from individual mice. **a**, AQP4-specific T cells in the naive repertoire. **b**, AQP4-specific T cells in P41-immunized mice. Values below 10[1] are not shown due to logarithmic scaling. **c**, Quantification of the frequencies of transcription factors FOXP3, RORγt, T-bet and BCL6 in AQP4-specific (Tet+) T cells isolated from the draining lymph nodes (dLN) of P41-immunized WT, *Aqp4−/−*, *Aqp4ΔTEC* and *Aqp4ΔB* mice as measured using intracellular flow cytometry.

Data are mean ± s.d. (the symbols indicate individual mice). Statistical analysis was performed using Kruskal–Wallis tests with Dunn's post test (**a** and **b**) and one-way analysis of variance (ANOVA) with Tukey's post test (**c**); **$P < 0.01$, ***$P < 0.001$, ****$P < 0.0001$. $T_{conv}$, conventional T cells. **d**, Generation of mixed bone marrow chimeras in an AQP4-deficient environment, in which the B cell compartment was either deficient (ΔB) or sufficient for AQP4. The diagram was created using Servier Medical Art under a Creative Commons license CC BY 3.0. **e**, The fraction and quantification of AQP4-specific T cells in the systemic immune compartment in P41-immunized AQP4-deficient mice with AQP4-deficient ($n = 3$ biological replicates) or AQP4-sufficient ($n = 4$ biological replicates) B cells. Data are mean ± s.d. Statistical analysis was performed using two-tailed unpaired *t*-tests. *$P < 0.05$.

Our data indicated an overlap in the transcriptomes of peripheral B cells engaged in T cell interactions and thymic B cells. We therefore tested signals associated with T cell–B cell interactions to induce AQP4 in B cells. Stimulation with anti-CD40 induced the expression of AQP4 in naive B cells, and IL-21 further potentiated AQP4 expression in anti-CD40-stimulated B cells, while additional triggering of the BCR with anti-IgM reduced CD40-mediated AQP4 expression (Fig. 3d). Similarly, human GC (but not naive) B cells expressed AQP4, and stimulation of naive (CD19+CD27−) human B cells with CD40L induced expression of AQP4 (Fig. 3e,f). To test whether CD40-stimulated B cells were able to present their endogenous AQP4 in the context of MHC-II, we co-cultured B cells that were prestimulated with anti-CD40 plus IL-21 with a T cell hybridoma engineered to express an AQP4-specific TCR

(clone 6; Extended Data Fig. 4d). In contrast to *Aqp4−/−* B cells, which did not stimulate the AQP4-specific T cell hybridoma in the absence of exogenous antigen, WT B cells presented their endogenous AQP4 and stimulated the NFAT–GFP reporter in the AQP4-specific T cell hybridoma (Fig. 3g and Extended Data Fig. 4e). Both AQP4-deficient B cells and WT B cells were equally able to present exogenous AQP4 and, in all of the conditions, the stimulation of the hybridoma was MHC-II dependent (Fig. 3g). Notably, thymic IgM+IgD− B cells isolated ex vivo from WT but not from *Aqp4−/−* mice were directly able to stimulate the AQP4-specific hybridoma without prestimulation with CD40 and in the absence of exogenous antigen, while ex vivo-isolated splenic B cells did not activate the hybridoma in the absence of exogenous P41 (Fig. 3h). As a fraction of thymic B cells was reported to express AIRE[14],

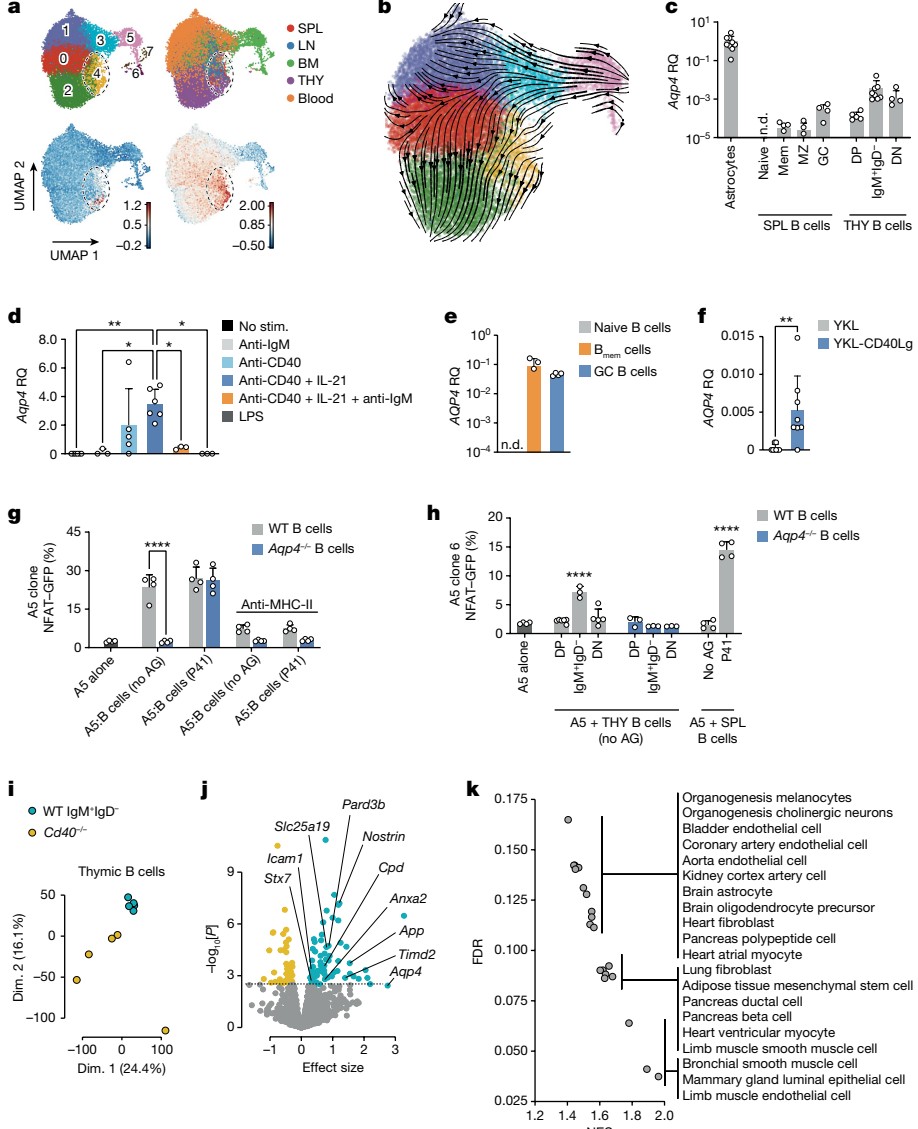

**Fig. 3 | Thymic B cells upregulate AQP4 in a CD40-dependent manner and present it to T cells in the context of MHC-II. a,b,** Uniform manifold approximation and projection (UMAP) representation of scRNA-seq data of B cells sorted from the spleen (SPL), lymph node (LN), bone marrow (BM), thymus (THY) and blood of young adult naive WT mice. **a,** Annotated Leiden clusters with a resolution of $r = 0.7$ (top left). Top right, cells colour-coded by organ (a detailed breakdown is provided in Extended Data Fig. 4a). Bottom left, the gene score based on a published gene signature associated with early CD40 responses in B cells[40]. Bottom right, the gene score based on a published gene signature of GC light-zone B cells[41,42]. The colour scale indicates relative gene score expression. Leiden cluster 4 is highlighted in all of the panels. **b,** RNA trajectory inference derived from spliced and unspliced mRNA ratios, as determined by UniTVelo. **c,** Quantification of the relative gene expression of *Aqp4* normalized to primary naive astrocytes. B cell subsets were sorted from unmanipulated WT mice. DN, double negative (IgM⁻IgD⁻); DP, double positive (IgM⁺IgD⁺); mem, memory; MZ, marginal zone; n.d., not detected. **d,** FACS-sorted CD19⁺ B cells from WT spleens were cultured and stimulated (stim.) for 2 days as indicated. Relative gene expression was normalized to control stimulation with goat anti-human IgG (H+L). **e,** Human naive (CD19⁺CD27⁻CD38⁻, naive B cells), memory (CD19⁺CD27⁺CD38⁻, B_mem cells) and GC (CD19⁺CD27⁺CD38⁺) B cells were FACS-sorted from human tonsil tissue ($n = 4$ biological replicates) and *AQP4* expression was analysed using qPCR. The symbols represent biological replicates. **f,** Naive human B cells were sorted

from peripheral blood mononuclear cells and stimulated with control fibroblastic feeder cells (YKL) or YKL cells equipped with membrane-bound CD40L (CD40Lg) ($n = 8$ biological replicates) before assessment of *AQP4* expression using qPCR. **g,h,** Quantification of NFAT–GFP expression in a coculture system with a T cell hybridoma cell line (A5 cells) engineered to express an AQP4-specific TCR and either B cells prestimulated with anti-CD40 plus IL-21 for 2 days (**g**) or thymic B cell subsets derived from WT and *Aqp4*⁻/⁻ mice at a ratio of 1:2.5 (**h**). AG, antigen. For **c–h**, data are mean ± s.d. Statistical analysis was performed using one-way ANOVA with Tukey's post test (**d**), two-tailed unpaired *t*-tests (**f**) and two-way ANOVA with Sidak's post test (**g** and **h**). The symbols indicate biological replicates. **i–k,** Total RNA was isolated from thymic B cells that were FACS-sorted from WT and *Cd40*⁻/⁻ mice ($n = 5$ biological replicates) and processed for bulk RNA-seq analysis. **i,** PCA analysis. Dim., dimension. **j,** Volcano plot of genes encoding membrane proteins. Differentially upregulated and downregulated genes in WT versus *Cd40*⁻/⁻ B cells are highlighted in blue and orange, respectively. Gene labels correspond to the differentially upregulated genes in thymic WT IgM⁺IgD⁻ B cells, which encode structural proteins with known membrane localization[43]. **k,** Gene set enrichment analysis for cell type signature genes (MSigDB M8) in WT IgM⁺IgD⁻ thymic B cells versus *Cd40*⁻/⁻ thymic B cells. A selection of significantly ($P < 0.05$, false-discovery rate (FDR) < 0.25) enriched gene sets (normalized enrichment score (NES)) is shown.

we tested whether AQP4 expression in B cells was AIRE dependent. However, even though AIRE expression paralleled AQP4 (Extended Data Fig. 4f), AIRE-deficient B cells still expressed AQP4 when they were stimulated with anti-CD40 plus IL-21 (Extended Data Fig. 4g) and were able to stimulate the AQP4-specific T cell hybridoma similarly to WT B cells (Extended Data Fig. 4h). Taken together, CD40 activation (perhaps in combination with IL-21, which is produced by CD4[+] single-positive thymocytes[17]) is the key signal for thymic B cells to express and present AQP4 to developing T cells.

It is unlikely that AQP4 is the sole disease-relevant autoantigen that is expressed and presented by thymic B cells in a CD40-dependent manner. Thus, we isolated WT IgM[+]IgD[−] thymic B cells and thymic B cells from CD40-deficient mice, which essentially lack the IgM[+]IgD[−] fraction due to the absent CD40 signal and, in this aspect, resemble thymic B cells from $Tcra^{−/−}$ mice that do not contain single-positive thymocytes capable of providing CD40L (Extended Data Fig. 5a,b). The absence of thymic B cell subsets beyond the fraction of double-positive IgM[+]IgD[+] cells was accompanied by a reduced expression of MHC-II and CD80 (Extended Data Fig. 5a). When comparing their transcriptomes, WT IgM[+]IgD[−] and $Cd40^{−/−}$ thymic B cells formed distinct clusters in the principal component analysis (PCA) (Fig. 3i), and the differentially expressed genes contained a number of disease-associated autoantigens besides AQP4 (Fig. 3j and Supplementary Table 1). In fact, age-associated autoantigens of many organ systems were enriched in thymic WT versus $Cd40^{−/−}$ B cells in gene set enrichment analyses (Fig. 3k). As expected, TECs of $Cd40^{−/−}$ mice expressed as much AQP4 as WT TECs, while $Cd40^{−/−}$ thymic B cells lacked AQP4 expression (Extended Data Fig. 5b,c). However, due to the complete lack of AQP4 expression in thymic B cells of CD40-deficient mice, the T cell repertoire of $Cd40^{−/−}$ mice was not purged of AQP4-specific TCRs and, after immunization with P41, AQP4-specific T cells expanded to the same extent in $Cd40^{−/−}$ mice as in $Aqp4^{−/−}$ mice (Extended Data Fig. 5d). In summary, these data indicate that CD40-licensed thymic B cells might be essential negative selectors not only for AQP4 but for a set of disease-relevant autoantigens in a variety of tissues.

## B cells ablate AQP4-specific thymocytes

When a T cell–B cell interaction can lead to the feeding of endogenous B cell antigens into the B-cell-intrinsic MHC-II pathway, this creates a potentially dangerous autoimmunity-prone situation if these B cells are erroneously helped by $T_{FH}$ cells. We speculated that thymic B cells are required to tolerize the T cell receptor repertoire against these B-cell-intrinsic antigens and, in fact, WT B cells were sufficient to purge the T cell repertoire of AQP4-specific reactivities in an otherwise AQP4-deficient background (Fig. 2e). By contrast, genetic ablation of $Aqp4$ in mTECs was never sufficient to render $Aqp4^{ΔTEC}$ mice susceptible to experimental autoimmune encephalomyelitis (EAE) after immunization with AQP4(201–220) even under conditions of $T_{reg}$ cell depletion (Extended Data Fig. 6a–c). However, it was still an open question whether AQP4-sufficient B cells educated the T cell compartment inside or outside the thymus. Notably, a mature T cell repertoire (derived from $Aqp4^{ΔB}$ mice and therefore containing P41-10–I-A[b+] T cells) was not depleted of AQP4-specific T cells when transferred into B-cell-sufficient $Tcra^{−/−}$ mice and instead was expanded after immunization with full-length AQP4 to induce an EAE-like disease (Extended Data Fig. 6d–f). We therefore reasoned that the negative selection of AQP4-specific TCRs by B cells must occur in the thymus, in which AQP4-specific thymocytes would not expand but go into apoptosis after cognate interaction with thymic B cells.

To directly address this hypothesis, we first measured the absolute number of AQP4-specific CD4[+] single-positive thymocytes in naive mice using our P41-10–I-A[b] tetramer. Similar to the peripheral immune compartment, B-cell-conditional but not TEC-conditional ablation of $Aqp4$ rescued the number of P41-10–I-A[b+]CD4[+] single-positive thymocytes to

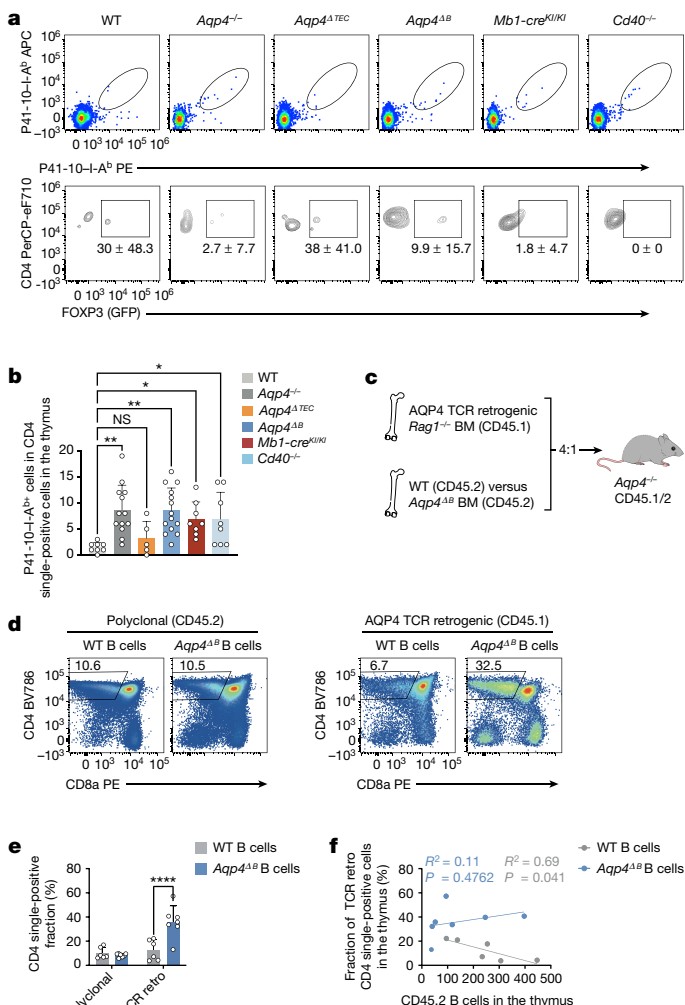

**Fig. 4 | The negative selection of AQP4-specific thymocytes is dependent on AQP4-sufficient B cells in the thymus. a**, Representative cytograms of P41-10–I-A[b+]CD4[+] single-positive thymocytes in naive WT, $Aqp4^{−/−}$, $Aqp4^{ΔTEC}$, $Aqp4^{ΔB}$, $Mb1-cre^{KI/KI}$ and $Cd40^{−/−}$ mice (top). Bottom, corresponding fractions of FOXP3 in AQP4-specific CD4[+] single-positive thymocytes. Data are mean ± s.d. **b**, Quantification of the absolute numbers of P41-10–I-A[b+] CD4[+] single-positive thymocytes. Data are mean ± s.d. (symbols indicate biological replicates). Statistical analysis was performed using one-way ANOVA with Dunnett's post test. **c**, Mixed bone marrow chimeras were generated by grafting congenically marked $Rag1^{−/−}$ bone marrow engineered to retrogenically express an AQP4-specific TCR (clone 6) along with bone marrow from either WT or $Aqp4^{ΔB}$ mice (4:1) into lethally irradiated $Aqp4^{−/−}$ recipients and tested 6 weeks after engraftment. The diagram was created using Servier Medical Art under a Creative Commons license CC BY 3.0. **d**,**e**, Representative cytograms of the polyclonal and the retrogenic (retro; AQP4-specific) thymic compartment facing a thymic environment equipped with either WT ($n = 6$ biological replicates) or AQP4-deficient B cells ($n = 7$ biological replicates) (**d**) and quantification of the thymic CD4[+] single-positive thymocyte fraction (**e**). Data are mean ± s.d. Statistical analysis was performed using two-way ANOVA with Sidak's post test. **f**, Correlation of thymic CD45.2 B cell counts with the retrogenic AQP4-specific TCR clone 6 CD4[+] single-positive fraction in the thymus. Statistical analysis was performed using Pearson's $R^2$ and simple linear regression. The individual symbols represent biological replicates.

the amount observed in the thymus of global-$Aqp4^{−/−}$ mice (Fig. 4a,b). Notably, the number of P41-10–I-A[b+]CD4[+] single-positive thymocytes was also significantly higher in the thymus of B-cell-deficient ($Mb1-cre^{KI/KI}$) and $Cd40^{−/−}$ mice compared with the thymus of WT mice (Fig. 4a,b). Together, these data suggested that licensed AQP4-sufficient

B cells contributed substantially to the thymic negative selection of AQP4-specific thymocytes. However, as we found a sizeable fraction of P41-10–I-A$^b$+CD4$^+$ single-positive thymocytes that expressed FOXP3 in $Aqp4^{ΔTEC}$ mice, in which B cells are a relevant alternative source of AQP4, it is probable that B cells may also contribute to the diversion of AQP4-specific thymocytes into the T$_{reg}$ cell lineage (Fig. 4a). To further corroborate the major function of thymic B cells as negative selectors of AQP4-specific thymocytes, we constructed a set of mixed bone marrow chimeras: congenically marked $Rag1^{-/-}$ bone marrow retrogenically expressing the AQP4-specific TCR clone 6 was mixed (4:1) with bone marrow from either WT mice or $Aqp4^{ΔB}$ mice and grafted into lethally irradiated $Aqp4^{-/-}$ host mice (Fig. 4c). In these compound mice, the clone 6 TCR$^+$ thymocytes would face a thymic environment with either WT or AQP4-deficient B cells. After reconstitution, we compared the retrogenic thymic compartment with the co-grafted polyclonal thymic compartment. Although the retrogenic CD4$^+$ single-positive compartment was significantly attrited in the presence of WT B cells, it was left untouched in the presence of AQP4-deficient B cells (Fig. 4d,e). In fact, the relative lack of TCR$^{high}$ and CD5$^{high}$ cells in the retrogenic CD4$^+$CD8$^+$ compartment in the presence of WT B cells compared with $Aqp4^{-/-}$ B cells suggested that agonist-mediated deletion might already start in the double-positive compartment (Extended Data Fig. 7a–c). The attrition of the single-positive compartment was more pronounced in cases in which there were more WT B cells present in the thymus (Fig. 4f). The size of the polyclonal CD4$^+$ single-positive compartment was not significantly affected by the perturbation of the thymic B cell genotype (Fig. 4d,e). Consistent with AQP4 expression in B cells being independent of AIRE, AIRE-deficient B cells did not rescue AQP4-specific thymocytes from deletion (Extended Data Fig. 7d,e). In summary, these data are consistent with the idea that a cognate interaction of AQP4-specific thymocytes with thymic B cells contributed to their deletion.

## AQP4-specific GC responses in $Aqp4^{ΔB}$ mice

Our data argue in favour of the concept that AQP4-specific TCRs are deleted from the thymic repertoire by B cells because AQP4 is expressed and presented by CD40-activated thymic B cells. A failure of this mechanism might therefore allow for AQP4-specific T cell–B cell interactions as AQP4-specific T$_{FH}$ cells would not be purged from the T cell repertoire as efficiently by mTECs alone. Indeed, immunization with full-length AQP4 resulted in the generation of T$_{FH}$ cells and GC B cells in the spleens of $Aqp4^{-/-}$ and $Aqp4^{ΔB}$ mice, which were almost entirely absent in WT mice (Extended Data Fig. 8a). Ablation of $Aqp4$ in B cells rescued at least partially the generation of T$_{FH}$ cells and GC B cells in the spleen after sensitization with full-length AQP4 (Extended Data Fig. 8b,c). Accordingly, $Aqp4^{ΔB}$ mice showed a robust serum response to AQP4 that was similar to the AQP4-specific serum response in $Aqp4^{-/-}$ mice while, as expected, WT mice did not raise AQP4-specific antibodies (Extended Data Fig. 8d,e). As both GC formation and antigen-specific serum responses were similar in WT and $Aqp4^{-/-}$ as well as $Aqp4^{ΔB}$ mice in response to recombinant MOG protein (Extended Data Fig. 8f,g), these data indicate that AQP4 has a very specific status as a paradigmatic autoantigen that is dependent on thymic B cells for complete tolerance. As the transcriptome analysis of thymic WT versus $Cd40^{-/-}$ B cells revealed a number of additional disease-relevant autoantigens, we tested whether a mature T cell repertoire educated in the absence of B cells would be prone to help autoantibody formation. We therefore transferred the CD4$^+$ T cell compartment of WT or $Mb1$-$cre^{KI/KI}$ mice into $Tcra^{-/-}$ mice, which lack T cells but are B cell sufficient, and immunized the hosts with a strong adjuvant (complete Freund's adjuvant (CFA)) in the absence of exogenously administered autoantigens. After 32 days, we collected serum from these mice and tested it for potential autoreactivity on organ sections of $Rag1^{-/-}$ mice (Extended Data Fig. 9a). While serum derived from WT-transferred $Tcra^{-/-}$ mice did not essentially stain $Rag1^{-/-}$ central nervous system (CNS), kidney or skin tissue, serum

collected from $Mb1$-$cre^{KI/KI}$-transferred $Tcra^{-/-}$ mice bound to ependyma and vessel structures in the CNS, to kidney collecting ducts and to epidermal structures in the skin (Extended Data Fig. 9b). Together, these data support the idea that thymic B cells might crucially contribute to tolerizing the T cell repertoire against a set of antigens that are potential targets of autoantibodies.

## Autoimmune astrocytopathy in $Aqp4^{ΔB}$ mice

Finally, we wondered whether the rescue of the AQP4-specific T cell repertoire in $Aqp4^{ΔB}$ mice was sufficient to result in clinical disease after antigen-specific sensitization. Thus, $Aqp4^{ΔTEC}$ and $Aqp4^{ΔB}$ mice were immunized with P41 in CFA and followed for signs of EAE. The disease incidence and severity was significantly higher in $Aqp4^{ΔB}$ mice compared with in $Aqp4^{ΔTEC}$ mice, which essentially behaved like WT control mice (Fig. 5a). To assess the immunopathology of EAE induced by P41, we compared the P41-induced disease with MOG(35–55)-EAE, which is well characterized in the C57BL/6 background. Although the incidence of P41-induced EAE was lower than the incidence of MOG(35–55)-induced EAE in $Aqp4^{ΔB}$ mice (Fig. 5b), they were phenotypically very similar (Fig. 5c and Extended Data Table 1). However, the lesion distribution in P41-induced EAE—while similar to MOG(35–55)-induced EAE in the spinal cord—was different in the brain of $Aqp4^{ΔB}$ mice. Here, the diencephalon showed prominent infiltrates when few lesions were observed in that region in MOG-induced EAE (Extended Data Fig. 10). Notably, retinal infiltrates were detected in P41-induced EAE, which were never found in MOG(35–55)-induced EAE in $Aqp4^{ΔB}$ mice[18] (Fig. 5d). Moreover, in contrast to MOG-peptide-induced EAE, P41-immunization resulted in the loss of AQP4 reactivity (and concomitant GFAP reactivity) at the glia limitans, in ventricular lining cells (tanycytes) and, to a lesser extent, in choroid plexus cells (Extended Data Fig. 10a,b). Together, these data indicate that the AQP4-specific T cell precursor frequency in $Aqp4^{ΔB}$ mice is sufficient to cause overt autoimmune disease after an appropriate trigger. The lesion distribution in P41-induced EAE of $Aqp4^{ΔB}$ mice was consistent with the expression pattern of the target protein (AQP4) in the CNS.

## Discussion

The presentation of AQP4 by thymic B cells is necessary and sufficient to purge the TCR repertoire of AQP4-reactive TCRs. Lack of AQP4 expression in B cells rescues the absolute number of AQP4-specific T cells in secondary lymphoid tissues to the level of $Aqp4^{-/-}$ mice, suggesting that thymic B cells might primarily eliminate AQP4-specific thymocytes rather than promote their diversion into the T$_{reg}$ cell lineage. Only when thymic B cells fail to express AQP4 is an AQP4-specific T$_{FH}$ cell response raised in secondary lymphoid tissues, and an AQP4-targeted astrocytopathy occurs.

Thymic B cells have been described previously[19,20]. Both in mice and in humans, thymic B cells are derived from mature peripheral B cells that get licensed in the thymus through CD40L-expressing single-positive thymocytes that also produce IL-21[14,17]. This process might occur early in life, might in part be dependent on type III interferons produced by epithelial cells and might result in the establishment of a residency program in thymic B cells[21]. The functional relevance of thymic B cells has not been known, although it was shown for model antigens that they are, in principle, able to shape the TCR repertoire both in terms of negative selection[12,14] and T$_{reg}$ cell induction[21,22]. When B cells are engineered to express an autoreactive BCR, they can capture the relevant antigen, present it in the thymus and educate the thymocyte repertoire accordingly[12]. By contrast, here we show that thymic B cells express AQP4 and a whole set of other self-antigens in a CD40-dependent manner that is then presented in the context of MHC-II. Although some of these B cell self-antigens might depend on AIRE[14], AQP4 expression and presentation in thymic B cells is independent of AIRE. Although B lymphoid cells

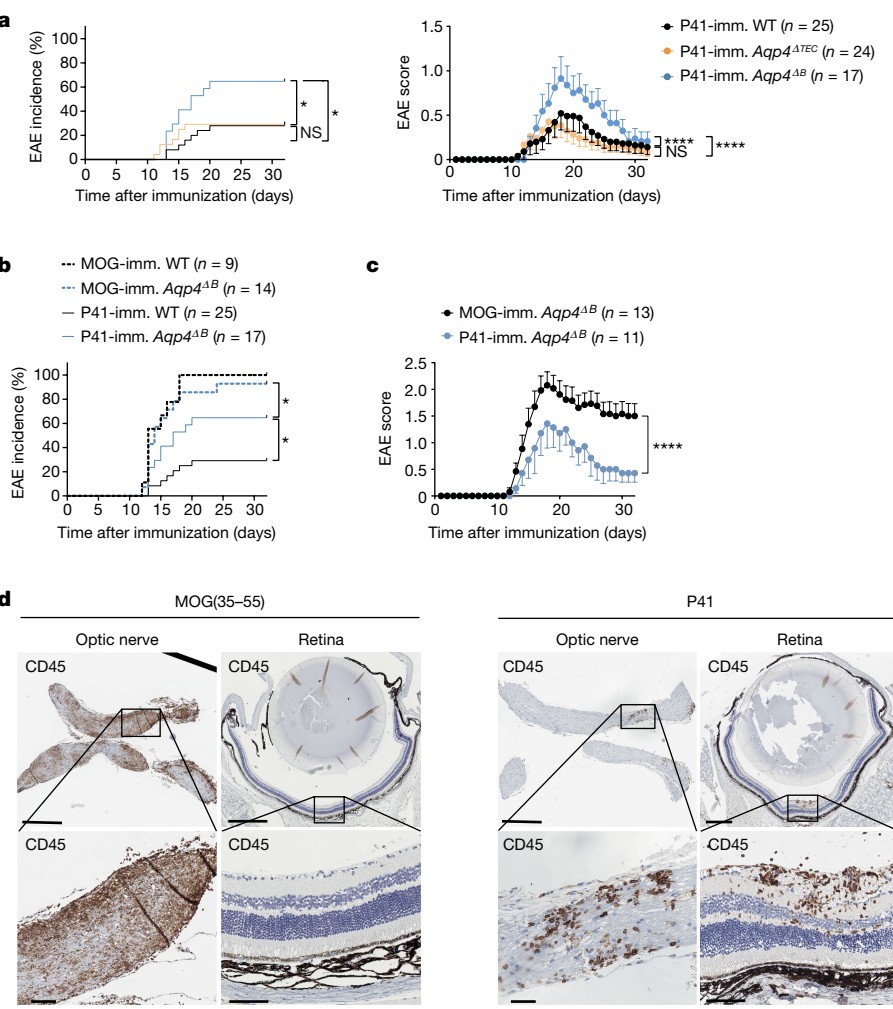

**Fig. 5 | AQP4-specific T cell precursor frequencies in *Aqp4^ΔB* mice, but not in *Aqp4^ΔTEC* mice, are sufficient to cause overt autoimmune disease in response to an antigen-specific trigger.** In contrast to WT mice and *Aqp4^ΔTEC* mice, *Aqp4^ΔB* mice were susceptible to EAE after immunization with P41 in CFA (see also Extended Data Fig. 6a–c). **a**, EAE incidence (left) and mean ± s.e.m. disease severity (right) in all P41-immunized WT, *Aqp4^ΔTEC* and *Aqp4^ΔB* mice. **b**, EAE incidence in all MOG(35–55)-immunized and P41-immunized WT mice and *Aqp4^ΔB* mice. **c**, The mean ± s.e.m. disease severity in clinically sick MOG(35–55)-immunized and clinically sick P41-immunized *Aqp4^ΔB* mice. Statistical analysis was performed using Mantel–Cox log-rank tests and two-way ANOVA with Sidak's post test to compare incidences and disease course, respectively. Only the relevant tests are indicated in **b** for legibility. **d**, Representative CD45 stainings of the optic nerve and retina in MOG(35–55)-immunized (left) and P41-immunized *Aqp4^ΔB* mice (right) at the peak of EAE. *n* = 2 independent experiments. Scale bars, 500 μm (top) and 50 μm (bottom).

have the ability to channel cell-intrinsic plasma membrane proteins into their MHC-II pathway[23,24], it remains to be elucidated whether and how AQP4 gains access to the endosomal compartment of B cells for MHC-II loading. While autophagy-dependent MHC-II loading has been shown for model antigens tagged to cytoplasmic compartments[25,26], endogenous plasma membrane antigens might be internalized by various pathways in B cells and gain access to the MHC-II pathway[27,28].

Here we show that the T cell repertoire is critically tolerized against a disease-relevant autoantigen (AQP4) by thymic B cells—a disease mechanism that is previously undescribed in NMO. The immunopathology in NMO is mediated by autoantibodies against AQP4 that bind to their target antigen and induce astrocyte lysis[2]. Our data prompt the concept that thymic negative selection against CD40-induced self-antigens in B cells is a mechanism to control self-destructive T cell–B cell interactions in the systemic immune compartment. In the systemic immune compartment, CD40-activated B cells that are not at the same time triggered in a cognate manner by their BCR (which downregulates the expression of CD40-induced self-antigens again) present CD40-induced intrinsic autoantigens. When CD40-signature antigen-specific T cells are removed from the repertoire in the thymus,

these aberrantly activated B cells will never be helped by T cells and will therefore not proceed into a GC reaction. This mode of tolerance against self-antigens expressed and presented by B cells appears to be a tailored and also non-redundant way to prevent autoimmunity against CD40-induced B cell antigens, which, besides AQP4, comprise additional potentially disease-relevant autoantigens such as *Anxa2*, *App* and *Cpd*. Autoantibodies to ANXA2 and APP are associated with antiphospholipid syndrome[29,30] and cerebral amyloid angiopathy-related inflammation[31]. Patients with hyper-IgM-syndrome due to mutations in *CD40* or *CD40L* counterintuitively experience a variety of autoimmune diseases, including autoimmune nephritis, hepatitis and discoid lupus erythematosus[32], and thymic selection has been suspected to be aberrant in these patients[33,34]. The fact that NMO is more than by chance associated with other antibody-mediated autoimmune diseases, including myasthenia gravis and Sjogren's syndrome[35,36], is consistent with the failure of a more universal tolerance checkpoint in this disease. Notably, AQP4 is not among the genes differentially expressed in WT versus AIRE-deficient mTECs[37]. Moreover, AQP4 is not a tissue-restricted antigen in the strict sense as it is expressed in a variety of tissues, including the stomach, muscle, kidney and CNS.

Thus, we propose that AQP4 is a paradigmatic member of a group of self-antigens that are tolerized by thymic B cells. Our study supports the concept that mTECs alone (but not thymic B cells) might fail to make these CD40-induced autoantigens, including AQP4(201–220), visible to the immune system for tolerance induction under steady-state conditions[38]. Conversely, AQP4-expressing B cells are sufficient to establish an AQP4-tolerant TCR repertoire even in the absence of AQP4 expression by mTECs.

Nevertheless, various TEC subsets express AQP4, and it is possible that 'TEC mimetics' that express AQP4[15] interact with thymic B cells to shape the AQP4-specific TCR repertoire. Even though our data would not support an antigen transfer from TECs to B cells, the temporospatial interplay between TECs and thymic B cells needs further analysis. In secondary lymphoid tissue, mature AQP4-specific T cells are not deleted by WT B cells that express AQP4 after immunization with full-length AQP4 but rather expand and induce encephalomyelitis. Thus, an extrathymic tolerization by CD40-activated B cells in secondary lymphoid tissues as has been described for extrathymic AIRE-expressing lymph node stroma cells[39] is unlikely.

In summary, the immune system is tolerized against the NMO-relevant autoantigen AQP4 by thymic B cells that present their endogenous AQP4 to AQP4-reactive thymocytes. AQP4 is a membrane antigen that may be a model antigen for a group of B-cell-associated autoantigens, for which the very B cells cover tolerization of the T cell repertoire in the thymus to prevent inappropriate help by $T_{FH}$ cells during GC reactions in secondary lymphoid tissues.

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

## Methods

### Mice

C57BL/6J, *Aire*<sup>flox/flox</sup>, *Rag1*<sup>−/−</sup>, *Cd40*<sup>−/−</sup> and *Tcra*<sup>−/−</sup> mice were obtained from Jackson Laboratories and bred in our facility. *Aqp4*<sup>−/−</sup> mice were provided by A. Verkman[44], *Aqp4*<sup>flox/flox</sup> mice[45] were provided by O. P. Ottersen, *Foxn1-cre* mice[46] were provided by L. Klein, *Mb1-cre* mice[47] were provided by M. Schmidt-Supprian and DEREG mice[48] were provided by Tim Sparwasser. To generate mice with cell-type-specific excision of *loxP*-flanked cassettes, mice were bred with respective *cre* mice as specified. All mouse strains were on the C57BL/6J background. Mice were housed in a pathogen-free facility at the Technical University of Munich. All experimental protocols were approved by the standing committee for experimentation with laboratory animals of the Bavarian state authorities and performed in accordance with the corresponding guidelines (ROB-55.2-2532.Vet_02-17-234, ROB-55.2-2532.Vet_02-20-01, ROB-55.2-2532.Vet_02-20-23, ROB-55.2-2532.Vet_02-21-154).

### Generation of bone marrow chimeras

For the generation of bone marrow chimeras, recipient mice were lethally irradiated. A total dose of 9 Gray (Gy) was delivered as two 4.5 Gy doses 3 h apart. When *Rag1*<sup>−/−</sup> mice were used as bone marrow recipients, they were irradiated with 4.5 Gy only once. A total of $5 \times 10^6$ bone marrow cells was injected intravenously (i.v.) into recipients 1 day after irradiation. For generation of mixed bone marrow chimeras, bone marrow from two distinct donor mice was pooled and injected as specified. Reconstituted mice were maintained on antibiotic water (enrofloxacin, Bayer, 0.1 mg ml<sup>−1</sup>) for 2 weeks after cell transfer. Reconstitution of the haematopoietic system was tested in the peripheral blood.

### Antigens

Mouse MOG(35–55) (MEVGWYRSPFSRVVHLYRNGK) and mouse AQP4(201–220) (HLFAINYTGASMNPARSFGP) were synthesized by Auspep or Biotrend, respectively. Human MOG protein was obtained from Biotrend, and mouse AQP4 protein was produced using a baculovirus-insect cell expression system and purified as previously described[6].

### EAE induction

Mice were immunized subcutaneously at the base of the tail with 200 μl of an emulsion containing 200 μg of MOG(35–55), 200 μg of AQP4(201–220), 100 μg of full-length AQP4 or 100 μg of full-length human MOG, all dissolved in PBS and emulsified with 250 μg *Mycobacterium tuberculosis* H37Ra (BD Difco) in mineral oil (CFA). Moreover, mice received 200 ng pertussis toxin (Sigma-Aldrich, P7208) i.v. on days 0 and 2 after immunization. Clinical signs of disease were monitored daily with scores as follows: 0, no disease; 1, loss of tail tone; 2, impaired righting; 3, paralysis of both hind limbs; 4, tetraplegia; 5, moribund state[49].

### Preparation of single-cell suspensions

Lymph nodes, spleens and thymi were passed through a 100 μm cell strainer (Greiner Bio-One), followed by gravity centrifugation (400*g*, 4 °C, 10 min). Spleen samples underwent erythrocyte lysis using BD Pharm Lyse (BD Biosciences).

### Flow cytometry

Single-cell suspensions of lymphoid tissues were incubated with LIVE/DEAD fixable dyes (Aqua; 405 nm excitation) and mouse Fc Block in phosphate-buffered saline (PBS) for 15 min on ice. Cells were washed with FACS buffer (2% FCS in PBS) and incubated with antibodies against surface markers for 30 min on ice. For intracellular staining, cells were additionally fixed and permeabilized (Cytofix/Cytoperm and Perm/Wash Buffer; BD Biosciences) and stained with antibodies against intracellular markers overnight. A list of all of the antibodies used is provided in Supplementary Table 2.

Flow cytometry analysis was performed on the CytoFLEX flow cytometer (Beckman Coulter) with CytExpert (v.2.3.1.22) software or a FACSAria III (BD Biosciences) system with the BD FACSDiva (v.8.0.1) software, and flow cytometry data were analysed using FlowJo (v.10.5.1) software (BD Biosciences).

### I-A<sup>b</sup>–tetramer staining

For I-A<sup>b</sup>–AQP4(205–215) and I-A<sup>b</sup>–PLP(9–20) tetramer stainings, I-A<sup>b</sup> tetramers were produced as described previously[50,51]. Cells were treated with 0.7 U ml<sup>−1</sup> of neuraminidase (Sigma-Aldrich, N-2133) and 10 nM of dasatinib (Selleckchem) for 30 min at 37 °C for 30 min and 5% $CO_2$, washed twice and treated with Fc block for 15 min on ice and subsequently stained with I-A<sup>b</sup> tetramers for 2 h at room temperature with repeated resuspension, and the cells were finally surface and intracellularly stained for flow cytometry analysis. Thymocytes from naive mice were also enriched using anti-PE and anti-APC magnetic-activated cell sorting (MACS, Miltenyi Biotec) magnetic beads according to the manufacturer's instructions before flow cytometry analysis.

### Isolation of thymic APCs

TECs were isolated as described previously[52]. After euthanizing mice under deep anaesthesia by intracardial perfusion with PBS, thymi were dissected thoroughly to avoid blood contamination and adhering lymph nodes were removed immediately. Dissected thymi were placed into a six-well plate containing an enzyme solution with RPMI, 0.05% liberase TH (Sigma-Aldrich) and 100 U ml<sup>−1</sup> DNase I (Sigma-Aldrich), and incubated at 37 °C with repeated mechanical dissociation by gently pipetting. After approximately 60 min, the supernatants were pooled, washed and passed through a 100 μm filter. Thymocytes were depleted with CD90.2 MACS beads according to the manufacturer's instructions. Finally, TECs were FACS-sorted on live CD45<sup>−</sup>EPCAM<sup>+</sup>, thymic B cells on live CD45<sup>+</sup>EPCAM<sup>−</sup>CD19<sup>+</sup> and thymic DCs on CD45<sup>+</sup>EPCAM<sup>−</sup>CD19<sup>−</sup>CD11c<sup>+</sup>MHC-II<sup>high</sup> into PBS with 2% BSA.

### Isolation of primary astrocytes

Primary astrocytes were isolated as previously described[53]. Brains from neonatal C57Bl/6J mice aged 1 to 3 days were dissected and cleaned from meninges, digested with DNase I (1 mg ml<sup>−1</sup>) and 0.25% trypsin-ethylenediaminetetraacetic acid (EDTA, Calbiochem) for 15 min, and passed through a cell strainer (70 μm). Single-cell suspensions were cultured at 37 °C on 175 cm<sup>2</sup> cell culture flasks coated with 2 μg ml<sup>−1</sup> poly-L-lysine (Sigma-Aldrich). After 7–10 days, the mixed glia cell culture reached confluence, and microglia were removed by sequentially shaking at 180 rpm for 30 min and 220 rpm for 2 h, changing the medium between and after the shaking steps.

### qPCR

Total RNA was isolated from sorted cells using the RNAeasy Mini kit (Qiagen). The isolated RNA was transcribed into cDNA using the High Capacity cDNA Reverse Transcription Kit (Thermo Fisher Scientific) according to the manufacturer's instructions. Probes were purchased from Life Technologies and the assays were performed using the TaqMan Fast Advanced Master mix (Thermo Fisher Scientific) on 384-well reaction plates (Life Technologies). The quantitative PCR (qPCR) was performed on the Quant Studio 5 system (Life Technologies). In all of the experiments, *GAPDH* (Mm99999915_g1 and Hs00230829_m1, Thermo Fisher Scientific) was used as a reference gene to calibrate gene expression.

### Histological analysis of mouse tissue

Mice were euthanized under deep anaesthesia by intracardial perfusion with PBS followed by perfusion with 4% (w/v) paraformaldehyde (PFA) dissolved in PBS. All of the organs were removed and fixed in 4% PFA overnight. Vertebral columns, including the spinal cords, were additionally decalcified with Osteosoft (Sigma-Aldrich) for 72 h before paraffin embedding; 2-μm-thick sections were prepared.

Immunohistochemistry was performed using the Leica Bond Rxm System with the Polymer Refine detection kit (Leica). A list of all of the antibodies used is provided in Supplementary Table 2. DAB was used as chromogen, and counterstaining was performed with haematoxylin. The slides were then scanned on the Leica AT2 system, and the images were analysed using QuPath v.0.3.2 (https://qupath.github.io, University of Edinburgh, Scotland).

Quantification of histological samples was performed automatically with computer-assisted algorithms provided by QuPath. To detect the total cell counts, regions of interest were annotated and analysed automatically by positive nuclear detection. All annotations were performed in a blinded manner.

### Histological analysis of human tissue

Human newborn thymic tissue was obtained from A. Büttner (approval by the local ethics committee, A2023-0038). Immunofluorescence was performed using EDTA buffer and steam cooking for antigen retrieval, followed by standard protocols. A list of all of the antibodies used is provided in Supplementary Table 2.

After deparaffinization, immunohistochemistry with anti-CD20 antibodies (DAKO) was conducted using 4 μm serial sections from formalin-fixed, paraffin-embedded (FFPE) tissues according to a standard protocol. Counterstaining was performed with 50% Gill's haematoxylin 1 (American MasterTech), bluing with tap water and 0.02% ammonium hydroxide water. Slides were mounted with xylene and EcoMount (EcoMount, Biocare Medical).

To test colocalization of CD20 and AQP4, human thymus sections were sequentially costained for both markers after deparaffinization and blocking. Stained sections were analysed on an inverted TCS SP8 confocal microscope (Leica) using a HC PL APO CS2 ×40/1.30 NA objective. Confocal images were acquired as tiled image stacks with 1 μm $z$ spacing and 3D image volumes were analysed using Imaris v.9.7 (Oxford Instruments).

### B cell scRNA-seq analysis

In addition to surface staining for flow cytometry analysis, single-cell suspensions from different immune compartments (spleen, lymph node, bone marrow, thymus, blood) were labelled with TotalSeq-C anti-mouse Hashtags 1–3 and 5–6 (M1/42; 39-F11, BioLegend, 1:100 for all) according to the manufacturer's instructions. A dump channel was defined by CD3, CD11b, F4/80 and NK1.1. From each compartment, 12.000 live dump⁻CD19⁺ cells were sorted separately into an FCS-coated 96-well plate.

scRNA-seq, scBCR-seq and cell hashing libraries were prepared using the 10x Chromium Single Cell 5′ Solution (Chromium Next GEM Single Cell VDJ v1.1 with feature barcoding technology for cell surface protein, 10x Genomics) as described previously[54]. In brief, sorted cells were transferred to the Chromium Next Gem chip (10x Genomics), and partitioning was performed automatically by the Chromium Controller (10x Genomics). Library preparation was performed according to the manufacturer's instructions. For quality control and quantification, a Bioanalyzer 2100 (Agilent Technologies) was used. Finally, libraries were sequenced on the Illumina NovaSeq 6000 system as provided by Novogene.

### Single-cell sequencing data processing

scRNA-seq reads, supplied by Novogene, were aligned to mouse reference genome mm10-2020-A using the Cell Ranger (v.7.1.0)[55] count pipeline with the option 'include introns' disabled. Preprocessing, clustering, annotation and visualization were performed using SCANPY (v.1.9.2)[56] according to established guidelines[57]. Specifically, for quality control, cells with more than 15% mitochondrial gene counts were excluded, as well as cells with more than 2% haemoglobin gene counts and cells with less than 5% ribosomal gene counts. Cells with less than 200 detected genes and cells with more than $1 \times 10^4$ counts per cell were also removed. Doublet exclusion was performed using scrublet (v.0.2.3)[58]. In total,

quality control removed 1,926 cells. A total of 16,783 genes detected in less than three cells was excluded. To focus on subset variability beyond B cell specificity, 254 variable B cell receptor chain genes detected in the dataset were excluded. Gene counts were normalized to $1 \times 10^4$ total counts per cell, and variance was stabilized by log1p transformation. Data were not scaled to preserve the original weighing of gene expression and no regression was applied due to the low impact of confounding parameters in most compartments (that is, proliferation scoring and mitochondrial gene content). Highly variable genes for clustering were determined using the SCANPY default settings based on normalized dispersion, batched by organ, yielding 1,291 genes for further analysis. On the basis of these genes, principal component analysis was performed using the default SCANPY settings. A neighbourhood graph was computed based on 30 principal components and 20 neighbours. Clustering was performed using the Leiden algorithm (leidenalg package, v.0.9.1) with a resolution of $r = 0.7$ and UMAP dimensionality reduction was computed using the default SCANPY settings.

Gene signature scores were calculated in SCANPY using the default settings. Relevant gene signatures were as follows: genes upregulated by ex vivo B cells 2 h after exposure to CD40L[40]: *Tnfaip3, Bcl2l1, Bcl2a1a, Gadd45b, Gadd45g, Fas, Slc16a1, Slc19a1, Prps1, St3gal6, Hmgcr, Ldlr, Fasn, Ccnd2, Stat5a, Nfkb1, Myc, Irf4, Cr2, Cd44, Il2ra, Ebi3, Lilrb4a, Fcer2a, Gpr65, Il1b, Traf1, Nfkbia, Nfkbib, Marcksl1, Icam1, Cd83, Jarid2, Bhlhe40, Gm4736*\* and *Tfg*. Gene signature of mouse splenic GC light zone B cells[41,42]: *Cd52, Cd83, Hspd1, Ran, Mif, Atp5b, Myc, Dkc1, Lrrc58* and *H2-Aa*. Genes annotated with an asterisk were part of the published signature but not detected in our dataset.

For cell trajectory inference, spliced/unspliced reads were generated from Cell Ranger-aligned sequences using the velocyto (v.0.17.17) run10x pipeline[59]. Data were merged with gene expression analysis outlined above using scvelo (v.0.2.5)[60] and trajectories were derived using UniTVelo (v.0.2.5.2)[61] configured to run the model based on 1,500 top variable genes. For trajectory inference, the highly proliferative and transcriptionally active bone marrow clusters 6 and 7, respectively, were excluded from the dataset.

### Isolation of B cell subsets

Single-cell suspensions from primary and secondary lymphoid tissues were enriched for CD19 with MACS beads according to the manufacturer's instructions, followed by surface staining for flow cytometry cell sorting. For sorting CD19⁺ cells from thymi and bone marrow, we defined three distinct groups on the basis of the expression of IgM and IgD. Double-positive cells were defined as live CD19⁺B220⁺IgM⁺IgD⁺ cells. Furthermore, we isolated CD19⁺B220⁺IgM⁺IgD⁻ and double-negative B cells, defined as CD19⁺B220⁺IgM⁻IgD⁻ cells. For sorting CD19⁺ cells from secondary lymphoid tissues, we defined four groups on the basis of their maturation status. Naive B cells were specified as live CD19⁺B220⁺IgD⁺CD21⁻CD95⁻GL7⁻ cells, GC B cells as live CD19⁺B220⁺CD95⁺GL7⁺ cells, marginal zone B cells as live CD19⁺B220⁺IgD⁻CD21⁺CD95⁻GL7⁻ cells and memory B cells as live CD19⁺B220⁺IgD⁻CD21⁻CD95⁻GL7⁻ cells. RNA from sorted cells was immediately isolated and processed for qPCR analysis as described above.

### Primary mouse B cell cultures

Single-cell suspensions from secondary lymphoid tissues were FACS-sorted for live CD19⁺ cells and cultured at 37 °C and 5% $CO_2$ for 2 days before RNA was isolated for qPCR. For stimulation, different combinations comprising 50 μg ml⁻¹ anti-mouse CD40 (FGK4.5, BioX-Cell), 20 ng μl⁻¹ recombinant IL-21 (Miltenyi Biotec), 10 μg ml⁻¹ goat anti-mouse IgG + IgM (H+L), 10 μg ml⁻¹ goat anti-human IgG (H+L, both Jackson Immuno Research) and 1 μg ml⁻¹ LPS were used.

### Primary human B cell cultures and stimulation

Human tonsillar tissue was obtained from routine tonsillectomies by the Department of Otorhinolaryngology of the University Hospital

Klinikum rechts der Isar of the Technical University of Munich School of Medicine with patients' informed consent. Single-cell suspensions were prepared from freshly collected human tonsil tissue and frozen. In brief, tonsil tissue was cut into small pieces and disaggregated through a cell strainer. After washing, cells were purified using Histopaque-1077 Hybri-Max (Sigma-Aldrich) gradient centrifugation according to the manufacturer's instructions. Single-cell suspensions from human tonsils were FACS-sorted for live CD19[+] cells and cultured at 37 °C and 5% $CO_2$ for 2 days before RNA was isolated for qPCR. For sorting different B cell subsets, we defined three distinct groups based on the expression of CD38, CD27, IgD and CD10. Human naive B cells were defined as live CD3[−]CD19[+]CD27[−]CD38[+] cells, memory B cells as live CD3[−]CD19[+]CD27[+]CD38[−] cells and GC B cells as live CD3[−]CD19[+]CD27[+]CD38[+] cells. RNA from sorted cells was isolated and processed for qPCR analysis as described above.

For the in vitro stimulation of human CD19[+] cells, we used a coculture system with immortalized follicular dendritic cells (YKL) that were equipped with membrane-bound CD40L (CD40Lg) kindly provided by D. Hodson[62]. CD15 MACS-beads-depleted human peripheral blood mononuclear cells were FACS-sorted for live CD3[−]CD19[+]CD27[−] cells and seeded on irradiated YKL cell layers for 5 days. For comparability, FACS-sorted cells were also seeded onto control YKL cells transduced with an empty vector instead of the CD40L vector. After 5 days, RNA was isolated and processed for qPCR analysis as described above.

## Recall assay

To test AQP4-specific recall responses, we used a system established previously[63], based on a T cell hybridoma cell line (A5 cells) that was equipped with a GFP reporter linked to NFAT—a downstream transcription factor of the IL-2 signalling pathway[64]. We transfected these cells with a high-affinity AQP4-reactive TCR (clone 4 or clone 6) to test APCs for their endogenous presentation of AQP4. The generation of AQP4-TCR-transduced A5 cells is described in "Transduction of A5 cells with AQP4-specific TCRs" below. As controls, we added either P41 exogenously (0.3 and 1.0 μg ml[−1]), anti-CD3 (1 μg ml[−1]) and/or anti-mouse MHC-II (I-A and I-E) blocking antibody (5 μg ml[−1]) to the coculture. We determined the fraction of NFAT−GFP[+] cells after 20–24 h by flow cytometry analysis.

## Bulk RNA-seq

Total RNA was isolated from FACS-sorted whole-thymic B cells and thymic B cell subsets using AmpureXP beads (Beckman Coulter). Library preparation for bulk-sequencing of poly(A)-RNA was performed as described previously[65]. In brief, barcoded cDNA of each sample was generated with Maxima RT polymerase (Thermo Fisher Scientific, EP0742) using oligo-dT primers containing barcodes, unique molecular identifiers (UMIs) and an adapter. The 5′-ends of the cDNAs were extended by a template switch oligo (TSO) and full-length cDNA was amplified with primers binding to the TSO-site and the adapter. The NEB UltraII FS kit was used to fragment cDNA. After end repair and A-tailing, a TruSeq adapter was ligated and 3′-end fragments were finally amplified using primers with Illumina P5 and P7 overhangs. In comparison to previous descriptions[65], the P5 and P7 sites were exchanged to enable sequencing of the cDNA in read1 and barcodes and UMIs in read2 to achieve a better cluster recognition. The library was sequenced on the NextSeq 500 (Illumina) system with 65 cycles for the cDNA in read1 and 19 cycles for the barcodes and UMIs in read2. Data were processed using the published Drop-seq pipeline (v.1.12) to generate sample- and gene-wise UMI tables[66]. Reference genome (GRCm38) was used for alignment. Transcript and gene definitions were used according to GENCODE version M25.

## Bulk RNA-seq data processing

Raw counts of two individual sequencing runs of the same library were merged, and non-overlapping genes were dropped. Differential expression analysis was performed using the EdgeR package (v.3.40.2)[67]. After excluding low-expressed genes (that is, genes for which an expression threshold of greater than 1 count per million was not attained in at least 4 samples), the negative binomial model was fitted. The resulting $P$ values were adjusted for multiple testing using the FDR correction. To increase the power, we limited our analysis to genes in the serumantibodyome[43]. Genes were considered to be differentially expressed if they had a less than 5% probability of being false positive ($P_{adj} < 0.05$). PCA was performed using the PCA function of the FactoMineR package on log-transformed counts per million (log[CPM]). Gene set enrichment analysis (GSEA) was performed on unfiltered DESeq2 normalized count data using the DESeq2 package (v.1.40.2)[68] and GSEA v.4.3.2 software[69,70] in conjunction with MSigDB (v.2023.1). The interrogated gene sets were derived from the M8 collection of cell type signature gene sets. Analysis was run with permutation-type phenotype and a FDR of 0.25.

## Generation of AQP4 TCR retrogenic mice

Retroviruses containing a high-affinity AQP4-specific TCR were produced by calcium phosphate precipitation of Platinum-E virus packaging cells with retroviral vectors (pMP71, a gift from D. Busch). To generate retrogenic mice, bone marrow was collected from the tibia and femur of 8–20-week-old $Cd45.1^{+/-} \times Rag1^{-/-}$ mice. After red blood cell lysis, single-cell suspensions were stained with anti-mouse Ly6A/E (Sca-1, 1:300) and anti-mouse CD3/CD19 (1:300) antibodies. FACS-sorted SCA1[+]CD3[−]CD19[−] stem cells were expanded with mouse IL-3 (2 ng ml[−1]), IL-6 (50 ng ml[−1]) and SCF (50 ng ml[−1]) for 3–4 days. Retroviral transduction of expanded stem cells was achieved by spinoculation[71]. In brief, 400 μl of Platinum-E supernatants were centrifuged in a retronectin-coated 48-well plate at 3,000$g$ for 2 h at 32 °C. Then, 200 μl of the medium was removed and filled up with expanded stem cells to a final concentration of 50,000 cells per 400 μl. After 2 days of culture, transduced stem cells were injected into irradiated recipient mice.

## Generation of mixed bone marrow chimeras

To assess the negative selection of AQP4-specific thymocytes in vivo, mixed bone marrow chimeras were generated by grafting congenically marked $Rag1^{-/-}$ bone marrow engineered to retrogenically express an AQP4-specific TCR (clone 6) along with bone marrow from either WT or $Aqp4^{\Delta B}$ mice (4:1) into lethally irradiated $Aqp4^{-/-}$ recipients so that maturing thymocytes from the polyclonal and the retrogenic (AQP4-specific) thymic compartments could be compared in the same host mouse. This setup also ensured that maturing thymocytes from both the polyclonal and retrogenic (AQP4-specific) thymic compartments encountered a thymic environment with either WT or AQP4-deficient B cells. AQP4 TCR retrogenic mice were generated as described earlier. Stem cells from the complementary donor were isolated, ex vivo stimulated in parallel and mixed with the retrogenic compartment before injection. Reconstitution was tested 5 to 6 weeks after transplantation.

## GC characterization

For characterization of GC reactions, mice were immunized with 200 μl emulsion containing 50 μg of either full-length AQP4 or recombinant human MOG protein. Sera were collected before (day −1) as well as on day 10 and day 21 after immunization. On day 21, the mice were euthanized for further analysis. Half of the draining lymph nodes and the spleen were isolated for flow cytometry analysis before mice were perfused with 4% PFA. The remaining half of the spleen and draining lymph nodes were isolated for histology. $T_{FH}$ cells were quantified using flow cytometry analysis of live CD3[+]CD4[+]PD-1[+]BCL6[+] cells. GC B cells were quantified histologically by immunoreactivity to BCL6 and by flow cytometry analysis of live CD3[−]CD19[+]B220[+]CD95[+]BCL6[+] cells.

## Anti-AQP4 and anti-MOG antibody detection assay

To detect anti-AQP4 or anti-MOG antibodies in the sera of protein-immunized mice, a cell-based flow cytometry assay was used[72,73].

Sera were diluted (1:50) in RPMI 1640 growth medium and added to a 96-well plate containing 30,000 LN18^AQP4 or LN18^MOG cells per well. As a control, every serum was tested on LN18^CTRL cells (transduced with an empty vector). The plate was incubated on ice on an orbital shaker for 25 min. Cells were washed twice with FACS buffer. To stain mice for LN18-cell bound mouse IgG, 50 µl of diluted (1:100 in washing buffer) Alexa-Fluor-488-labelled goat anti-mouse IgG H+L (Life Technologies, Thermo Fisher Scientific) was added to each well. After incubation for 25 min on ice, cells were washed twice with FACS buffer.

## Statistical analysis

Statistical evaluations of cell frequency measurements and cell numbers were performed using one-way-ANOVA and post hoc tests when more than two populations were compared. Two-way ANOVA followed by post hoc multiple comparison tests was used as indicated in the figure legends. Correlation and linear regression were calculated for fractions of thymic CD4 SP and thymic B cells. EAE incidence was calculated using Kaplan–Meier analysis and the $P$ values were analysed with a log-rank test (Mantel–Cox). Age at immunization, day of disease onset, peak EAE scores and cumulative EAE scores were compared using Mann–Whitney $U$-tests. $P < 0.05$ was considered to be significant. Calculations and the generation of graphs were performed using Graph Pad Prism v.10.9.0 (GraphPad), RStudio (v.2023.06.01) and Python (v.3.9.16). Figures were prepared using Adobe Illustrator 2022 (v.26.0.1).

## Materials and reagents

An extensive list of materials and reagents is provided in Supplementary Table 2.

## Mouse histology heat maps

Heat maps representing CD19 density in mouse thymi were created using Visiopharm (v.2023.1). First, tissue regions with high expression of EPCAM were identified and outlined based on the marker intensity. CD19+ cells were identified based on DAPI and CD19 expression. For heat map creation, CD19+ cells overlapping within a radius of 50 µm were considered. Figures were prepared using Adobe Photoshop CS6 (v.13.0).

## Analysis of published thymic scRNA-seq data for mouse and human

Annotated matrix files (h5ad) with precalculated UMAPs and cell type classifications of the original authors[16] were downloaded from Zenodo (https://doi.org/10.5281/zenodo.5500511). All cells with any *Aqp4* or *AQP4* expression greater than zero were included in the analysis of positive cells. An MHC-II score was generated in SCANPY to summarize mouse H2-Aa, H2-Ab1, H2-Eb1, H2-Eb2 and human HLA-DPA1, HLA-DPB1, HLA-DQA1, HLA-DQA2, HLA-DQB1, HLA-DQB2, HLA-DRA, HLA-DRB1 and HLA-DRB5 expression. Pre-annotated classifications as stored in the dataset's .obs data frame 'cell types' and 'Anno_level_fig1' were used for mouse and human cell type labelling, respectively.

## Transduction of A5 cells with AQP4-specific TCRs

Retroviruses containing AQP4-specific TCRs were produced by calcium phosphate precipitation of Platinum-E virus packaging cells with retroviral vectors (pMP71) as described previously[71]. Virus-containing supernatants were collected after 2 days, centrifuged to dispose of cell debris, and either used immediately for spinoculation or kept at 4 °C for a maximum of 4 weeks. For spinoculation, 400 µl of Platinum-E supernatants were centrifuged in a 48-well plate at 3,000*g* for 2 h at 32 °C. Then, 200 µl medium was removed and filled up with A5 cells to a final concentration of 50,000 cells per 400 µl. After 2 days of culture, the expression of TCR alpha and beta variable chains in transfected A5 cells was tested by flow cytometry analysis. A5 cells expressing both chains of the respective TCR were then enriched by FACS sorting, stimulated with P41-pulsed APCs and tested for their NFAT–GFP expression after stimulation using flow cytometry.

## T_reg cell depletion

DEREG mice provide an efficient in vivo model for inducible T_reg cell depletion[48]. This model was investigated in EAE previously and required some modifications of the immunization procedure[74]. In contrast to the earlier described immunization protocol, mice received a reduced amount of pertussis toxin (50 ng intravenously on days 0 and 2 after immunization) and 0.5 µg of diphtheria toxin intraperitoneally (i.p.) two days before and on days 5 and 6 after immunization.

## Transfer of mature T cells into *Tcra*^−/− mice

Mature CD4+ T cells were isolated from unmanipulated *Aqp4*^ΔB mice by CD4 MACS bead enrichment (Miltenyi Biotec). In total, $5 × 10^6$ T cells were injected i.v. into *Tcra*^−/− mice. Then, 1 day after transfer, *Tcra*^−/− recipient mice were immunized as indicated and analysed for AQP4-specific T cell frequencies in the transferred T cell repertoire by P41–I-A^b tetramer staining 2 days after EAE onset.

## Adoptive transfer EAE

A total of $5 × 10^6$ FACS-sorted CD4+ T cells was transferred intravenously into *Tcra*^−/− mice (day 0), followed by subcutaneous immunization at both flanks with an emulsion containing PBS and CFA (day 1). Mice were scored and weighed daily before they were euthanized on day 32 after immunization. Serum was preserved after cardiac blood collection and tested for autoantibodies.

## Autoantibody assay

To screen for autoantibodies, the preserved mouse sera were tested on histological cryosections from lymphocyte-deficient *Rag1*^−/− mice using secondary anti-mouse IgG (H+L) AF488 antibodies (Thermo Fisher Scientific). Mice were euthanized by sequential intracardial perfusion with ice-cold PBS followed by 4% PFA. Dissected organs were then embedded in Tissue-Tek O.C.T. compound and immediately frozen with liquid nitrogen. After fixation with precooled acetone (−20 °C) for 1 min and subsequent blocking with normal goat serum (30% dilution in PBS) for 10 min, 10 µm cryosections were incubated with preserved mouse sera (1:50) for 1 h. The samples were next treated with secondary anti-mouse IgG (H+L) AF488 antibodies (1:500) for 30 min. Finally, the sections were washed and mounted with ProLong Gold Antifade Mounting reagent containing DAPI (Thermo Fisher Scientific, P36931). The staining was performed using the Leica Bond RXm device, all washing steps were performed with Bond Wash solution (Leica, AR9590) and all dilutions were prepared using Bond Primary Antibody Diluent (Leica, AR9352).

## Quantification of AQP4 loss

AQP4 loss was determined in a semi-quantitative approach counting the extent of adjacent CNS lesions. The extent of CNS lesions was measured automatically using QuPath's positive nuclear detection of CD45+ cells and their engaging area. The AQP4 signal was calculated using QuPath's positive pixel count algorithm in the adjacent area with a defined radius of 100 µm to CD45+ infiltrates (region of interest). For reasons of comparison, the AQP4 signal was normalized to the extent of CNS lesions. All annotations were performed in a blinded manner.

## Licenses

Parts of Figs. 2 and 4, as well as Extended Data Figs. 3, 6, 7 and 9 were drawn using pictures from Servier Medical Art. Servier Medical Art by Servier is licensed under a Creative Commons licence CC BY 3.0. Parts from Extended Data Fig. 9a were created using BioRender.

## Reporting summary

Further information on research design is available in the Nature Portfolio Reporting Summary linked to this article.

## Data availability

scRNA-seq and bulk RNA-seq data generated for this study have been deposited at the Gene Expression Omnibus under accession numbers GSE234188 and GSE244363, respectively. Source data are provided with this paper.

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

**Acknowledgements** We thank the members of the core facility for Comparative Experimental Pathology (Institute of Pathology, Technical University of Munich) for their technical support in processing the histology specimens; and M. Moldt for her support with the purification of AQP4 protein. A.M.A. received intramural funding from the Technical University of Munich (KKF E-09) and was supported by the Deutsche Forschungsgemeinschaft (DFG, German Research Foundation) under Germany's Excellence Strategy within the framework of the Munich Cluster for Systems Neurology (EXC 2145 SyNergy, 390857198); G.S. by a YLSY doctoral scholarship from the Republic of Türkiye Ministry of National Education; J.P.B. by the Deutsche Forschungsgemeinschaft (ID 424926990); D.M. by the Swiss National Science Foundation (310030_215050 and 310030B_201271) and the ERC (865026); D.H.B. by the Deutsche Forschungsgemeinschaft (TRR338 (452881907)); and T.K. by the Deutsche Forschungsgemeinschaft (SFB1054 (210592381), TRR128 (213904703), TRR274 (408885537), TRR355 (490846870), GRK2668 (435874434) and EXC 2145 (SyNergy, 390857198)) and the Hertie Network of Clinical Neuroscience.

**Author contributions** A.M.A., L.N., C.S., M.P., O.U., T.H., C.F., E.P., S.R.K., H.H.C., S.T., A. Muschawechk, C.D., A.B., K.S. and G.S. designed and performed major experiments, analysed data and drafted the manuscript. K.L. designed and performed the expression of mouse AQP4 full-length protein. R.Ö., R.R. and I.B.P. sequenced and analysed data. S.J., A.S., A. Mühlbauer and S.G. performed TCR expression. J.P.B. performed confocal imaging. I.W., M.K. and D.M. performed multiplex immune fluorescence experiments. O.P.O., B.H., M.S.S., V.R.B., S.H. and D.H.B. designed and conceptualized experiments. L.K. and T.K. analysed data, conceptualized and supervised the study, and wrote the manuscript.

**Funding** Open access funding provided by Technische Universität München.

**Competing interests** The authors declare no competing interests.

**Additional information**
**Correspondence and requests for materials** should be addressed to Thomas Korn.

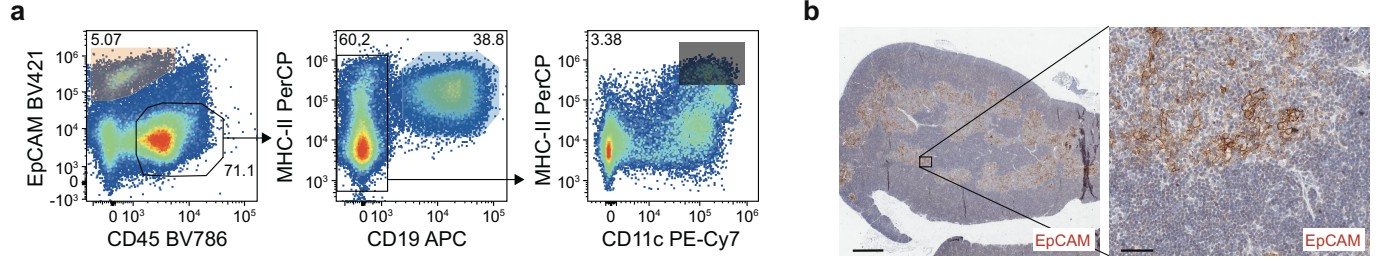

**Extended Data Fig. 1 | B cells in the thymus.** (**a**) Gating strategy of Thy1-depleted wild-type thymus for FACS-sorting of thymic epithelial cells (TECs, live CD45⁻ EpCAM⁺), thymic B cells (live CD45⁺EpCAM⁻CD19⁺), and thymic dendritic cells (live CD45⁺EpCAM⁻CD19⁻CD11c⁺MHC-II$^{high}$). (**b**) Representative EpCAM immunostaining in wild-type thymus samples (n = 3 independent experiments, scale bar left 500 μm and right 50 μm). (**c**) CD19⁺ B cells (red) in relation to EpCAM-expressing cells (white) in wild-type (leftmost columns) and *Aqp4⁻/⁻* thymus (right column). Middle row: EpCAM-expressing areas (red outlines) and CD19⁺ B cells (white outlines) are detected in DAPI-stained (blue) wild-type (left columns) and *Aqp4⁻/⁻* thymi (right column). Bottom row: The density of B cells is depicted as a 5-colour heatmap within a radius of 50 μm around CD19⁺ cells and overlayed on the area of EpCAM-expressing cells. n = 2 independent experiments.

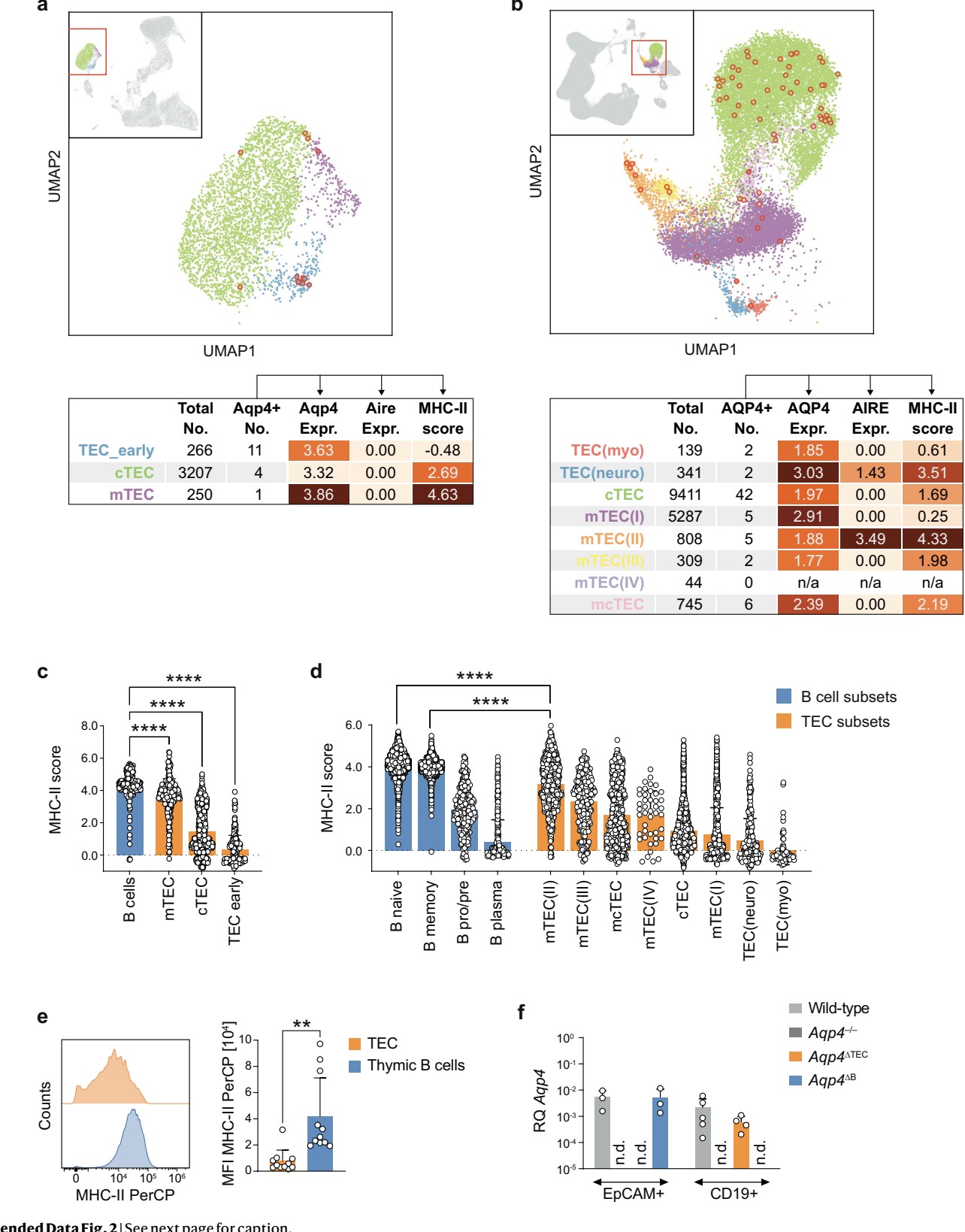

**Extended Data Fig. 2** | See next page for caption.

**Extended Data Fig. 2 | AQP4 and MHC-II expression in thymic B cells and thymic epithelial cells.** (**a**-**d**) Analysis of previously published mouse (**a**, **c**) and human (**b**, **d**) thymic scRNAseq datasets. (**a**, **b**) Upper panel: UMAP representation of TEC subsets as selected from the total dataset shown in the upper-left quadrant. The red square indicates the magnified region, and the red circles indicate *Aqp4*- and *AQP4*-expressing cells for mouse (**a**) and human data (**b**), respectively. Colour code of cells in UMAP as annotated in the table row titles below. Lower panel: Table showing the total population size in the dataset along with the number of *Aqp4*- and *AQP4*-expressing cells and (only for these positive cells) their mean expression level of *Aqp4* and *AQP4*, as well as *Aire* and *AIRE*, and mean MHC-II score for mouse (**a**) and human subsets (**b**), respectively. (**c**, **d**) Comparison of MHC-II scores from B cell and TEC subsets irrespective of their *Aqp4* or *AQP4* status in mouse (**c**) and human (**d**). Each circle represents a cell in the dataset. Data shown as mean ± SD. \*\*\*\* P < 0.0001 following one-way ANOVA, specifically Dunnett's multiple comparison test to compare B cells (n = 207) with all TEC subsets (N as indicated in the corresponding table above) in mouse (**c**) and Sidak's multiple comparison test to compare all B cell subsets (n = 2161 for memory, n = 2152 for naive, n = 479 for plasma and n = 290 for pro/pre B cells) with all TEC subsets (n as indicated in the corresponding table above) in human (**d**). To maintain legibility, only two representative tests were indicated in (**d**), with naive and memory B cells also outranking all other TEC subsets at the same significance level. (**e**) Representative histogram overlay of MHC-II expression in TECs and thymic B cells and corresponding quantification. Data are shown as mean fluorescence intensity (MFI) ± SD, two-tailed unpaired t-test, \*\* P < 0.01. Symbols represent biological replicates. (**f**) *Aqp4* expression in TECs and thymic B cells FACS-sorted from wild-type, *Aqp4*[−/−], *Aqp4*[ΔTEC], and *Aqp4*[ΔB] mice. Total RNA was isolated and probed for *Aqp4*. *Aqp4* expression was normalized to astrocytes. Mean relative RQ ± SD. Symbols indicate biological replicates; zero-values are not depicted due to logarithmic scaling. n.d., not detected.

**a**

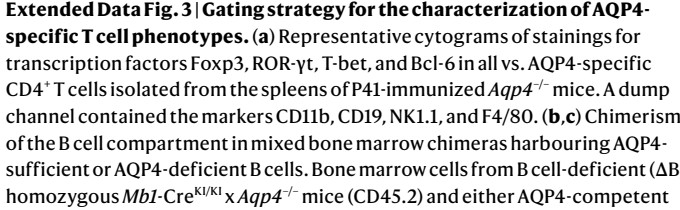

(The panel **a** consists of flow cytometry cytograms)

Top row:
- Dump BV510 vs CD4 PerCP eF710
- CD4 PerCP-eF710 vs Foxp3 FITC: 10.5
- CD4 PerCP-eF710 vs T-bet PE-Cy7: 8.01
- CD4 PerCP-eF710 vs ROR-γt BV786: 5.14
- CD4 PerCP-eF710 vs Bcl-6 BV421: 1.12

Bottom row:
- P41/I-A$^b$ APC vs P41/I-A$^b$ PE
- CD4 PerCP-eF710 vs Foxp3 FITC: 7.2
- CD4 PerCP-eF710 vs T-bet PE-Cy7: 53.2
- CD4 PerCP-eF710 vs ROR-γt BV786: 6.32
- CD4 PerCP-eF710 vs Bcl-6 BV421: 13.7

**b**

$Aqp4^{-/-}$ x $Mb1$-Cre$^{KI/KI}$ ($\Delta$B)
CD45.2

$Aqp4^{+/+}$ (WT) vs. $Aqp4^{-/-}$
CD45.1

9:1 →

$Aqp4^{-/-}$
CD45.1/CD45.2

**c**

SPL

B220 PE-Cy7 vs CD19 PerCP: 29.5
SSC-A [10^5] vs CD45.1 PB450: 88.7 / 11.3
SSC-A [10^5] vs CD45.1 PB450: 3.43 / 96.6

THY

B220 PE-Cy7 vs CD19 PerCP: 0.17
SSC-A [10^5] vs CD45.1 PB450: 99.9 / 0.08
SSC-A [10^5] vs CD45.1 PB450: 11.0 / 87.7

**Extended Data Fig. 3 | Gating strategy for the characterization of AQP4-specific T cell phenotypes.** (**a**) Representative cytograms of stainings for transcription factors Foxp3, ROR-γt, T-bet, and Bcl-6 in all vs. AQP4-specific CD4$^+$ T cells isolated from the spleens of P41-immunized $Aqp4^{-/-}$ mice. A dump channel contained the markers CD11b, CD19, NK1.1, and F4/80. (**b,c**) Chimerism of the B cell compartment in mixed bone marrow chimeras harbouring AQP4-sufficient or AQP4-deficient B cells. Bone marrow cells from B cell-deficient ($\Delta$B) homozygous $Mb1$-Cre$^{KI/KI}$ x $Aqp4^{-/-}$ mice (CD45.2) and either AQP4-competent wild-type or AQP4-deficient $Aqp4^{-/-}$ mice (CD45.1) (9:1) were grafted into $Aqp4^{-/-}$ mice (CD45.1$^{+/-}$) to generate mixed bone marrow chimeras (MBMC), in which only B cells were competent in AQP4 expression in an otherwise AQP4-deficient environment. After six weeks, the lymphoid compartments (SPL, spleen; THY, thymus) were fully reconstituted, with the B cell compartment largely derived from the CD45.1$^+$ AQP4-competent or deficient donors, respectively. The diagram in (**b**) was created using Servier Medical Art under a Creative Commons licence CC BY 3.0.

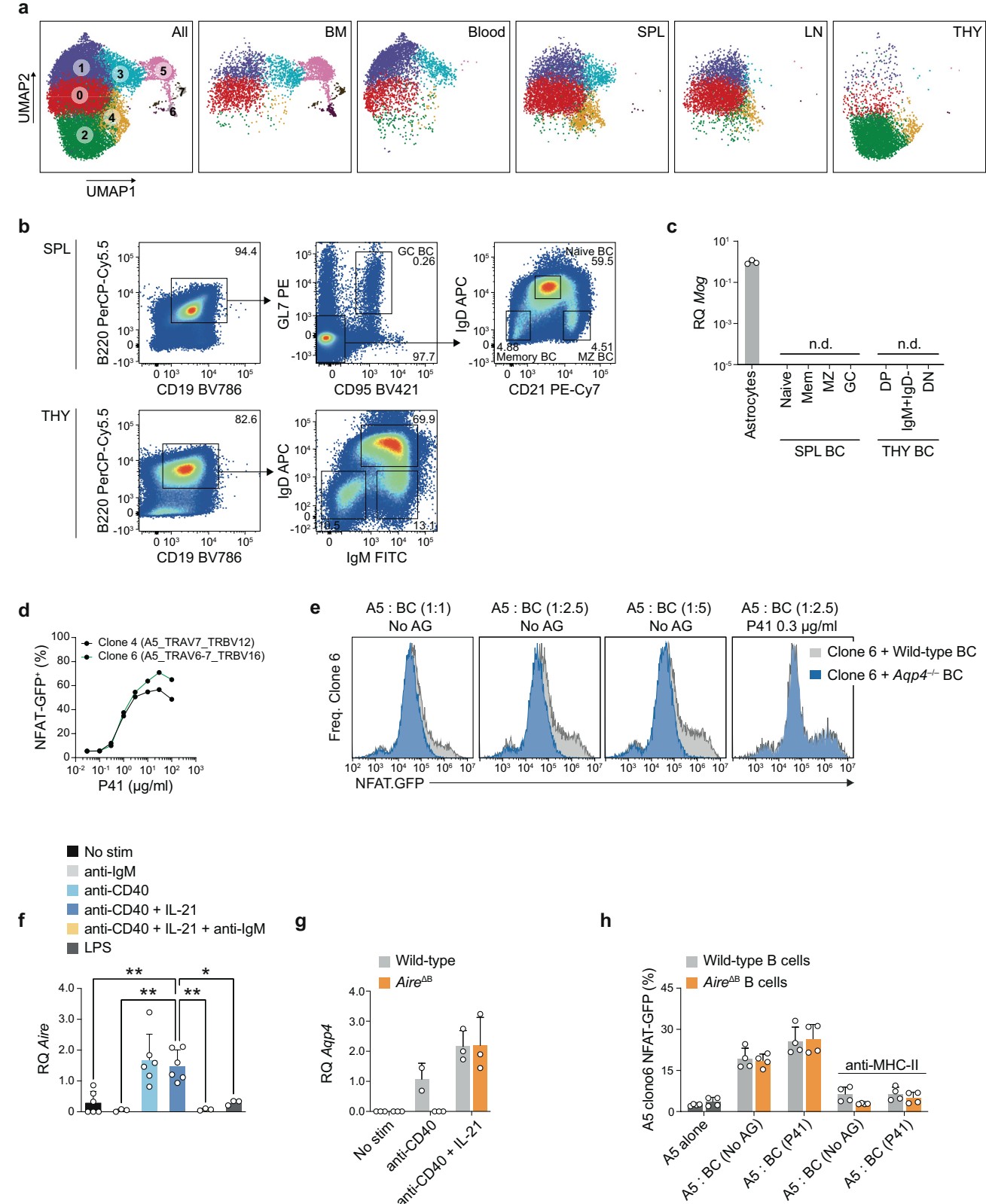

**Extended Data Fig. 4** | See next page for caption.

**Extended Data Fig. 4 | Thymic B cells are distinct from naïve peripheral B cells and express and present AQP4.** (**a**) UMAP representation of single-cell RNA sequencing data of B cells sorted from bone marrow (BM), blood, spleen (SPL), lymph nodes (LN), and thymus (THY) of naïve wild-type mice. Left panel: Clusters as determined by Leiden with resolution r = 0.7, shown for cells of all organs. Other panels: Only cells from the respective organs are depicted as indicated. (**b**) Splenic and thymic B cell subsets were sorted from naïve wild-type mice according to the indicated gating strategy, and total RNA was isolated for quantitative PCR. (**c**) Relative gene expression (RQ) of myelin oligodendrocyte glycoprotein (MOG) in thymic and splenic B cell subsets (n = 2 biological replicates), normalized to astrocytes (n = 3). Data are shown as mean RQ ± SD. Symbols indicate biological replicates; zero-values are not depicted due to logarithmic scaling. n.d., not detected. (**d**) NFAT-GFP response as a measure of TCR triggering to titrated P41-concentrations in a coculture system of antigen-presenting cells (APC) and the T cell hybridoma cell line A5 transfected with either AQP4 TCR clone 4 or clone 6 (both n = 2). (**e**) Representative histogram overlays of NFAT-GFP responses in a coculture system of AQP4-specific TCR clone 6-transfected A5 cells and either wild-type or AQP4-deficient B cells in the absence or presence of exogenous P41. (**f**) *Aire* in wild-type and (**g**) *Aqp4* in *Aire*$^{\Delta B}$ splenic CD19$^+$ B cells stimulated for two days under conditions as indicated. RQ was normalized to control stimulation with goat anti-human IgG (H + L). (**h**) Quantification of NFAT-GFP expression in a coculture system with a T cell hybridoma cell line (A5 cells) engineered to express an AQP4-specific TCR and B cells prestimulated with anti-CD40 plus IL-21 for two days, derived from wild-type or *Aire*$^{\Delta B}$ mice (n = 4 biological replicates). All data in (**f-h**) are shown as mean ± SD and tested with (**f**) one-way ANOVA and Tukey's post-test or (**g, h**) two-way ANOVA and Sidak's post-test. * P < 0.05, ** P < 0.01. Unless otherwise specified, symbols represent biological replicates.

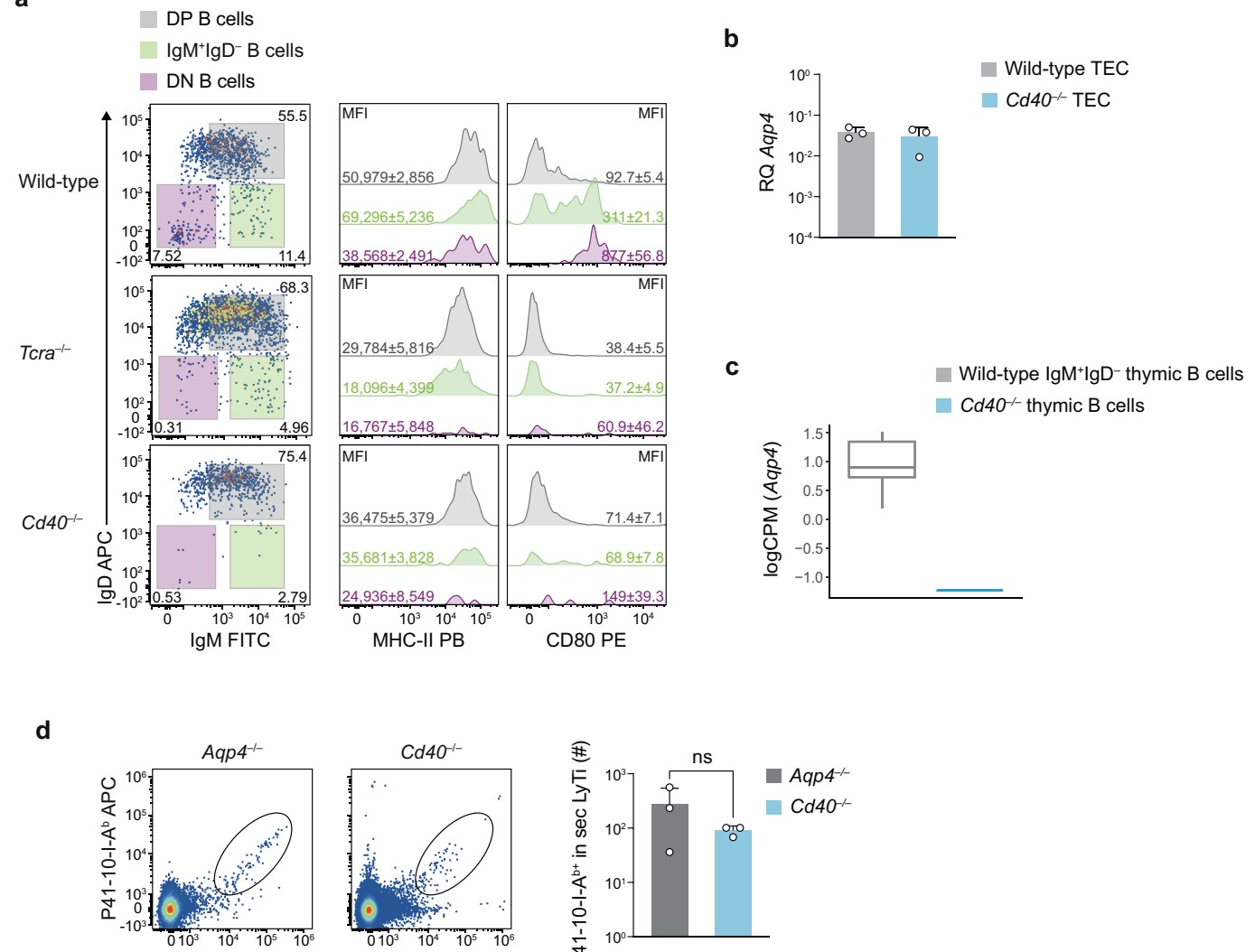

**Extended Data Fig. 5 | CD40 is essential in licensing APC properties in thymic B cells.** (**a**) Thymic B cells from wild-type, *Tcra*[−/−], and *Cd40*[−/−] mice (all n = 5 biological replicates) were characterized for their expression of surface markers IgD, IgM, MHC class II, and CD80. Representative cytograms and histograms of thymic B cell subsets (DP = double positive IgM⁺IgD⁺, IgM⁺IgD⁻, and DN = double negative IgM⁻IgD⁻) are shown along with mean fluorescence intensities (MFI) ± SD of MHC class II and CD80 next to the corresponding histograms. (**b**) *Aqp4* expression in EpCAM⁺ thymic epithelial cells (TECs) isolated from wild-type and *Cd40*[−/−] mice (n = 3 biological replicates). Mean

RQ ± SD normalized to astrocytes. (**c**) Expression of *Aqp4* in wild-type IgM⁺IgD⁻ thymic B cells vs. *Cd40*[−/−] thymic B cells. Box plot derived from the RNAseq data in Fig. 3i, with the median as the centre, the first and third quartiles as the boundaries of the box, and 1.5 times the IQR as the whiskers. (**d**) Representative cytograms and quantification of P41/I-A^b-reactive T cells isolated from secondary lymphoid tissue (sec LyTi, spleen plus draining lymph nodes) of P41-immunized *Aqp4*[−/−] and *Cd40*[−/−] mice (both n = 3 biological replicates) on day 10 after immunization. Data are shown as mean ± SD tested with a two-tailed unpaired t-test, ns = not significant.

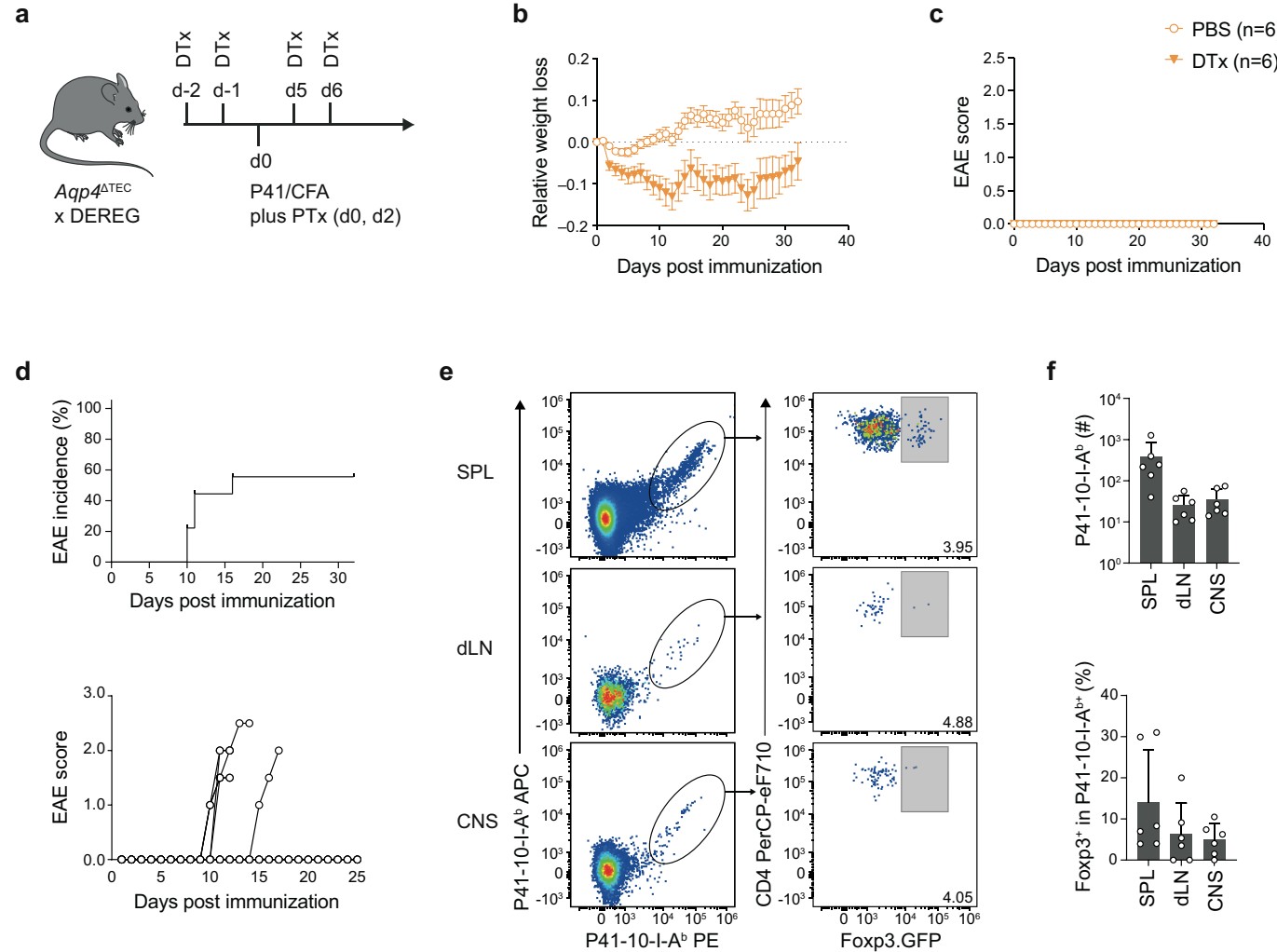

**Extended Data Fig. 6 | Mature AQP4-specific T cells are not eliminated but expanded in the systemic immune compartment of AQP4-sufficient host mice.** (**a**-**c**) Foxp3$^+$ Treg cells were depleted in $Aqp4^{\Delta TEC}$ x DEREG mice through sequential intraperitoneal (i.p.) injection of diphtheria toxin (DTx) or PBS as a negative control as indicated (both n = 6 biological replicates). Mice were concomitantly immunized with P41 in CFA plus PTx as indicated (**a**). Immunized mice were weighed (**b**) and scored (**c**) daily for 32 days. (**d**-**f**) The mature T cell repertoire of $Aqp4^{\Delta B}$ mice was transferred into $Tcra^{-/-}$ mice, followed by immunization with full-length AQP4 (n = 10 mice). (**d**) Incidence and individual disease courses of immunized recipient mice. (**e**) Representative P41/I-A$^b$ tetramer and Foxp3 stainings in CD4$^+$ T cells isolated from secondary lymphoid tissues (SPL, spleen; dLN, draining lymph node) and the CNS of AQP4-immunized recipient mice with an EAE phenotype. (**f**) Quantification of the absolute number of P41/I-A$^b$ reactive T cells and the fraction of Foxp3$^+$ cells among antigen-specific T cells shown as mean ± SD. Symbols indicate biological replicates; zero-values are not depicted due to logarithmic scaling (n = 6 biological replicates). The diagram in (**a**) was created using Servier Medical Art under a Creative Commons licence CC BY 3.0.

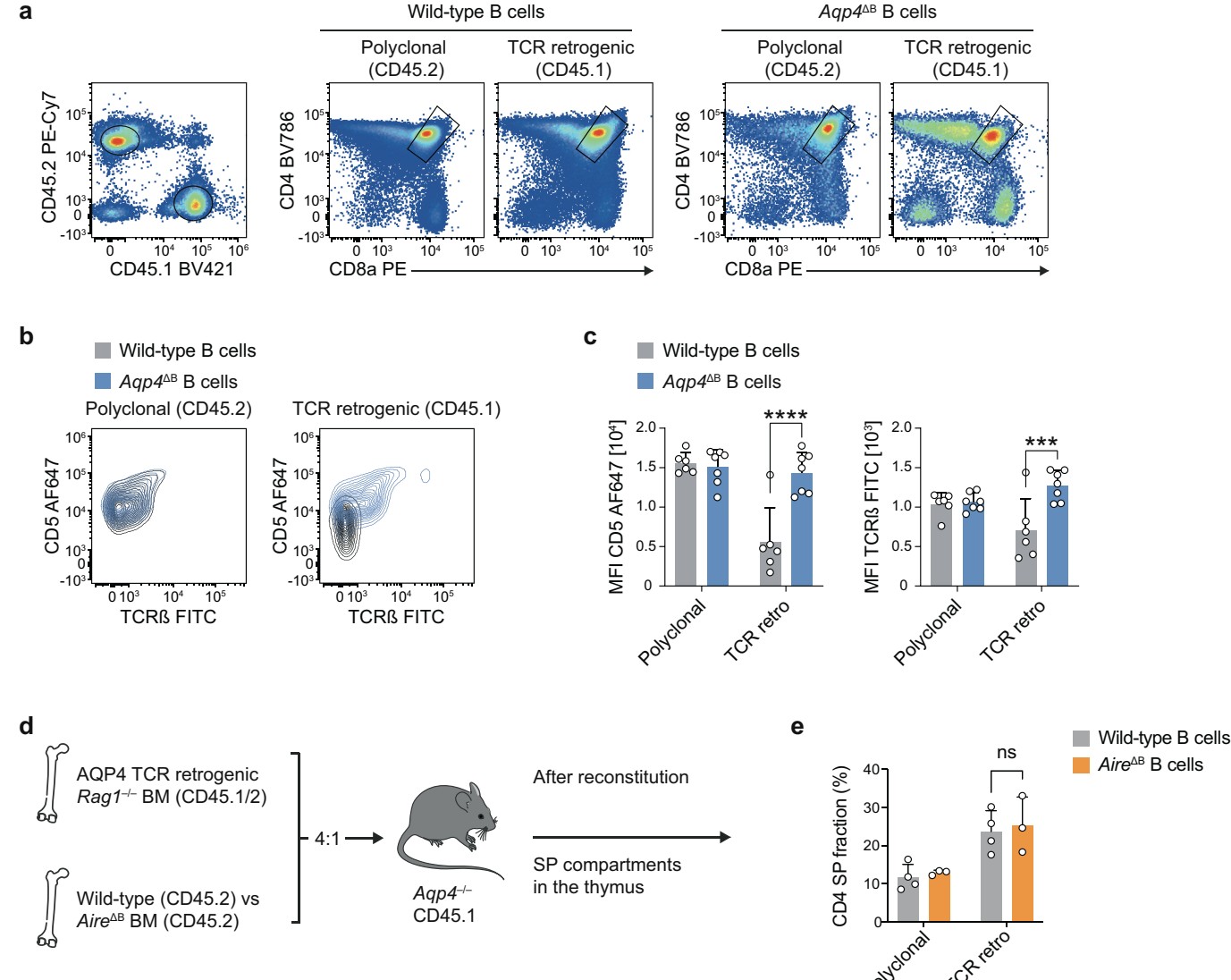

**Extended Data Fig. 7 | The negative selection of AQP4-specific thymocytes is independent of Aire.** (**a-c**) Mixed bone marrow chimeras (MBMC) were generated by grafting congenically marked *Rag1*[−/−] bone marrow engineered to retrogenically express AQP4-specific TCR clone 6 mixed with bone marrow from either wild-type or *Aqp4*[ΔB] mice (4:1) into lethally irradiated *Aqp4*[−/−] recipients and tested 6 weeks after engraftment. (**a**) Representative cytograms of the polyclonal and the retrogenic (AQP4-specific) thymic compartment facing a thymic environment equipped with either wild-type (n = 6 biological replicates) or AQP4 deficient B cells (n = 7 biological replicates). (**b**, **c**) CD5 and TCR-β expression in CD4⁺CD8⁺ double positive (DP)

thymocytes. (**b**) Representative cytogram overlays and (**c**) quantification of DP thymocytes from the polyclonal and the retrogenic (AQP4-specific) thymic compartments. Symbols represent biological replicates. (**d**) MBMCs were generated as described in (**a-c**) with wild-type (n = 4 biological replicates) and *Aire*[ΔB] (n = 3 biological replicates) donor mice (4:1). (**e**) Quantification of the thymic CD4⁺ single positive (SP) fraction. Data in (**c**, **e**) are shown as mean ± SD and tested with two-way ANOVA and Sidak's post-test. *** P < 0.001, **** P < 0.0001. ns = not significant. The diagram in (**d**) was created using Servier Medical Art under a Creative Commons licence CC BY 3.0.

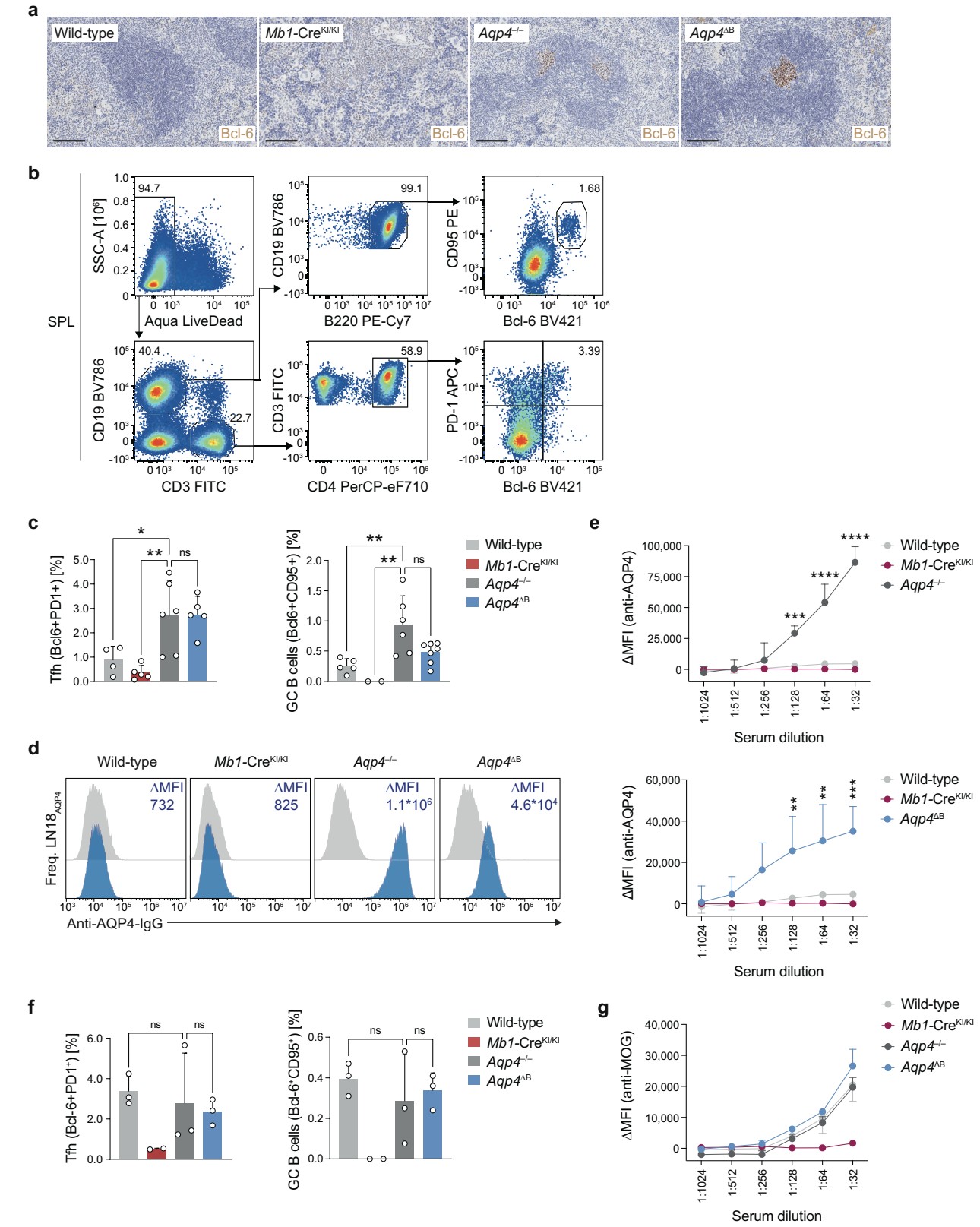

**Extended Data Fig. 8** | See next page for caption.

**Extended Data Fig. 8 | Characterization of germinal centre responses to full-length AQP4 immunization.** Wild-type, B cell-deficient (*Mb1*-Cre^KI/KI^), *Aqp4*^−/−^, and *Aqp4*^ΔB^ mice were immunized with full-length AQP4 or human recombinant full-length MOG protein and tested for germinal centre (GC) responses on d12 after immunization. (**a**) Representative immunostainings of Bcl-6 in spleens of full-length AQP4-immunized mice (n = 2 independent experiments, scale bar 100 μm). (**b**) Representative cytograms and gating strategy of splenic Tfh cells (live CD19⁻CD3⁺CD4⁺Bcl-6⁺PD-1⁺) and GC B cells (live CD19⁺CD3⁻B220⁺Bcl-6⁺CD95⁺). (**c**) Flow cytometric quantification of splenic Tfh cell and GC B cell frequencies in spleens of AQP4-immunized wild-type, B cell-deficient (*Mb1*-Cre^KI/KI^), *Aqp4*^−/−^, and *Aqp4*^ΔB^ mice. (**d**, **e**) The AQP4-specific serum response was tested with serial dilutions in a cell-based assay with sera isolated prior to (d-1) and on d21 after immunization with full-length AQP4. (**d**) Representative histograms of the anti-AQP4-serum response tested in LN18^AQP4^ cells at d-1 (grey histograms) and d21 (blue histograms). (**e**) Quantification of the anti-AQP4-serum response for *Aqp4*^−/−^ (n = 2, upper panel) and *Aqp4*^ΔB^ (n = 3, lower panel) alongside wild-type (n = 4) and *Mb1*-Cre^KI/KI^ (n = 2, depicted in both panels for reference), tested in serial dilutions as indicated on the x-axis. (**f**) Quantification of Tfh cell and GC B cell frequencies in secondary lymphoid tissues of MOG protein-immunized wild-type (n = 3), B cell-deficient *Mb1*-Cre^KI/KI^ (n = 2), *Aqp4*^−/−^ (n = 3), and *Aqp4*^ΔB^ (n = 3) mice. (**g**) The MOG-specific serum response was tested with serial dilutions in a cell-based assay with sera isolated on d-1 and d21 after immunization with full-length MOG protein. Quantification of the anti-MOG-serum response tested in serial dilutions as indicated on the x-axis. Data in (**c**, **f**) are shown as mean ± SD and tested by one-way ANOVA with Tukey's post-test. Data in (**e**, **g**) are shown as delta mean fluorescence intensity (ΔMFI = MFI$_{d21}$ − MFI$_{d-1}$) ± SD and tested by two-way ANOVA with Sidak's post-test. * P < 0.05, ** P < 0.01, *** P < 0.001, **** P < 0.0001. ns = not significant. Symbols indicate biological replicates.

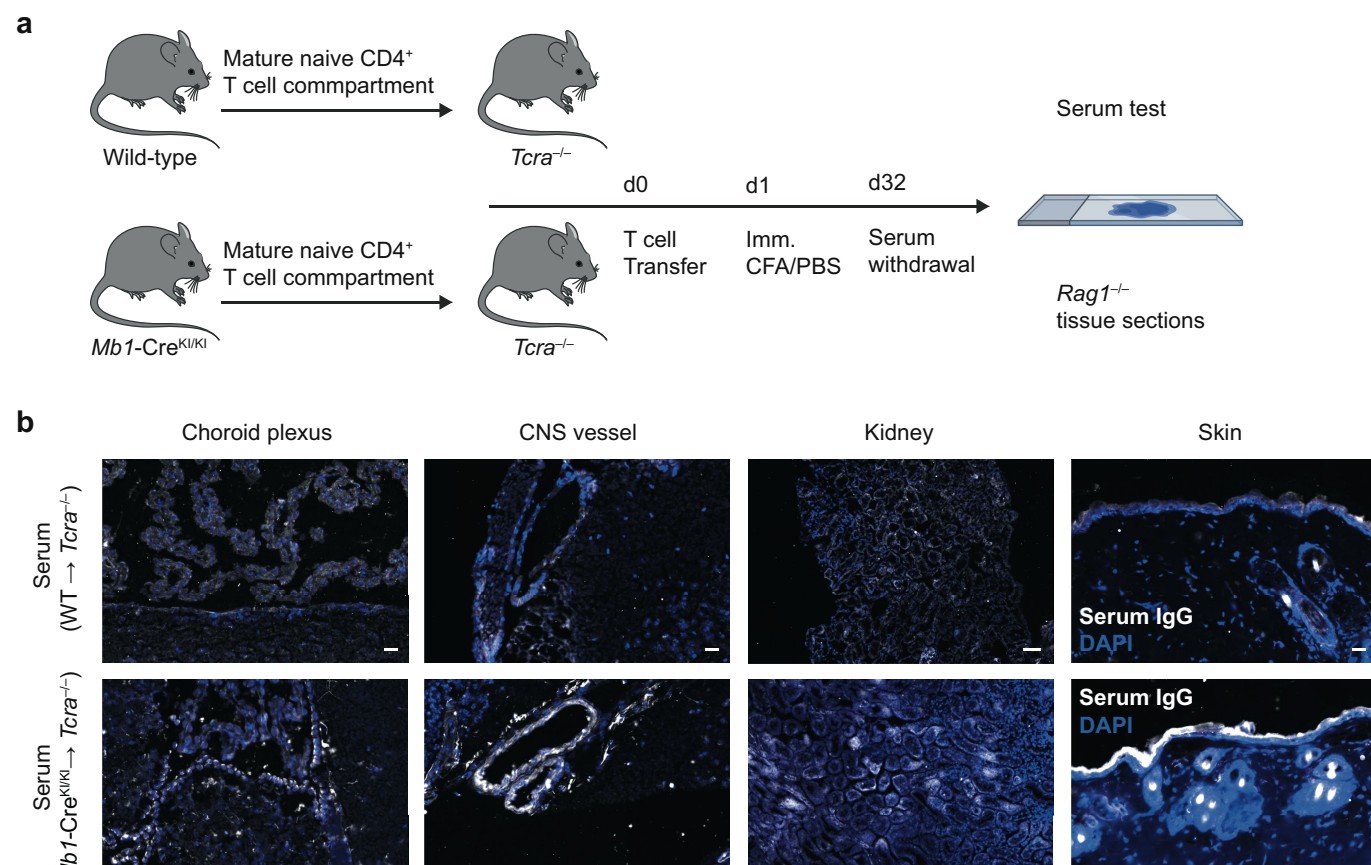

**Extended Data Fig. 9 | The T cell compartment educated in B cell-deficient mice facilitates the generation of autoantibodies. (a)** Mature CD4[+] cells from either wild-type or B cell-deficient *Mb1*-Cre[KI/KI] mice were transferred intravenously (i.v.) into *Tcra*[−/−] recipient mice (d0) followed by subcutaneous immunization with PBS and CFA on the day after transfer (d1). Sera were collected on d32 after immunization and tested for autoantibodies (IgG) on cryosections of different organs dissected from *Rag1*[−/−] mice using a secondary anti-mouse IgG (H + L) antibody. **(b)** Representative immunofluorescence stainings from various anatomical niches (choroid plexus, CNS vessel, kidney, skin) of n = 3 independent wild-type-educated sera and n = 4 independent *Mb1*-Cre[KI/KI]-educated sera. Scale bar = 20 μm (except kidney: 50 μm). The diagram in (**a**) was created using Servier Medical Art and BioRender under a Creative Commons licence CC BY 3.0.

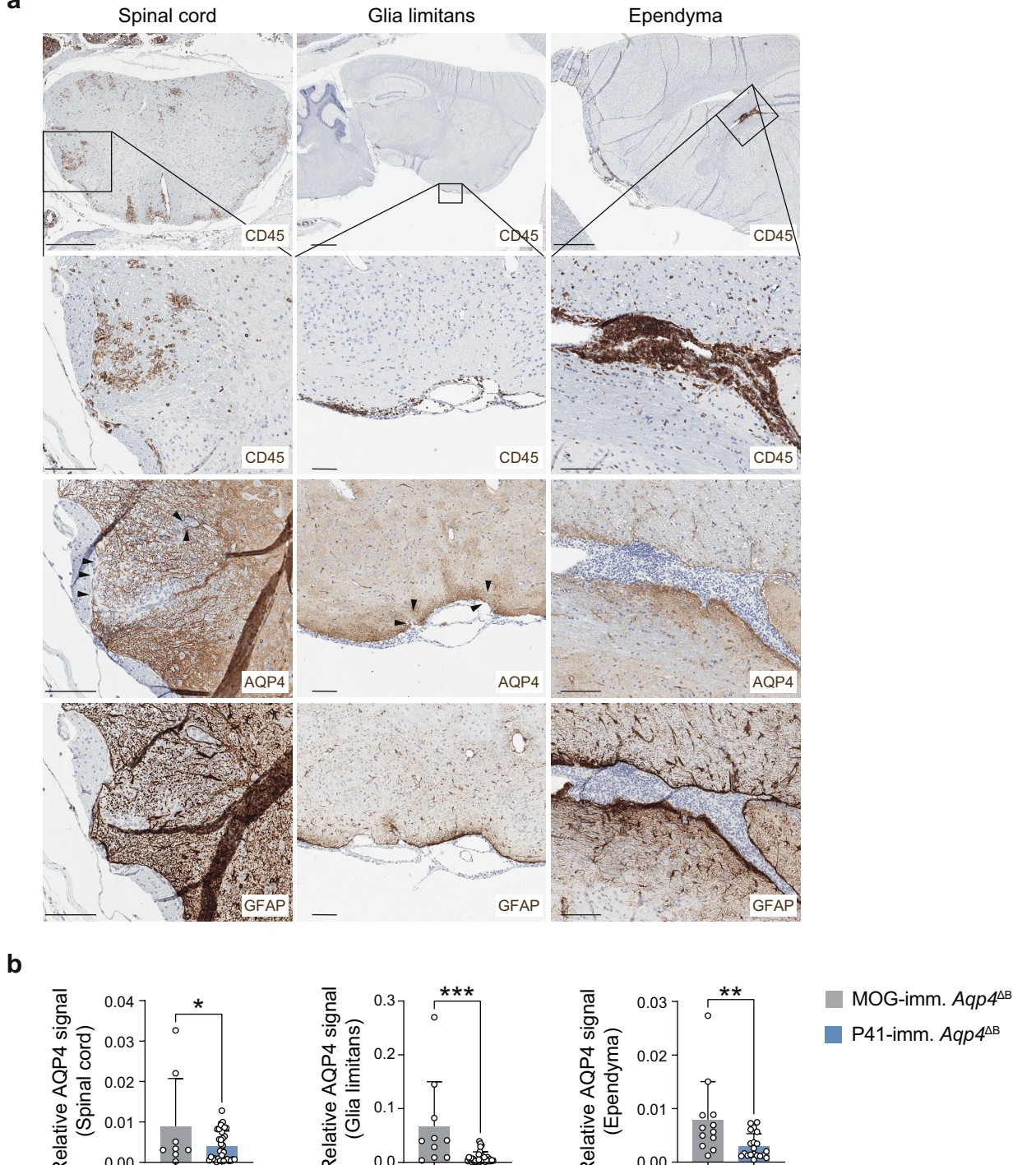

**Extended Data Fig. 10 | AQP4-directed autoimmunity in mice recapitulates histological hallmarks of NMOSD.** (**a**) Representative CD45, AQP4, and GFAP stainings of the spinal cord, cerebellum, and ependyma of P41-immunized *Aqp4*$^{\Delta B}$ mice at the peak of EAE (scale bars top row 1 mm except for spinal cord 200 μm, all other rows 100 μm). (**b**) AQP4 loss was determined in a semi-quantitative approach normalized to the extent of adjacent CNS lesions defined by CD45⁺ immunoreactivity. The AQP4 signal was calculated by QuPath's positive pixel count algorithm in the adjacent area with a defined radius of 100 μm with respect to CD45⁺ infiltrates (region of interest). Data are shown as mean ± SD and tested with a two-tailed unpaired t-test. *P < 0.05, **P < 0.01, ***P < 0.001. Symbols represent individual CNS lesions detected in two biological replicates for each group.

**Extended Data Table 1 | Disease characteristics of MOG(35–55)- and AQP4(201-220, P41)-induced EAE in wild-type and *Aqp4*<sup>ΔB</sup> mice**

|  | Wild-type (MOG) | Wild-type (P41) | *Aqp4*<sup>ΔB</sup> (MOG) | *Aqp4*<sup>ΔB</sup> (P41) |
|---|---|---|---|---|
| **Number of mice, n** | 9 | 25 | 14 | 17 |
| **Mean age at imm. (± SD)** | 17.6 (± 10.4) | 13.9 (± 4.8) | 14.5 (± 4.5) | 15.2 (± 4.4) |
| **Number of clinically sick mice, n [%]** | 9 [100] | 8 [32] | 13 [93] | 11 [65] |
| **Mean day of onset (± SD), sick mice only** | 15.8 (± 2.4) | 15.4 (± 3.0) | 15.0 (± 3.2) | 15.4 (± 2.5) |
| **Mean peak EAE score (± SD), sick mice only** | 2.2 (± 0.8) | 2.5 (± 1.1) | 2.5 (± 0.7) | 2.0 (± 0.7) |

Total number of mice immunized, mean age at immunization, number of clinically sick mice [incidence], mean day of disease onset, and mean peak EAE score in mice that developed clinical disease. SD, standard deviation.

# Reporting Summary

## Statistics

For all statistical analyses, confirm that the following items are present in the figure legend, table legend, main text, or Methods section.

| n/a | Confirmed | |
|---|---|---|
| ☐ | ☒ | The exact sample size (*n*) for each experimental group/condition, given as a discrete number and unit of measurement |
| ☐ | ☒ | A statement on whether measurements were taken from distinct samples or whether the same sample was measured repeatedly |
| ☐ | ☒ | The statistical test(s) used AND whether they are one- or two-sided *Only common tests should be described solely by name; describe more complex techniques in the Methods section.* |
| ☐ | ☒ | A description of all covariates tested |
| ☐ | ☒ | A description of any assumptions or corrections, such as tests of normality and adjustment for multiple comparisons |
| ☐ | ☒ | A full description of the statistical parameters including central tendency (e.g. means) or other basic estimates (e.g. regression coefficient) AND variation (e.g. standard deviation) or associated estimates of uncertainty (e.g. confidence intervals) |
| ☐ | ☒ | For null hypothesis testing, the test statistic (e.g. *F*, *t*, *r*) with confidence intervals, effect sizes, degrees of freedom and *P* value noted *Give P values as exact values whenever suitable.* |
| ☒ | ☐ | For Bayesian analysis, information on the choice of priors and Markov chain Monte Carlo settings |
| ☐ | ☒ | For hierarchical and complex designs, identification of the appropriate level for tests and full reporting of outcomes |
| ☐ | ☒ | Estimates of effect sizes (e.g. Cohen's *d*, Pearson's *r*), indicating how they were calculated |

*Our web collection on statistics for biologists contains articles on many of the points above.*

## Software and code

Policy information about availability of computer code

| Data collection | BD FACSDIVA (v.8.0.1); CytExpert (v.2.5.0.77); NextSeq 500 (Illumina); Leica Applications Suite X (v.3.5.6.21594); Leica LCS (v.2.6.1.5173); NIS Elements AR (v.5.20.00); Zeiss Axio Imager Z2 (ZEN 3.6); Leica TCS SP8 confocal microscope with HC PL APO CS2 40x/1.30NA objective; Imaris 9.7 software (Oxford Instruments); 3DHistech Pannoramic Flash II with Visopharm software (v2023.1). |
|---|---|
| Data analysis | Cell Ranger (v.7.1.0); Python (v.3.9.16); Scanpy (v.1.9.2); Scrublet (v.0.2.3); Velocyto (v.0.17.17); ScVelo (v.0.2.5); UnitVelo (v.0.2.5.2); QuPath (v.0.3.2); Prism (v.10.9.0); FlowJo (v.10.5.1); Adobe Illustrator 2022 (v.26.0.1); Adobe Photoshop CS6 (v.13.0); R (v.4.2.2); RStudio (v2023.06.01); Drop-seq pipeline (v1.12); edgeR (v.3.40.2); limma (v3.54.0); FactoMineR (v.2.6); factoextra (v.1.0.7); ggplot2 (v2.3.4); leidenalg package (0.9.1), DESeq2 (v1.40.2); GSEA software (v4.3.2). |

For manuscripts utilizing custom algorithms or software that are central to the research but not yet described in published literature, software must be made available to editors and reviewers. We strongly encourage code deposition in a community repository (e.g. GitHub). See the Nature Portfolio guidelines for submitting code & software for further information.

## Data

Policy information about availability of data

All manuscripts must include a data availability statement. This statement should provide the following information, where applicable:

- Accession codes, unique identifiers, or web links for publicly available datasets
- A description of any restrictions on data availability
- For clinical datasets or third party data, please ensure that the statement adheres to our policy

Reference genome (GRCm38) was used for alignment of bulk RNAseq data. Transcript and gene definitions were used according to GENCODE version M25. Single cell RNAseq reads were aligned to mouse reference genome mm10-2020 as provided by 10x genomics. NGS raw data, processed gene expression data and spliced/unspliced count data have been deposited in the GEO repository under the accession number GSE234188 (scRNAseq mouse) and GSE244363 (bulkRNAseq mouse). No custom algorithms were used in this study. Public thymic scRNAseq data was downloaded from https://zenodo.org/record/5500511 (DOI: 10.5281/zenodo.5500511)

## Research involving human participants, their data, or biological material

Policy information about studies with human participants or human data. See also policy information about sex, gender (identity/presentation), and sexual orientation and race, ethnicity and racism.

| | |
|---|---|
| Reporting on sex and gender | This information has not been collected due to its irrelevance for the study. |
| Reporting on race, ethnicity, or other socially relevant groupings | This information has not been collected due to its irrelevance for the study. |
| Population characteristics | This information has not been collected due to its irrelevance for the study. |
| Recruitment | Human tonsillar tissue was obtained from routine tonsillectomies by the Department of Otorhinolaryngology of the University Hospital "Klinikum rechts der Isar" of the Technical University of Munich (School of Medicine, Germany) with patient's informed consent. Human newborn thymic tissue was obtained from Andreas Büttner, University of Rostock (Approval by the local ethics committee). |
| Ethics oversight | The study was approved by the local ethics committee of the University of Rostock (A 2023-0038). |

Note that full information on the approval of the study protocol must also be provided in the manuscript.

# Field-specific reporting

Please select the one below that is the best fit for your research. If you are not sure, read the appropriate sections before making your selection.

☒ Life sciences        ☐ Behavioural & social sciences        ☐ Ecological, evolutionary & environmental sciences

For a reference copy of the document with all sections, see nature.com/documents/nr-reporting-summary-flat.pdf

# Life sciences study design

All studies must disclose on these points even when the disclosure is negative.

| | |
|---|---|
| Sample size | This study is purely explorative. No statistical methods were used to predetermine the sample size. Sample sizes were determined to be sufficient based on established standards for explorative studies in the field. Importantly, whenever statistical analysis was applied the sample size was ≥3 biological replicates per group. |
| Data exclusions | Figure 3 scRNAseq data: For quality control, cells with more than 15% mitochondrial gene counts were excluded as well as cells with more than 2% hemoglobin gene counts and cells with less than 5% ribosomal gene counts. Cells with less than 200 detected genes and cells with more than 1E4 counts per cell were also removed. Doublet exclusion was performed using scrublet.\n\nFigure 3 bulkRNA seq data: For quality control, genes with expression values lower than 1 count per million in at least 4 samples were excluded. To increase our power, we limited our analysis to genes in the serumantibodyome, as explained in the Methods section. |
| Replication | Biological replicates were used in this study to ensure reproducibility. Most of the experiments were repeated at least twice as stated in the figure legends. All attempts at replication were successful. |
| Randomization | Randomization was not necessary for this study since there was no particular intervention (e.g. treatment) planned. Whenever possible littermate controls were used and equal number of mice were allocated to different groups within one cage to rule out cage effects. |
| Blinding | For mice experiments blinding during data collecting was usually not possible due to required cage labeling. However, scoring of EAE mice was occasionally performed by members of the staff who were not familiar with the experimental design. Furthermore, data analysis used in this |

study was strictly quantitative (and not subjective) and therefore blinding was not necessary.
For the human data, samples were provided to us in a blinded manner by different collaborators without any patient's individual information.

# Reporting for specific materials, systems and methods

We require information from authors about some types of materials, experimental systems and methods used in many studies. Here, indicate whether each material, system or method listed is relevant to your study. If you are not sure if a list item applies to your research, read the appropriate section before selecting a response.

## Materials & experimental systems

| n/a | Involved in the study |
|-----|-----------------------|
| ☐ | ☒ Antibodies |
| ☐ | ☒ Eukaryotic cell lines |
| ☒ | ☐ Palaeontology and archaeology |
| ☐ | ☒ Animals and other organisms |
| ☒ | ☐ Clinical data |
| ☒ | ☐ Dual use research of concern |
| ☒ | ☐ Plants |

## Methods

| n/a | Involved in the study |
|-----|-----------------------|
| ☒ | ☐ ChIP-seq |
| ☐ | ☒ Flow cytometry |
| ☒ | ☐ MRI-based neuroimaging |

## Antibodies

| Antibodies used | The following antibodies were used for flow cytometry:<br>Arm. Hamster BV421 CD95 (Jo2, BD Biosciences, #562633, 1:300, RRID: AB_2737690)<br>Arm. Hamster PE CD95 (Jo2, BD Biosciences, #554258, 1:300, RRID: AB_395330)<br>Arm. Hamster FITC CD3e (145-2C11, eBioscience, #11-0031, 1:300, RRID: AB_464882)<br>Arm. Hamster FITC TCRß (H57-597, BD, #553171, 1:300, RRID: AB_394683)<br>Arm. Hamster PE-Cy7 CD11c (N418, BioLegend, #117318, 1:300, RRID: AB_493569)<br>Mouse Alexa Fluor 488 IgD (IA6-2, Biolegend, #348216, 1:200, RRID: AB_11150595)<br>Mouse APC CD10 (97C5, Miltenyi, #130-119-675, 1:100, RRID: AB_2660858)<br>Mouse APC-CD19 (SJ25C1, Biolegend, #363005, 1:300, RRID: AB_2564127)<br>Mouse BV421 Bcl6 (K112-91, BD Biosciences, #563363, 1:200, RRID: AB_2738159)<br>Mouse BV421 CD2 (RPA-2.10, BD Biossciences, #562667, 1:300, RRID: AB_2737695)<br>Mouse BV510 CD14 (M5E2, BioLegend, #301842, 1:300, RRID: AB_2561379)<br>Mouse BV421 CD19 (HIB19, BioLegend, #302234, 1:200, RRID: AB_11142678)<br>Mouse BV510 CD27 (M-T271, Biolegend, #356420, 1:200, RRID: AB_2562603)<br>Mouse BV421 CD45.1 (A20, BioLegend, #110732, 1:300, RRID: AB_10896425)<br>Mouse BV421 NK1.1 (PK136, eBioscience, #48-5941, 1:300, RRID: AB_2043877)<br>Mouse BV510 NK1.1 (PK136, BioLegend, #108738, 1:300, RRID: AB_2562216)<br>Mouse BV711 CD38 (HIT2, BD Biosciences, #563965, 1:100, RRID: AB_2738516)<br>Mouse BV786 CD45.2 (104, BD Biosciences, #563686, 1:300, RRID: AB_2738375)<br>Mouse BV786 RORyt (Q31-378 , BD Biosciences, #564723, 1:100, RRID: AB_2738916)<br>Mouse FITC IgM (AF6-78, BD Biosciences, #553520, 1:300, RRID: AB_394901)<br>Mouse FITC NK1.1 (PK136, eBioscience, #11-5941, 1:300, RRID: AB_465319)<br>Mouse PE CD3 (UCHT1, Beckman Coulter, #IM1282U, 1:200, RRID: AB_467059)<br>Mouse PE-Cy7 T-bet (4B10, BioLegend, #644824, 1:200, RRID: AB_2561760)<br>Mouse PE-Cy7-CD3 (SK7, BD Biosciences, #341111, 1:300, RRID: AB_10596664)<br>Mouse PerCP/Cy5.5-CD27 (O323, BioLegend, #302819, 1:300, RRID: AB_11218994)<br>Goat AF488 IgG (H+L) (ThermoFisher, #A-11029, 1:100, RRID: AB_2534088)<br>Rat CD16/CD32 (2.4G2, BD Biosciences, #553142, 1:100, RRID: AB_394657)<br>Rat AF488 Foxp3 (FJK-16s, eBioscience, #53-5773, 1:200, RRID: AB_763537)<br>Rat AF647 CD5 (53-7.3, BioLegend, #100614, 1:300, RRID: AB_493168)<br>Rat APC CD19 (1D3, BD Biosciences, #550992, 1:300, RRID: AB_398483)<br>Rat APC IgD (11-26c.2a, BioLegend, #405714, 1:300, RRID: AB_10645480)<br>Rat APC Ly6A/E (D7, BioLegend, #122512, 1:300, RRID: AB_756196)<br>Rat APC MHC-II (M5/114.15.2, eBioscience, #17-5321, 1:300, RRID: AB_469454)<br>Rat APC PD-1 (29F.1A12, BioLegend, #135210, 1:300, RRID: AB_2251944)<br>Rat APC-R700 B220 (RA3-6B2, BioLegend, #103232, 1:300, RRID: AB_493716)<br>Rat BV421 B220 (RA3-6B2, BioLegend, #103227, 1:300, RRID: AB_492877)<br>Rat BV421 EpCAM (G8.8, BioLegend, #118225, 1:300, RRID: AB_2563983)<br>Rat BV510 CD11b (M1/70, BioLegend, #101263, 1:300, RRID: AB_2561390)<br>Rat BV510 CD19 (6D5, BioLegend, #115546, 1:300, RRID: AB_2562136)<br>Rat BV510 F4/80 (BM8, BioLegend, #123135, 1:300, RRID: AB_2562622)<br>Rat BV786 CD19 (6D5, Biolegend, #115543, 1:300, RRID: AB_11218994)<br>Rat BV786 Vb6 (RR4-7, BD Biosciences, #744595, 1:200, RRID: AB_2742344)<br>Rat FITC CD11b (M1/70, BioLegend, #101205, 1:300, RRID: AB_312788)<br>Rat FITC CD4 (GK1.5, BD Biosciences, #553729, 1:500, RRID: AB_395013)<br>Rat FITC F4/80 (BM8, abcam, #ab60343, 1:300, RRID: AB_2637191)<br>Rat PE CD19 (1D3, BD Biosciences, #557399, 1:300, RRID: AB_395050)<br>Rat PE CD8a (53-6.7, BD Bioscience, #553033, 1:300, RRID: AB_394571) |

Rat PE Ly6A/E (D7, Thermo, #12-5981-82, 1:300, RRID: AB_466086)
Rat PE GL7 (GL7, BD Biosciences, #561530, 1:300, RRID: AB_10715834)
Rat PE-Cy7 B220 (RA3-6B2, BioLegend, #103222, 1:300, RRID: AB_313004)
Rat PE-Cy7 CD21 (7E9, BioLegend, #123420, 1:300, RRID: AB_1953276)
Rat PerCP CD19 (6D5, BioLegend, #115532, 1:300, RRID: AB_893278)
Rat PerCP-Cy5.5 B220 (RA3-6B2, BioLegend, #103236, 1:300, RRID: AB_893356)
Rat PerCP-eF710 CD4 (RM4-5, eBioscience, #46-0042, 1:500, RRID: AB_1834431)

The following antibodies were used in cell hashing: TotalSeq-C anti-mouse Hashtag 1 to 9 (M1/42; 39-F11, Biolegend, #155861, #155863, #155865, #155867 ,#155869, #155871, #155873, #155875, #155877). All antibodies were diluted 1:100.

The following antibodies were used in T cell cultures:
Mouse anti-CD3e (500A2, eBioscience, #16-0033-86, 1:2000, RRID: AB_842783) and
Rat anti-CD40 (FGK45, BioXCell, BE0016-2, AB_1107601)
AffiniPure F(ab')₂ Fragment goat anti-IgG (H+L) (Jackson Immuno Research, 109-006-088, AB_2337549)
AffiniPure F(ab')₂ Fragment goat anti-IgG + anti-IgM (H+L) (Jackson Immuno Research, 115-006-068, AB_2338471 )

The following antibodies were used in immunohistochemistry:
Mouse CD20 (L26, Dako, #M0755, 1:500, validated by manufacturer)
Mouse GFAP (G-A-5, Sigma-Aldrich, #G6171, 1:400, validated by manufacturer)
Rabbit AQP4 (Sigma, #HPA014784, 1:2000, validated by manufacturer)
Rabbit Bcl6 (D65C10, Cell signaling, #5650, 1:100, validated by manufacturer)
Rabbit CD19 (D4V4B, Cell signaling, #90176, 1:400, validated by manufacturer)
Rabbit EpCAM (abcam, #ab71916, 1:200, validated by manufacturer)
Rat B220 (RA3-6B2, BD Biosciences, #550286, RRID: AB_394619)
Rat CD19 (6OMP31, eBiosciences, #14-0194-80, 1:200, RRID: AB_2637171)
Rat CD45 (30-F11, ThermoFisher, #14-0451--82, 1:500, RRID: AB_467251)
Donkey polyclonal secondary antibody to mouse IgG (H&L), preasorbed, Alexa Fluor 568 (Thermo, #A10037, 1:2000, RRID: AB_2534013)
Donkey polyclonal secondary antibody to rabbit IgG (H&L), preasorbed, Alexa Fluor 488 (Thermo, #A21206, 1:2000, RRID: AB_2535792)
Donkey polyclonal secondary antibody to rabbit IgG (H&L), preasorbed, Alexa Fluor 647 (Life, #A31573, 1:200, RRID: AB_2536183)
Goat polyclonal secondary antibody to rabbit IgG (H&L), preasorbed, Alexa Fluor 647 (Abcam, #ab150087, 1:400, validated by manufacturer)
Goat polyclonal secondary antibody to rat IgG, conjugated to horseradish peroxidase (HRP) (Vector, #MP-7444, 1:3000, validated by manufacturer).

| Validation | All antibodies are commercially available, widely used and validated by the manufacturer for the same application and species as used in this study. Please find details about validation in the RRID registry (https://scicrunch.org/resources/Antibodies/search?l=&q=%2A, RRID identifiers are listed above) and the manufacturer's website. In addition, control stainings were performed with cell types known to express or lack the relevant antigens. |
|---|---|

## Eukaryotic cell lines

Policy information about cell lines and Sex and Gender in Research

| Cell line source(s) | The T cell hybridoma cell line A5 was kindly provided by L. Klein (LMU Munich, Germany) as described in the method section. The fibroblastic YKL cell line was kindly provided by M. Schmidt-Supprian (TU Munich, Gemany) as described in the method section. The packing cell line Plat-E was kindly provided by D. Busch (TU Munich, Germany) as described in the method section. Various manufacturers offer these cells. |
|---|---|
| Authentication | Cell line was authenticated prior to receipt by the commercial vendor using the STR method. |
| Mycoplasma contamination | Cells were not tested for mycoplasma after receipt. |
| Commonly misidentified lines (See ICLAC register) | No commonly misidentified cell lines were used. |

## Animals and other research organisms

Policy information about studies involving animals; ARRIVE guidelines recommended for reporting animal research, and Sex and Gender in Research

| Laboratory animals | For animal experiments sex- and age-matched female and male mice on C57BL/6J background were used. The mice used in studies investigating thymi were between 6 and 10 weeks old, while those used for immunization were between 10 and 18 weeks old.

C57BL/6J, Aire-flox/flox, Rag1-/-, Cd40–/– and TCRa–/– mice were obtained from The Jackson Laboratory.
Aqp4–/– mice were kindly provided by A. Verkman (University of California, San Francisco UCSF).
Aqp4flox/flox mice were kindly provided by O. P. Ottersen (University of Oslo).
Foxn1-Cre mice were kindly provided by L. Klein (Ludwig Maximillians University, Munich).
Mb1-Cre mice were kindly provided by M. Schmidt-Supprian (Technical University of Munich).
DEREG mice were kindly provided by Tim Sparwasser (Johannes Guttenberg University of Mainz). |
|---|---|

Mice were housed with a dark/light cycle of 12 hours, a temperature of 20-24°C, and a humidity of 45-60%.

| | |
|---|---|
| Wild animals | The study did not involve wild animals. |
| Reporting on sex | Male and female mice were used equally for all experiments. |
| Field-collected samples | The study did not involve samples collected form the field. |
| Ethics oversight | Experimental procedures were approved and performed according to the the Bavarian state authorities as indicated in the method section. |

Note that full information on the approval of the study protocol must also be provided in the manuscript.

# Flow Cytometry

## Plots

Confirm that:

☒ The axis labels state the marker and fluorochrome used (e.g. CD4-FITC).

☒ The axis scales are clearly visible. Include numbers along axes only for bottom left plot of group (a 'group' is an analysis of identical markers).

☒ All plots are contour plots with outliers or pseudocolor plots.

☒ A numerical value for number of cells or percentage (with statistics) is provided.

## Methodology

| | |
|---|---|
| Sample preparation | Sample preparation as described in the methods. |
| Instrument | CytoFLEX (Beckman Coulter); FACS Aria III (BD Biosciences) |
| Software | BD FACSDIVA (v.8.0.1); CytExpert (v.2.3.1.22); FlowJo (v10.9.0) |
| Cell population abundance | Cell sorting was performed with the strictest purity setting (4-Way Purity), ensuring a purity of >95%. |
| Gating strategy | See flow cytometric gating strategy as clarified in the Extended data figures. Debris and dead cells were gated out based on a distinctive FSC and SSC gate specific for lymphocytes as is common practice. Singlets were gated based on FSC-A/FSC-H and SSC-A/SSC-H. Dump channel exclusion of dead cells (Aqua LiveDead positive), CD11b, NK1.1, F4/80 and either CD19 or CD3 as further clarified in the method section. |

☒ Tick this box to confirm that a figure exemplifying the gating strategy is provided in the Supplementary Information.

