## [Peer Review File · Nature]

Manuscript Title: B cells control autoimmunity against AQP4 by negative selection of antigen-specific thymocytes

Editorial Notes:

Reviewer Comments & Author Rebuttals

Reviewer Reports on the Initial Version:

Referees' comments:

Referee #1 (Remarks to the Author):

This study uses a previously defined mouse model to better understand T cell tolerance to aquaporin 4 (Aqp4), the target autoantigen in human neuromyelitis optic (NMO). The authors find that both medullary thymic epithelial cells and B cells express Aqp4, and that B cells depend on CD40 dependent activation (previously coined “licensing” by one of the authors) is required for Aqp4 expression. Using IAb tetramers presenting a previously identified Aqp4 epitope—P41—to quantify Aqp4 reactive T cells, the authors argue that Aqp4 expression in B cells (but not mTEC) is essential for T cell tolerance. Furthermore, they suggest the mechanism of tolerance is clonal deletion (but not Treg induction). Lastly, they show that mice lacking Aqp4 in B cells exhibit EAE symptoms, including retinal infiltration, when immunized with P41. Overall, it is proposed by B cell licensing in the thymus is a novel mechanism required for tolerance to “germinal center-associated” antigens.

These are novel and interesting findings, although the conclusions are not consistent with the data on two key points. Furthermore the report, as is does not provide insight as to the scope of this potentially unique tolerance mechanism.

1. The finding that “negative selection of Aqp4 specific thymocytes is entirely dependent on the expression and presentation of Aqp4 by thymic B cells.” is not supported by the data. For example, in Figure 1a, Aqp4 specific T cells are expanded less in Aqp4^{-/-} → WT chimeric mice than in Aqp4^{-/-} → Aqp4^{-/-} chimeric mice, implying a role for Aqp4 expressed by radioresistant cells. Likewise, in Figure 2b, tolerance to Aqp4 was apparent in both Aqp4^{delta}TEC and Aqp4^{delta}B mice. Lastly, there was a lower titre and MFI of autoantibodies to Aqp4 in Aqp4^{delta}B mice than Aqp4^{-/-} mice, again implying a role for non-B cell derived Aqp4 in tolerance. Although it’s clear that B cells efficiently induce tolerance to Aqp4, and it is more accurate to say that both TEC and B cells can induce T cell tolerance to Aqp4.

Were the data to support the notion that B cells are the sole APC required for T cell tolerance to Aqp4 despite its thymic expression by mTEC, then further analysis would be needed to understand why this is the case. But as it is, the data on tolerance seem consistent with the expression of Aqp4 in the thymus.

2. The conclusion that Aqp4 expression only impacts clonal deletion and does not alter Treg selection is also not supported by the data. In Figure 2c, Treg frequency is increased among Aqp4 specific T cells in WT mice (that express Aqp4) compared to Aqp4^{-/-} mice, as expected. Likewise, mice that express Aqp4 only in B cells (Aqp4^{delta}TEC) have increased Treg frequency, suggesting that B cell expression of Aqp4 supports clonal diversion to the T reg lineage.

3. The authors assert that thymic B cells tolerize against a group of germinal center-associated self-antigens in order to prevent inappropriate help by Tfh cells during GC reactions. This is a very attractive idea. However, the current study addresses only a single antigen, and disease is only shown in the context of immunization with the self-antigen. If the authors could provide more information about the potential scope of this process (how many CD40 dependent antigens are expressed in thymic B cells?) or show autoimmunity to Aqp4 can develop in situations besides Aqp4 immunization (e.g. with heterologous infections, or chronic infection?) then the impact of the study would be much greater. [The abstract states that “thymic B cells emulate a CD40-stimulated B cell transcriptome”, but there are no data that directly address this. They also state “Aqp4 appears to be a paradigmatic antigen in this context”, but what evidence is this based on?]

4. As an animal model of neuromyelitis optica, it is intriguing that the authors state that retina infiltration was only observed in Aqp4^{delta}B mice immunized with P41, but not those immunized with MOG. But this data is not shown in Figure 6! This data should be included. Also, the authors should compare the retina infiltration in Aqp4^{-/-}, Aqp4^{delta}B, and WT mice (immunized with P41).

Other suggestions for improving the report:

5. Figure 3 shows that CD40 can upregulate Aqp4 in naïve B cells, but it would be more definitive to show that thymic B cells from CD40 deficient mice do not express Aqp4. It would be interesting to know if TEC expression of Aqp4 was also dependent on CD40, since Aqp4 expression in TEC was Aire-independent (Figure 3i).

6. Figure 4 is problematic. Why are there no DP thymocytes present amongst retrogenic derived cells in Aqp4^{delta}B mixed BMC, but they are present in WT mixed BMC? Also, the number of B cells per thymus in these chimeras is extremely low (Figure 4d). Perhaps the fact that radiation destroys the thymic niche for B cells makes this approach unsuitable for assessing the role of B cell derived antigens in tolerance. Was there any expression of Foxp3 in retrogenic CD4 SP?

7. Since the Aqp4^{delta}TEC and Aqp4^{delta}B mice are critical for the conclusions made, it is imperative that the authors provide evidence of appropriate tissue specificity of the deletion of Aqp4, especially since both of these Cre lines can lead to germline deletion. Related to this issue about rigor, please state in the legend exactly what the “Aqp4^{delta}B” mice are (presumably they are MB1Cre^{+/+}-Aqp4^{fl/fl}) and describe the breeding strategy used.

8. In figure 6, WT mice immunized with P41 show a reduced incidence of EAE when compared to MOG immunized Aqp4-B-delta mice. However, the clinical scoring of these WT mice is not shown in figure 6b. Is there a difference in clinical score, or is it only that fewer mice get sick, but at the same severity as

P41-immunized Aqp4-B-delta mice? If this the case, how does this fit into the hypothesis presented?

9. In Figure 1f, I see little to no co-expression of CD20 and AQP4, is there further data to support the conclusion that human thymic B cells express AQP4 at steady state?

10. In the methods, both AQP4 peptide and full-length protein are noted to be used for EAE induction. In Figure 6, EAE is only showed for P41 peptide induction. In MOG induced EAE, use of full-length protein vs peptide alters the APC requirements. Does full length AQP4 protein give different EAE than when using peptide in this conditional knock out system?

Minor points

11. Material and methods needs to be completed. The mouse immunization protocol doesn't include the adjuvant. The tetramer staining protocol also does not include an enrichment step. If an enrichment step was not performed to identify tetramer positive cells in naïve mice, this would be concerning for the interpretation of naïve repertoire experiments.

12. Panel 4a, please state how many weeks after reconstitution the animals were analyzed.

13. Panel 5b, please state whether the flow analysis of "Tfh" and "GC B cells" was from spleen or LN.

14. Line 240 has a typographical error.

Referee #2 (Remarks to the Author):

In this manuscript Afzali et al demonstrate that thymic B cells are the primary source of AQP4 expression in the thymus and play a non-redundant role in the induction of T cell tolerance to this specific and clinically relevant self-antigen. The role of thymic B cells has remained quiet fuzzy with some evidence for their possible role in tolerance induction, but this was thought to be rather redundant and inferior to mTEC. To the best of my knowledge, this is the first study that experimentally shows a clear and non-redundant role for thymic B cells in tolerance induction.

This is a great and very elegant study, which provides a very convincing and needed evidence for the rather enigmatic role of B cells in the thymus, by using AQP4 as a model self-antigen. The methodology used to address this question is appropriate and elegant - and is primarily based on tracking AQP4-reactive T cells using tetramers in various mouse models based either on reciprocal BM chimeras or B- or TEC specific knockouts of AQP4.

Moreover, the study has important clinical implications, as AQP4 is a well-established autoimmune target in humans and thus these discoveries could potentially be instrumental in designing therapeutic strategies for treatment of human autoimmune diseases, in which AQP4 is specifically targeted, including neuromyelitis optica.

The study is written very clearly with a logical flow and the statements are supported by the data

While I would be happy to see this important study being published in journals like Nature in the future, I think the current version would benefit from further addressing the following points:

1) While the authors provide very convincing evidence that B cells are both necessary and sufficient to induce tolerance to AQP4 in the thymus, through elimination of AQP4-specific T cells and induction of AQP4-specific Treg, Fig 2a,b suggest that minor role may also be played by AQP4-expressing TEC. The reason for this notion is that TEC-specific deletion of AQP4 still results in generation of higher number of AQP4 specific T cells than observed for control mice. Similarly, Fig1a suggests that transfer of AQP4 KO BM into AQP4 KO recipients has bigger effect than AQP4 KO BM into AQP4 WT recipients (the authors should show the P val for this comparison).

a. It would be very informative if the authors could provide more info (e.g. using publicly available scRNAseq datasets) which of the TEC compartments expresses AQP4? Is it expressed in Aire+ mTEC in an Aire-dependent manner or is it expressed by one of the identified TEC mimetics (e.g. the neuroTEC) in an Aire- independent manner?

b. Moreover, most of the TEC mimetics express low levels of MHCII, suggesting that they are poor antigen presenting cells and probably serve only as a passive source of AQP4. I think the study would benefit from providing this info, as it could possibly explain why the expression of AQP4 in TEC is redundant and in B cells it is not.

c. Correspondingly, if the AQP4 is expressed by one of the TEC mimetics, are thymic B cells somehow associated with these AQP4+ TEC?

2) The authors convincingly demonstrate that both CD40 and IL21 signaling induce AQP4 expression in B cells. This is very interesting and suggests that the contact with potentially self-reactive T cell provides the B cells with the necessary ligands (i.e. CD40L, IL21).

a. Could the authors show (e.g. using publicly available scRNAseq data for mouse/human thymus) what is the primary source of IL21 in the thymus? I assume it is the CD4T cell, but it would be beneficially to show that.

b. Does the CD40/IL21 signaling in thymic B cells induce expression of TRA genes other than AQP4? What other relevant genes (e.g. antigen presentation, cytokine, chemokine, etc.) are induced by this signaling? The study would benefit by providing this info in much more detailed manner.

3) Could the authors show the generation of AQP4-specific antibodies (either in ELISA to relevant peptides or WB from AQP4-expressing cells in non-immunized vs., immunized ctrl, or in mice B- or TEC specific deletion of AQP4?

4) Fig 6a, seems to be missing an important control – WT-MOG immunized mice as a baseline

Referee #3 (Remarks to the Author):

In this study, Afzali et al focused on a self-antigen which acts as the target in the human autoimmune disease neuromyelitis optica (NMO) and investigated the mechanisms by which immune tolerance against that self-antigen is established. The author's data show that the self-antigen AQP4 expressed in

thymic B cells is important for inducing negative selection of immature T cells reactive to AQP4-derived peptide and establishing immune tolerance to AQP4, contributing to prevent AQP4-related autoimmune disorders. AQP4 expression in CD40-stimulated B cells was observed in humans as well as in mice, suggesting that humans B cells also contribute to the self-antigen expression and tolerance induction and that failure of this tolerance mechanism might result in autoimmune disease. Although the finding that thymic B cells express self-antigens in the thymus to induce immune tolerance against such self-antigens is not so new and has been previously reported by the authors themselves, many of the experiments performed in the present paper are logically well connected to each other and the results appear clear. The authors emphasize that their present data strongly support the hypothesis that thymic B cells tolerize the T cell repertoire against B cell antigen to prevent inappropriate immune responses upon inappropriate interaction between Tfh cells and B cells during GC formation in the peripheral lymphoid tissues. How this mechanism breaks down in patients with NMO remains unclear and needs future studies.

This study is well designed and manuscript is well written. However, some concerns are raised about the experimental methods and data related to the key experiments in this study. The authors need to address the following concerns.

#1

The method of P41-IAb tetramer staining is unclear. It is not stated whether the authors perform tetramer enrichment and how the tetramer+ cells were counted (by using counting beads?). These methods are essential to accurately measure the number of tetramer+ cells (Taniguchi et al, PNAS 2012). What does 'in sec LyTi (#)' mean in Fig.1a and Fig 2a,b,e?

#2

In Fig. 1a, the number of P41/IAb tetramer+ cells in KO -> WT BMC mice is nearly two orders of magnitude lower than in KO -> KO BMC mice. This data may indicate limited contribution of hematopoietic cells to the negative selection of AQP4-specific CD4 T cells. Or is this due to the low contribution of KO hematopoietic cells in these BMC mice? If so, how could the data from WT -> KO mice be interpreted?

#3

In Fig. 1c, the mRNA expression level of AQP4 is 10-fold higher in TECs than in thymic B cells. According to the ImmGen or BioGPS databases, the expression of AQP4 is not detected in both TECs and B cells. In addition, scRNA-seq data in Fig. 3 shows very faint mRNA expression of AQP4 in thymic B cells (less than 1/1000th of astrocytes). It is puzzling that despite such low levels of mRNA expression, AQP4 protein expression is readily detected in the thymus by immunohistochemistry in Fig. 1f. Is AQP4 signal detected in CD20-positive B cells? It does not appear so, though.

Of further importance to point out in this immunohistochemistry data (Fig. 1f) is that the detection specificity of anti-AQP4 antibody is not demonstrated. The authors should show the immunohistostaining with the anti-AQP4 antibody of the thymus from AQP4-KO and B-cell-specific AQP4-cKO mice.

#4

In Fig. 2a, the number of P41-specific CD4 T cells in non-immunized mice should be examined in the thymus as well. As described above, tetramer enrichment and counting beads are essential. Because the main issue of this paper is negative selection in the thymus, data of P41-specific CD4SP thymocytes would certainly reinforce the authors' conclusion. To further clarify the role of thymic B cells, the authors should examine P41-specific CD4SP thymocytes in Ighm-KO (muMT) or CD40-KO mice.

#5

It is questionable whether experiments in Fig. 4 have been performed properly. The flow cytometry profile of AQP4 TCR retrogenic delta-B mice looks very strange; Why do the mice have CD8SP cells? Do the CD4SP and CD8SP cells express AQP4-specific TCR? Why are DP thymocytes in the mice so few? Are the mice stressed, resulting in thymus hypoplasia due to decrease of DP thymocytes?

The authors should show data on total thymocytes numbers and flow cytometry profiles for CD45.1 and CD45.2 to evaluate the chimerism of the BMC mice (as supplementary data). In addition, the frequency of CD4SP cells in WT BMC mice shown in Fig. 4b does not match the mean frequency of the same group of mice shown in Fig. 4c. A representative FCM profile should be shown.

In Fig. 4b, flow cytometry profiles of polyclonal (CD45.2) cells look very different between WT and delta-B mice. Are these data from the same experiment and from the same individual as the AQP4 TCR retrogenic (CD45.1/2) cells on the right panel?

#6

Fig. 3g shows that spleen B cells expressing AQP4 can stimulate T cells with AQP4-specific TCR in vitro, and the authors stated in the abstract 'the negative selection of AQP4-specific thymocytes is entirely dependent on the expression and presentation of AQP4 by thymic B cells'. However, this might be overstatement. Is there any data showing the AQP4 antigen presentation by B cells in vivo? At least, the authors should show the data that thymic B cells isolated from WT mice can stimulate AQP4-reactive T cells in vitro but those from AQP4-KO or delta-B mice cannot.

#7

Related to #6, there should be more discussion (or data if possible) on how thymic B cells present antigen peptides derived from their own proteins with MHC class II. Loading of MHC class II by endogenous membrane proteins (shown in Ref. 18) is a phenomenon observed during viral infection or in the absence of usual antigen presentation pathway, so it is not clear whether it also works in thymic B cells. As reported previously (Nedjic et al, Nature 2008), do thymic B cells have the high autophagy activity, similar to TECs and DCs, and degrade ER-targeted membrane proteins to load MHC class II?

Minor comments:

#8

The legend of Fig. 1c describes 'medullary thymic epithelial cells (mTECs, live CD45-EpCAM+)', but this is wrong. CD45-EpCAM+ cells are just thymic epithelial cells (TECs), not mTECs.

#9

In Fig. 1f, what is blue-colored? DNA stained with DAPI?

#10

In Fig. 6, why are the N numbers different between panels a and b? In Fig. 6b, the data of MOG- or P41-immunized WT mice should be shown and compared.

Author Rebuttals to Initial Comments:**Referee #1 (Remarks to the Author):**

This study uses a previously defined mouse model to better understand T cell tolerance to aquaporin 4 (Aqp4), the target autoantigen in human neuromyelitis optica (NMO). The authors find that both medullary thymic epithelial cells and B cells express Aqp4, and that B cells depend on CD40-dependent activation (previously coined “licensing” by one of the authors) is required for Aqp4 expression. Using IAb tetramers presenting a previously identified Aqp4 epitope—P41—to quantify Aqp4 reactive T cells, the authors argue that Aqp4 expression in B cells (but not mTEC) is essential for T cell tolerance. Furthermore, they suggest the mechanism of tolerance is clonal deletion (but not Treg induction). Lastly, they show that mice lacking Aqp4 in B cells exhibit EAE symptoms, including retinal infiltration, when immunized with P41. Overall, it is proposed by B cell licensing in the thymus is a novel mechanism required for tolerance to “germinal center-associated” antigens.

These are novel and interesting findings, although the conclusions are not consistent with the data on two key points. Furthermore, the report, as is does not provide insight as to the scope of this potentially unique tolerance mechanism.

We want to thank this reviewer for the overall encouraging assessment. As detailed below, we have performed additional experiments to address this referee's concerns (especially regarding the key points that are mentioned in this review).

1. The finding that “negative selection of Aqp4 specific thymocytes is entirely dependent on the expression and presentation of Aqp4 by thymic B cells.” is not supported by the data. For example, in Figure 1a, Aqp4 specific T cells are expanded less in Aqp4^{-/-} → WT chimeric mice than in Aqp4^{-/-} → Aqp4^{-/-} chimeric mice, implying a role for Aqp4 expressed by radioresistant cells. Likewise, in Figure 2b, tolerance to Aqp4 was apparent in both Aqp4^{delta}TEC and Aqp4^{delta}B mice. Lastly, there was a lower titre and MFI of autoantibodies to Aqp4 in Aqp4^{delta}B mice than Aqp4^{-/-} mice, again implying a role for non-B cell-derived Aqp4 in tolerance. Although it's clear that B cells efficiently induce tolerance to Aqp4, and it is more accurate to say that both TEC and B cells can induce T cell tolerance to Aqp4.

We apologize if the wording in the original version of this manuscript was overstating our discovery. In fact, our data show that TECs and B cells are two sources of endogenous AQP4 in the thymus. Thymic B cells are necessary and sufficient to delete AQP4-specific thymocytes with the functional outcome that the peripheral T cell repertoire of wild-type but not Aqp4^{ΔB} mice is largely depleted of conventional AQP4-specific T cells. Yet, AQP4-expressing TECs might also (directly or indirectly) contribute to the shaping of the AQP4-specific T cell repertoire.

In order to directly address this point, we measured the frequencies and absolute numbers of conventional and Foxp3⁺ CD4⁺ single-positive (SP) thymocytes in wild-type, Aqp4^{-/-}, Aqp4^{ΔTEC}, and Aqp4^{ΔB} mice. The

fraction of Foxp3⁺ SP cells was higher in *Aqp4*^{ΔTEC} than in wild-type mice, suggesting that thymic B cells can more efficiently divert AQP4-specific thymocytes into the Treg cell lineage when TECs lack AQP4 than when they are sufficient in AQP4. While these data indicate that TECs are involved in the shaping of the AQP4-specific TCR repertoire, the key finding of our study, i.e. that the major load of negative selection is supported by B cells and not by TECs, holds up. We provide this novel data as additional panels in **Fig. 4 (panels Fig. 4a and b)**. The text of the manuscript was amended accordingly.

Were the data to support the notion that B cells are the sole APC required for T cell tolerance to *Aqp4* despite its thymic expression by mTEC, then further analysis would be needed to understand why this is the case. But as it is, the data on tolerance seem consistent with the expression of *Aqp4* in the thymus.

Our data support that B cells are the major negative selectors of AQP4-specific thymocytes despite the expression of AQP4 by TECs (that has also been reported by others [3]). Our direct loss-of-function (gene knock-out) approach in TECs and B cells is – in our minds – the most stringent approach to investigate this question. Whether there is a cooperation between AQP4 TECs and AQP4-expressing thymic B cells, needs to be explored in further studies. However, in order to narrow down this reviewer's particular question as to why B cells are so efficient in deleting AQP4-specific thymocytes, we performed MHC class II staining on thymic B cells in comparison with TECs and found that thymic B cells express MHC class II at much higher levels than TECs (and in particular those TEC subsets that also express AQP4), suggesting that the APC capacity of AQP4⁺ thymic B cells may largely surpass that of TECs, in particular since B cells are also strategically positioned at the corticomedullary boundary in the thymus. Using available data sets of scRNAseq data of thymic cells from the human cell atlas [4], we confirmed that the B cell subsets present in the human thymus exhibit higher expression of MHC class II molecules than any subset of TECs. These data are now reported in the **revised Extended Data Fig. 2**.

2. The conclusion that *Aqp4* expression only impacts clonal deletion and does not alter Treg selection is also not supported by the data. In Figure 2c, Treg frequency is increased among *Aqp4* specific T cells in WT mice (that express *Aqp4*) compared to *Aqp4*^{-/-} mice, as expected. Likewise, mice that express *Aqp4* only in B cells (*Aqp4*^{ΔTEC}) have increased Treg frequency, suggesting that B cell expression of *Aqp4* supports clonal diversion to the T reg lineage.

It is correct that *Aqp4*^{ΔTEC} mice have a similarly high fraction of P41/I-A^{b+} (AQP4-specific) Treg cells as wild-type mice in the peripheral immune compartment, and this fraction is significantly higher than the Treg cell fraction in *Aqp4*^{-/-} mice and in *Aqp4*^{ΔB} mice. In order to directly test the capacity of the relevant thymic AQP4-expressing APC types, i.e. TECs and B cells, to divert AQP4-specific thymocytes into the Treg cell lineage, we crossed a Foxp3 (GFP) reporter into our conditional *Aqp4*^{ΔTEC} mice and *Aqp4*^{ΔB} mice and measured the Foxp3 (GFP)⁺ fraction within the P41/I-A^{b+} CD4⁺ single-positive (SP) thymocytes. In fact, in *Aqp4*^{ΔTEC} mice, in which B cells are one remaining endogenous source of AQP4 in the thymus, about 40 percent of the few remaining P41/I A^{b+} SP thymocytes were Foxp3 (GFP)⁺, suggesting that B cells can in principle divert AQP4-specific thymocytes into the Treg cell lineage. However, this diversion is not clinically relevant since we failed to induce disease by immunization with AQP4(201-220) in Treg cell-depleted *Aqp4*^{ΔTEC} x DREG mice (that we had also crossed and in which Foxp3⁺ Treg cells express the DTx receptor so that they can be depleted by injection of DTx). See **Extended Data Fig. 6a-c** of the revised manuscript. Therefore, our statement that thymic B cells are major deleters of AQP4-specific T cells is supported by data. Yet, the reviewer is correct that B cells can, in principle, also divert AQP4-specific thymocytes into the Treg cell lineage. We amended the manuscript accordingly (**p. 11**) and provided the 'thymus tetramer

staining' data in the **revised Fig. 4.** and the 'Treg cell depletion in *Aqp4*^{ΔTEC} mice' data in **Extended Data Fig. 6.**

3. The authors assert that thymic B cells tolerize against a group of germinal center-associated self-antigens in order to prevent inappropriate help by Tfh cells during GC reactions. This is a very attractive idea. However, the current study addresses only a single antigen, and disease is only shown in the context of immunization with the self-antigen. If the authors could provide more information about the potential scope of this process (how many CD40 dependent antigens are expressed in thymic B cells?) or show autoimmunity to Aqp4 can develop in situations besides Aqp4 immunization (e.g. with heterologous infections or chronic infection?) then the impact of the study would be much greater. [The abstract states that “thymic B cells emulate a CD40-stimulated B cell transcriptome”, but there are no data that directly address this. They also state “Aqp4 appears to be a paradigmatic antigen in this context”, but what evidence is this based on?]

We want to thank the reviewer for these comments. These are highly relevant questions. First, in order to address these points experimentally, we generated new RNA sequencing data from wild-type thymic IgM⁺IgD⁻ B cells and compared their transcriptome with thymic B cells from *Cd40*^{-/-} mice, in which B cells cannot respond to CD40L. The genes differentially expressed in wild-type thymic B cells as compared to *Cd40*^{-/-} thymic B cells comprise potential autoimmune target antigens. These new data are provided in the **revised Fig. 3i-k.**

Second, we tested the autoimmune proneness of a T cell repertoire that grew up in the absence of thymic B cells by adoptively transferring the mature T cell repertoire of control (wild-type) mice or B cell-deficient *Mb1-Cre*^{KI/KI} mice into *Tcra*^{-/-} host mice and immunized these animals with CFA (in the absence of sensitizing exogenous autoantigens). After 4 weeks, the serum of the host mice was tested for autoantibodies by screening sections of various organs (brain, kidney, skin) of *Rag1*^{-/-} mice. Notably, sera derived from *Tcra*^{-/-} mice loaded with T cells from B cell-deficient mice showed a higher degree of autoreactivity than sera from *Tcra*^{-/-} host mice transferred with T cells from wild-type mice. Of course, we cannot deduce the identity of the target-autoantigens from the staining pattern. However, potential autoantigens in multiple organ systems appear to be involved. We have added these new data to the **Extended Data Fig. 8 (panels e and f).**

4. As an animal model of neuromyelitis optica, it is intriguing that the authors state that retina infiltration was only observed in *Aqp4*ΔB mice immunized with P41, but not those immunized with MOG. But this data is not shown in Figure 6! This data should be included. Also, the authors should compare the retina infiltration in *Aqp4*^{-/-}, *Aqp4*ΔB, and WT mice (immunized with P41).

We apologize for this. In the revised version, we show side-by-side that P41-immunized but not MOG(35-55)-immunized *Aqp4*^{ΔB} mice acquire retinal infiltrates (**revised Fig. 6c**). P41-immunized *Aqp4*^{-/-} mice never developed any signs of disease since they lack the antigenic target (we refrained from showing this obvious phenomenon). In addition, we have previously reported the immunopathology of MOG(35-55) immunization in *Aqp4*^{-/-} mice as compared to MOG(35-55)-immunized wild-type mice [5], and we now cited this study in the discussion.

Other suggestions for improving the report:

5. Figure 3 shows that CD40 can upregulate Aqp4 in naïve B cells, but it would be more definitive to show that thymic B cells from CD40 deficient mice do not express Aqp4. It would be interesting to know if TEC

expression of *Aqp4* was also dependent on CD40 since *Aqp4* expression in TEC was Aire-independent (Figure 3i).

In the revised version of the manuscript, we now included an RNAseq experiment of wild-type vs. *Cd40*^{-/-} thymic B cells, confirming the lack of AQP4 expression in CD40-deficient thymic B cells (revised Fig. 3i-k, and **Extended Data Fig. 5c**). Conversely, as expected, TECs from *Cd40*^{-/-} mice express AQP4 to the same level as their wild-type counterparts (**revised Extended Data Fig. 5b**).

The original Fig. 3i, showed that AQP4 expression in CD40 plus IL-21 stimulated B cells is not dependent on Aire since Aire-deficient B cells upregulated AQP4 similarly to wild-type B cells in response to CD40. We have not investigated the Aire-dependence of AQP4 in TECs.

6. Figure 4 is problematic. Why are there no DP thymocytes present amongst retrogenic derived cells in *Aqp4*^{ΔB} mixed BMC, but they are present in WT mixed BMC? Also, the number of B cells per thymus in these chimeras is extremely low (Figure 4d). Perhaps the fact that radiation destroys the thymic niche for B cells makes this approach unsuitable for assessing the role of B cell derived antigens in tolerance. Was there any expression of Foxp3 in retrogenic CD4 SP?

In the revised version of this manuscript, we improved the technical quality of the mixed bone marrow chimera experiment reported in Fig. 4. In fact, we repeated the experiment and now treated the adjunct polyclonal bone marrow stem cells (derived from either wild-type or *Aqp4*^{ΔB} mice) like the *Rag1*^{-/-} bone marrow stem cells, i.e. we kept them both in culture (and did not use the polyclonal stem cells directly ex vivo) before transferring them into the irradiated *Aqp4*^{-/-} host mice. This improved protocol yielded very consistent results with a clear CD4⁺CD8⁺ double positive (DP) population in the TCR retrogenic compartment. We now provide these improved data in the **revised version of Fig. 4d-f**. We also decided to show the gating strategy of the thymus staining (including CD5 staining of the DP compartment) in the **Extended Data Fig. 7a-c**.

7. Since the *Aqp4*^{ΔTEC} and *Aqp4*^{ΔB} mice are critical for the conclusions made, it is imperative that the authors provide evidence of appropriate tissue specificity of the deletion of *Aqp4*, especially since both of these Cre lines can lead to germline deletion. Related to this issue about rigor, please state in the legend exactly what the “*Aqp4*^{ΔB}” mice are (presumably they are MB1Cre^{+/-}*Aqp4*^{fl/fl}) and describe the breeding strategy used.

As suggested, we confirmed the tissue-specific deletion of AQP4 on many levels. Most importantly, we sorted TECs and thymic B cells from wild-type, *Aqp4*^{-/-}, *Aqp4*^{ΔTEC}, and *Aqp4*^{ΔB} mice and tested them for *Aqp4* mRNA by quantitative PCR. These new data are shown in **Extended Data Fig. 2f**. Germline deletions were always excluded by genotyping controls. The exact genetic denomination of the strains used in this manuscript are now mentioned upon first appearance in the text: *Aqp4*^{ΔTEC} mice (*Foxn1*-Cre⁺; *Aqp4*^{fl/fl}), *Aqp4*^{ΔB} mice (*Mb1*-Cre^{Kl/wt}; *Aqp4*^{fl/fl}).

8. In figure 6, WT mice immunized with P41 show a reduced incidence of EAE when compared to MOG immunized *Aqp4*^{ΔB} mice. However, the clinical scoring of these WT mice is not shown in figure 6b. Is there a difference in clinical score, or is it only that fewer mice get sick, but at the same severity as P41-immunized *Aqp4*^{ΔB} mice? If this the case, how does this fit into the hypothesis presented?

Very few wild-type mice develop clinical signs of disease when immunized with P41 (See **Fig. 6a of the revised manuscript**). When a threshold for clinical disease is surpassed, the EAE score is often not a

good discriminator of the overall extent of immunopathology anymore, in particular, if the spinal cord is prominently affected, which is really the main determinant of the motor deficits of the mice. The complete clinical data are now provided in **Extended Data Table 2**.

9. In Figure 1f, I see little to no co-expression of CD20 and AQP4, is there further data to support the conclusion that human thymic B cells express AQP4 at steady state?

In the revised version of the manuscript, we provide new examples of human B cells from neonatal thymus samples co-expressing AQP4 and CD20, including confocal images, that clearly support the existence of thymic B cells expressing AQP4. See **Fig. 1f-i of the revised manuscript**.

10. In the methods, both AQP4 peptide and full-length protein are noted to be used for EAE induction. In Figure 6, EAE is only showed for P41 peptide induction. In MOG induced EAE, use of full-length protein vs peptide alters the APC requirements. Does full length AQP4 protein give different EAE than when using peptide in this conditional knock out system?

Thank you for this comment. It has taken many years to gather a mechanistic understanding of the involvement of different APC subsets in the immunopathology of MOG(35-55)-induced vs full-length MOG protein (or extracellular domain)-induced EAE (e.g. [6-9]). With the tools in hand, we will embark on this project in future studies for AQP4, but we cannot complete a thorough and meaningful investigation of this issue (comparing the immunopathology of P41 vs full-length AQP4-induced EAE) during the limited time of this revision process. Also, we hope that this reviewer agrees that such a study – irrespective of its clear value per se – will likely not provide further depth to the current manuscript.

Minor points

11. Material and methods needs to be completed. The mouse immunization protocol doesn't include the adjuvant. The tetramer staining protocol also does not include an enrichment step. If an enrichment step was not performed to identify tetramer positive cells in naïve mice, this would be concerning for the interpretation of naïve repertoire experiments.

We apologize for any insufficient information on the experimental procedures. We have now amended the Materials and Methods section where necessary and indicated the exact procedures in the Materials and Methods section for every experiment. (See also comments to reviewer #3, **including Figure L1 in this letter**.)

12. Panel 4a, please state how many weeks after reconstitution the animals were analyzed.

The animals were analyzed 6 weeks after grafting the mixed bone marrow. This information is now provided in the Materials and Methods section.

13. Panel 5b, please state whether the flow analysis of "Tfh" and "GC B cells" was from spleen or LN.

The data shown in **Fig. 5b** was obtained from spleen. We amended the figure legend accordingly.

14. Line 240 has a typographical error.

We are sorry for this. The typographical error (duplication) was corrected in the revised version of the manuscript.

Referee #2 (Remarks to the Author):

In this manuscript, Afzali et al demonstrate that thymic B cells are the primary source of AQP4 expression in the thymus and play a non-redundant role in the induction of T cell tolerance to this specific and clinically relevant self-antigen. The role of thymic B cells has remained quite fuzzy with some evidence for their possible role in tolerance induction, but this was thought to be rather redundant and inferior to mTEC. To the best of my knowledge, this is the first study that experimentally shows a clear and non-redundant role for thymic B cells in tolerance induction.

This is a great and very elegant study, which provides a very convincing and needed evidence for the rather enigmatic role of B cells in the thymus, by using AQP4 as a model self-antigen. The methodology used to address this question is appropriate and elegant - and is primarily based on tracking AQP4-reactive T cells using tetramers in various mouse models based either on reciprocal BM chimeras or B- or TEC specific knockouts of AQP4.

Moreover, the study has important clinical implications, as AQP4 is a well-established autoimmune target in humans and thus these discoveries could potentially be instrumental in designing therapeutic strategies for treatment of human autoimmune diseases, in which AQP4 is specifically targeted, including neuromyelitis optica.

The study is written very clearly with a logical flow and the statements are supported by the data.

While I would be happy to see this important study being published in journals like Nature in the future, I think the current version would benefit from further addressing the following points:

Thank you so much for this highly supportive assessment.

1) While the authors provide very convincing evidence that B cells are both necessary and sufficient to induce tolerance to AQP4 in the thymus, through elimination of AQP4-specific T cells and induction of AQP4-specific Treg, Fig 2a,b suggest that minor role may also be played by AQP4-expressing TEC. The reason for this notion is that TEC-specific deletion of AQP4 still results in generation of higher number of AQP4 specific T cells than observed for control mice. Similarly, Fig1a suggests that transfer of AQP4 KO BM into AQP4 KO recipients has bigger effect than AQP4 KO BM into AQP4 WT recipients (the authors should show the P val for this comparison).

This is a correct observation. While the p-value of the comparison of tetramer binding T cells in KO→WT BMCs vs. KO→KO BMCs was not significant ($P=0.0858$), we agree that AQP4 expression in TECs might have a (small and perhaps indirect) contribution in shaping the AQP-specific T cell repertoire. The best indication for this notion comes from the direct tetramer staining of CD4⁺ SP cells in the thymus of naive wild-type vs. *Aqp4*^{ΔTEC} mice that we now show in the revised version of the manuscript (**Fig. 4a, b**). In both genotypes, thymic B cells would be sufficient in AQP4 and lift the main load of negative selection. However, the fraction of AQP4-specific Foxp3⁺ SP cells is slightly larger in *Aqp4*^{ΔTEC} mice than in wild-type controls (**revised Fig. 4**), suggesting that AQP4-specific thymocytes are more readily diverted into the Treg cell lineage when B cells are the only cognate APCs as compared to a situation when both B cells and TECs

provide the antigen. The exact delineation of this observation is beyond the scope of this study and would require further models. Yet, this discovery does not take anything away from the concept of B cells as major negative selectors of AQP-specific thymocytes.

a. It would be very informative if the authors could provide more info (e.g. using publicly available scRNAseq datasets) which of the TEC compartments expresses AQP4? Is it expressed in Aire⁺ mTEC in an Aire-dependent manner or is it expressed by one of the identified TEC mimetics (e.g. the neuroTEC) in an Aire-independent manner?

Thank you for this suggestion. We re-analyzed publicly available scRNAseq data sets from human and mouse thymus [4, 10]. The number of AQP4-expressors in TEC subsets was exceedingly low. Nevertheless, we displayed the AQP4 and AIRE-expressors in the various annotated TEC subsets in **Extended Data Fig. 2a, b** of the revised manuscript.

While we could use these data for an estimate of the MHC class II expression level in different thymic B cell and TEC subsets (see **Extended Data Fig. 2c, d**), the sequencing depth was generally too low to analyze AQP4 expression in a meaningful manner. For the analysis of AQP4 expression in various B cell subsets (including thymic *Cd40*^{-/-} B cells), we had to perform bulk sequencing of sorted thymic B cells. These data that show that AQP4 is expressed in thymic IgM⁺IgD⁻ B cells but not in *Cd40*^{-/-} thymic B cells are now provided in the **revised Fig. 3** and **Extended Data Fig. 5c**.

b. Moreover, most of the TEC mimetics express low levels of MHCII, suggesting that they are poor antigen presenting cells and probably serve only as a passive source of AQP4. I think the study would benefit from providing this info, as it could possibly explain why the expression of AQP4 in TEC is redundant and in B cells it is not.

This is actually an excellent idea and was – to some degree – also asked by reviewer #1. While the exact interaction of the two relevant AQP4 sources that could potentially serve as APCs in the thymus (TECs and B cells) would require a more detailed follow-up study, the investigation of the MHC class II expression level on TECs vs. thymic B cells might already provide a framework to assess the relative significance of either cell type as thymic APC for particular antigens. Therefore, we investigated both human and mouse thymic B cells for the expression of MHC class II. We found that thymic B cells express MHC class II at much higher levels than any TEC subset as assessed by re-analysis of publicly available scRNAseq data (**Extended Data Fig. 2a-d**) and new flow cytometric analysis of MHC class II on TECs vs. thymic B cells (**Extended Data Fig. 2e**).

c. Correspondingly, if the AQP4 is expressed by one of the TEC mimetics, are thymic B cells somehow associated with these AQP4⁺ TEC?

Thymic B cells are strategically located at the cortico-medullary junction of the thymus – in a position where they might be among the first APCs to interact with CD4⁺CD8⁺ double positive (DP) thymocytes for initiating selection processes (see also **new Extended Data Fig. 1c**). While it is possible that thymic B cells also talk to TEC mimetics in a direct or indirect manner, our data suggest that B cells present their endogenous AQP4 and do not pick up the antigen from other donating cells. The evidence for this idea is based on the rescue of AQP4-specific thymocytes from negative selection when AQP4 is genetically ablated specifically in B cells. If thymic B cells were provided with AQP4 by TEC mimetics, genetic ablation of AQP4 in B cells should be irrelevant or perhaps be associated with only a minor phenotype.

Nevertheless, a possible interaction between thymic B cells and TEC mimetics needs to be further investigated. Technically, this is a problem that would call for a spatial transcriptomics approach (perhaps in combination with a scRNAseq data set). While we feel that we have performed the best possible experiments to support our conclusions, a spatial transcriptomics approach may not directly contribute to increasing the depth of our study but extend the topic to a different aspect, which – in our opinion – should be followed up in a future study.

2) The authors convincingly demonstrate that both CD40 and IL21 signaling induce AQP4 expression in B cells. This is very interesting and suggests that the contact with potentially self-reactive T cells provides the B cells with the necessary ligands (i.e. CD40L, IL21).

We agree with the reviewer and would strongly put forward such a hypothesis. See also the response to the next comment.

a. Could the authors show (e.g. using publicly available scRNAseq data for mouse/human thymus) what is the primary source of IL21 in the thymus? I assume it is the CD4T cell, but it would be beneficially to show that.

The sources of IL-21 in the thymus are indeed SP thymocytes. This has already convincingly been shown using an IL-21 reporter mouse [11]. We now cite this reference in the revised version of the manuscript and thank the reviewer for enabling us to add this context.

b. Does the CD40/IL21 signaling in thymic B cells induce expression of TRA genes other than AQP4? What other relevant genes (e.g. antigen presentation, cytokine, chemokine, etc.) are induced by this signaling? The study would benefit by providing this info in much more detailed manner.

This is an intriguing question, and we thank the reviewer for bringing it up. The most direct way to address this issue is a RNAseq analysis of wild-type thymic B cells as compared with *Cd40*^{-/-} thymic B cells. In order to acquire the most comprehensive picture of the CD40-dependent transcriptome of thymic B cells, we compared the thymic CD19⁺IgM⁺IgD⁻ B cell compartment of wild-type mice (which expresses the highest level of AQP4) with the thymic CD19⁺ compartment of CD40-deficient mice (that essentially lack the IgM⁺IgD⁻ B cell compartment in the thymus). Excitingly, the analysis of this new experiment provided – besides AQP4 – a list of further targets of known disease-relevant autoantibodies. These data provide a valuable resource. Whether tolerance to these autoantigens is (like in the case of AQP4) also "handled" by thymic B cells needs be determined in future studies. We now included these data in the **revised Fig. 3 (panels i-k)**.

3) Could the authors show the generation of AQP4-specific antibodies (either in ELISA to relevant peptides or WB from AQP4-expressing cells in non-immunized vs., immunized ctrl, or in mice B- or TEC specific deletion of AQP4?

Wild-type mice (immunized or non-immunized) do not contain anti-AQP4 antibodies (as determined by our cell-based assay). However, it is an intriguing question whether the autoimmunity-prone T cell repertoire of B cell-deficient mice or – in a more antigen-focused manner – the T cell repertoire of *Aqp4*^{AB} mice provide help for producing autoantibodies in a spontaneous manner or in the context of a heterologous immune stimulus. In order to test this, we transferred the mature T cell repertoire of wild-type mice or B cell-deficient mice (*Mb1-Cre*^{KI/KI}) into *Tcra*^{-/-} recipients and immunized them with CFA (without sensitization for an

autoantigen). We tested the sera of these mice for autoantibodies after 4 weeks and found more widespread antibody staining of *Rag1*^{-/-} tissue sections by the sera provided by the hosts that had received T cells educated in the absence of B cells as compared to T cells educated in the presence of thymic B cells. These results were analyzed in a qualitative manner by showing differentially stained tissue structures in CNS, kidney, and skin and are shown in the **revised Extended Data Fig. 8 (panels e and f)**.

4) Fig 6a, seems to be missing an important control – WT-MOG immunized mice as a baseline

This is the standard MOG(35-55) induced EAE. We added the incidence and severity data of this control group to the data of **Fig. 6a**. There is no difference in MOG(35-55)-induced EAE in wild-type vs. *Aqp4*^{ΔB} mice.

Referee #3 (Remarks to the Author):

In this study, Afzali et al focused on a self-antigen which acts as the target in the human autoimmune disease neuromyelitis optica (NMO) and investigated the mechanisms by which immune tolerance against that self-antigen is established. The author's data show that the self-antigen AQP4 expressed in thymic B cells is important for inducing negative selection of immature T cells reactive to AQP4-derived peptide and establishing immune tolerance to AQP4, contributing to prevent AQP4-related autoimmune disorders. AQP4 expression in CD40 stimulated B cells was observed in humans as well as in mice, suggesting that humans B cells also contribute to the self-antigen expression and tolerance induction and that failure of this tolerance mechanism might results in autoimmune disease. Although the finding that thymic B cells express self-antigens in the thymus to induce immune tolerance against such self-antigens is not so new and has been previously reported by the authors themselves, many of the experiments performed in the present paper are logically well connected to each other and the results appear clear. The authors emphasize that their present data strongly support the hypothesis that thymic B cells tolerize the T cell repertoire against B cell antigen to prevent inappropriate immune responses upon inappropriate interaction between Tfh cells and B cells during GC formation in the peripheral lymphoid tissues. How this mechanism breaks down in patients with NMO remains unclear and needs future studies.

This study is well designed and manuscript is well written. However, some concerns are raised about the experimental methods and data related to the key experiments in this study. The authors need to address the following concerns.

We would like to thank the reviewer for this overall positive assessment of our study.

#1 The method of P41-IAb tetramer staining is unclear. It is not stated whether the authors perform tetramer enrichment and how the tetramer+ cells were counted (by using counting beads?). These methods are essential to accurately measure the number of tetramer+ cells (Taniguchi et al, PNAS 2012). What does 'in sec LyTi (#)' mean in Fig.1a and Fig 2a,b,e?

We apologize for any lack of clarity in the description of the staining and counting method used. We have amended this in the revised version of the manuscript. See the revised 'Materials and Methods' section of the manuscript (sec LyTi; secondary lymphoid tissue, i.e. spleen plus all accessible peripheral lymph nodes).

For validating the absolute number of events in the tetramer⁺ gate, we have indeed spiked representative samples with counting beads and observed a recovery rate in the range of 90 percent (or better) (see Figure L1):

Counts:

- Cell counting beads only: 21028 events in counting beads gate (90% recovery)
- dLN: 1. 23348 (99.35 %) 2. 22560 (96 %)
- SPL: 1. 20481 (87 %) 2. 21272 (90.5 %)

Cell counting beads only
250 μ l FACS buffer
+ 50 μ l counting beads
0.47 * 10⁵ beads/ml
= 23.500 cells/50 μ l

Aqp4^{-/-} P41-imm. d10
250 μ l cell suspension
+ 50 μ l counting beads

Plotted as dot plots with 50.000 events

Figure L1. Quantification of absolute AQP4 P41/I-A^b+ T cell numbers. CountBright™ Absolute Counting Beads for flow cytometry (Thermo, # C36950) were added to the cell suspension immediately before acquisition following the manufacturer’s protocol (50 μ l beads in 250 μ l). The concentration of counting beads, as stated by the manufacturer (0.47 x 10⁵ beads/ml), was routinely verified through acquisition in FACS buffer (“empty control”). Representative cytograms of the forward and sideward scatters of the beads alone and the draining lymph nodes (dLN), as well as the lymphocyte gate of the dLN, spiked with counting beads and gated for P41/I-A^b double positive tetramer binding cells from a P41-immunized *Aqp4*^{-/-} mouse on day 10 after immunization.

#2 In Fig. 1a, the number of P41/IAb tetramer+ cells in KO -> WT BMC mice is nearly two order of magnitude lower than in KO -> KO BMC mice. This data may indicate limited contribution of hematopoietic cells to the negative selection of AQP4-specific CD4 T cells. Or is this due to the low contribution of KO hematopoietic cells in these BMC mice? If so, how could the data from WT -> KO mice be interpreted?

Thank you for pointing out this observation. We performed statistical analysis of the total number of tetramer-positive CD4⁺ T cells in the secondary lymphoid tissue of P41-immunized bone marrow chimeras, and in fact – due to some variation – the number of tetramer-binding T cells is not significantly different between KO→WT BMCs vs KO→KO BMCs (P=0.0858). However, both BMCs had significantly more tetramer-positive T cells than the WT→WT BMC controls. While we cannot exclude that the irradiation alters the thymus niche, a legitimate interpretation of these data is that the hematopoietic compartment, more than the radioresistant compartment of the host mice, contributes to the selection of AQP4-specific T cells. Of course, from this experiment alone, it is not possible to conclude that the shaping of this AQP4-specific T cell repertoire occurs in the thymus.

#3 In Fig. 1c, the mRNA expression level of AQP4 is 10-fold higher in TECs than in thymic B cells. According to the ImmGen or BioGPS databases, the expression AQP4 is not detected in both TECs and B cells. In addition, scRNAseq data in Fig. 3 shows very faint mRNA expression of AQP4 in thymic B cells (less than 1/1000th of astrocytes). It is puzzling that despite such low levels of mRNA expression, AQP4 protein expression is readily detected in the thymus by immunohistochemistry in Fig. 1f. Is AQP4 signal detected in CD20-positive B cells? It does not appear so, though. Of further important to point out in this immunohistochemistry data (Fig. 1f) is that the detection specificity of anti-AQP4 antibody is not demonstrated. The authors should show the immunohistostaining with the anti-AQP4 antibody of the thymus from AQP4-KO and B-cell-specific AQP4-cKO mice.

We want to thank the author for this relevant comment. In order to properly detect AQP4 expression in TECs (which other groups have reported as well [3], see also the reconstruction of the AQP4 status of various TEC subsets from publicly available scRNAseq data in **the revised Extended Data Fig. 2a,b**) and in thymic B cells (where we are the first to report this in this study), quantitative PCR or deep sequencing in a bulk RNAseq approach is required.

We agree that the quality of the immunostainings needs to be improved. Therefore, we reanalyzed the co-staining of CD20 and AQP4 in human neonatal thymus tissue, including confocal imaging (see **amended Fig. 1f-i**). In addition, as suggested by the reviewer, we performed co-staining of TECs (EpCAM), B cells (CD19), and AQP4 in the thymus of wild-type mice and *Aqp4*^{-/-} mice in order to have appropriate staining controls for AQP4. With this approach, we were able to confirm the co-expression of AQP4 and CD19 in thymic B cells. These new data are included in **Fig. 1e** and in **Extended Data Fig. 1c** of the revised manuscript.

#4 In Fig. 2a, the number of P41-specific CD4 T cells in non-immunized mice should be examined in the thymus as well. As described above, tetramer enrichment and counting beads are essential. Because the main issue of this paper is negative selection in the thymus, data of P41-specific CD4SP thymocytes would certainly reinforce the authors' conclusion. To further clarify the role of thymic B cells, the authors should examine P41-specific CD4SP thymocytes in Ighm-KO (muMT) or CD40-KO mice.

According to this recommendation, we assessed the total number of tetramer-positive CD4⁺ SP thymocytes in naive wild-type, *Aqp4*^{-/-} mice, *Aqp4*^{ΔTEC} mice, and *Aqp4*^{ΔB} mice and also measured the fraction of Foxp3⁺ cells in tetramer-positive thymocytes. These data largely reflect the number of tetramer-positive T cells in the peripheral immune compartment of naive mice and thus support the contribution of AQP4-expressing and presenting thymic B cells to negative selection.

Deleting AQP4 in B cells is the most direct way to assess the role of the expression of endogenous AQP4 in B cells for the negative selection of AQP4-specific thymocytes (detected by tetramer staining). However, we agree that further analysis of the thymic B cell biology would be required to grasp the wider perspective of thymic B cell-mediated negative selection or diversion of thymocytes into the Treg cell lineage. Therefore, we performed RNAseq experiments on wild-type vs. CD40-deficient thymic B cells (addressing especially the expression of AQP4 and other potential B cell-expressed autoantigens) and included these novel data in the **revised Fig. 3 (panel i-k)**.

#5 It is questionable whether experiments in Fig. 4 have been performed properly. The flow cytometry profile of AQP4 TCR retrogenic delta-B mice looks very strange; Why do the mice have CD8SP cells? Do the

CD4SP and CD8SP cells express AQP4-specific TCR? Why are DP thymocytes in the mice so few? Are the mice stressed, resulting in thymus hypoplasia due to a decrease of DP thymocytes?

The authors should show data on total thymocytes numbers and flow cytometry profiles for CD45.1 and CD45.2 to evaluate the chimerism of the BMC mice (as supplementary data). In addition, the frequency of CD4SP cells in WT BMC mice shown in Fig. 4b does not match the mean frequency of the same group of mice shown in Fig. 4c. A representative FCM profile should be shown.

In Fig. 4b, flow cytometry profiles of polyclonal (CD45.2) cells look very different between WT and delta-B mice. Are these data from the same experiment and from the same individual as the AQP4 TCR retrogenic (CD45.1/2) cells on the right panel?

This is an important point of critique that we took very seriously. Therefore, we repeated this experiment with an improved protocol and now provide the new data and the exact protocol in the revised version of the manuscript. Gating strategies and additional stainings for CD5 are now shown in **Extended Data Fig. 7 (panels a-c)**. Essentially, we confirmed that the TCR retrogenic (AQP4-specific) CD4⁺ SP compartment is significantly constricted in the presence of AQP4-sufficient B cells in the thymus.

#6 Fig. 3g shows that spleen B cells expressing AQP4 can stimulate T cells with AQP4-specific TCR *in vitro*, and the authors stated in the abstract 'the negative selection of AQP4-specific thymocytes is entirely dependent on the expression and presentation of AQP4 by thymic B cells'. However, this might be overstatement. Is there any data showing the AQP4 antigen presentation by B cells *in vivo*? At least, the authors should show the data that thymic B cells isolated from WT mice can stimulate AQP4-reactive T cells *in vitro* but those from AQP4-KO or delta-B mice cannot.

We want to thank the reviewer for these comments. We performed exactly the experiment suggested by the reviewer: We isolated thymic B cells (IgM⁺IgD⁻) from wild-type mice and used them directly *ex vivo* (without additional anti-CD40 stimulation) in a co-culture assay with our clone 6-expressing A5 hybridoma. Indeed, while *ex vivo* isolated IgM⁺IgD⁻ thymic B cells from AQP4 KO mice failed to do so, thymic wild-type B cells stimulated the hybridoma. In contrast, *ex vivo* isolated wild-type splenic B cells did not stimulate the hybridoma either unless they were pre-stimulated with anti-CD40 and IL-21. These data indicate that thymic B cells (after licensing *in vivo*, as entailed by their IgD⁻ status) are not only able to express AQP4 but also present AQP4 in the context of MHC class II for cognate interaction with T cells. We included these new data in the **revised Fig. 3 (new panel 3h)**.

#7 Related to #6, there should be more discussion (or data if possible) on how thymic B cells present antigen peptides derived from their own proteins with MHC class II. Loading of MHC class II by endogenous membrane proteins (shown in Ref. 18) is a phenomenon observed during viral infection or in the absence of usual antigen presentation pathway, so it is not clear whether it also works in thymic B cells. As reported previously (Nedjic et al, Nature 2008), do thymic B cells have the high autophagy activity, similar to TECs and DCs, and degrade ER-targeted membrane proteins to load MHC class II?

These are all highly appropriate and intriguing questions. Extracellular epitopes of transmembrane proteins like AQP4 and MHC class II molecules may be sorted into subcellular compartments where loading of MHC class II is, in principle, possible. For instance, it has been described for other transmembrane proteins, e.g. for OVA fused to the transferrin receptor, that they can have access to the endosomal compartment and MHC class II loading [12, 13]. In fact, the "haplodome" of HLA class II molecules of an EBV-transformed B-lymphoblastoid cell line comprised a prominent fraction of plasma membrane protein-derived peptides [14]. We now refer to this literature in the revised discussion of our manuscript. However, an in-depth investigation of this process is a new (cell biological) project that would require extensive experimentation.

Here, we refer to the principle possibility of such a class II loading and cite appropriate literature from other laboratories, but we would argue that the detailed description of class II loading of AQP4 epitopes is beyond the scope of our present study.

Minor comments:

#8 The legend of Fig. 1c describes 'medullary thymic epithelial cells (mTECs, live CD45–EpCAM+)', but this is wrong. CD45–EpCAM+ cells are just thymic epithelial cells (TECs), not mTECs.

We apologize for this inaccuracy and have amended this in the legend to Fig. 1 in the revised version of the manuscript.

#9 In Fig. 1f, what is blue-colored? DNA stained with DAPI?

Correct. The blue color is DAPI. In the revised version of the manuscript, we show improved double staining for CD20 and AQP4 in human neonatal thymus tissue, including confocal imaging (**revised Fig. 1f-i**).

#10 In Fig. 6, why are the N numbers different between panels a and b? In Fig. 6b, the data of MOG- or P41immunized WT mice should be shown and compared.

As stated in the figure legend, in the incidence panel (**Fig. 6a**), all mice are shown, including the mice that did not develop clinical signs of disease, whereas in **Fig. 6b**, only the sick mice are shown in each group in order to assess the severity of EAE in those individuals that surpassed the threshold of manifest clinical disease. The entire data set is now provided in **Extended Data Table 2**.

References

1. Yamano T, Nedjic J, Hinterberger M, Steinert M, Koser S, Pinto S, Gerdes N, Lutgens E, Ishimaru N, Busslinger M, et al. (2015) Thymic B Cells Are Licensed to Present Self Antigens for Central T Cell Tolerance Induction. *Immunity* 42: 1048-1061. DOI 10.1016/j.immuni.2015.05.013
2. Perera J, Meng L, Meng F, Huang H (2013) Autoreactive thymic B cells are efficient antigen-presenting cells of cognate self-antigens for T cell negative selection. *Proc Natl Acad Sci U S A* 110: 17011-17016. DOI 10.1073/pnas.1313001110
3. Michelson DA, Hase K, Kaisho T, Benoist C, Mathis D (2022) Thymic epithelial cells co-opt lineage-defining transcription factors to eliminate autoreactive T cells. *Cell* 185: 2542-2558 e2518. DOI 10.1016/j.cell.2022.05.018
4. Park JE, Botting RA, Dominguez Conde C, Popescu DM, Lavaert M, Kunz DJ, Goh I, Stephenson E, Ragazzini R, Tuck E, et al. (2020) A cell atlas of human thymic development defines T cell repertoire formation. *Science* 367. DOI 10.1126/science.aay3224
5. Maisam Afzali A, Stuve L, Pfaller M, Aly L, Steiger K, Knier B, Korn T (2022) Aquaporin-4 prevents exaggerated astrocytosis and structural damage in retinal inflammation. *Journal of molecular medicine* 100: 933-946. DOI 10.1007/s00109-022-02202-6
6. Hjelmstrom P, Juedes AE, Fjell J, Ruddle NH (1998) B-cell-deficient mice develop experimental allergic encephalomyelitis with demyelination after myelin oligodendrocyte glycoprotein sensitization. *Journal of immunology* 161: 4480-4483
7. Oliver AR, Lyon GM, Ruddle NH (2003) Rat and human myelin oligodendrocyte glycoproteins induce experimental autoimmune encephalomyelitis by different mechanisms in C57BL/6 mice. *Journal of immunology* 171: 462-468. DOI 10.4049/jimmunol.171.1.462
8. Molnarfi N, Schulze-Topphoff U, Weber MS, Patarroyo JC, Prod'homme T, Varrin-Doyer M, Shetty A, Linington C, Slavin AJ, Hidalgo J, et al. (2013) MHC class II-dependent B cell APC function is required for induction of CNS autoimmunity independent of myelin-specific antibodies. *J Exp Med* 210: 2921-2937. DOI 10.1084/jem.20130699
9. Flach AC, Litke T, Strauss J, Haberl M, Gomez CC, Reindl M, Saiz A, Fehling HJ, Wienands J, Odoardi F, et al. (2016) Autoantibody-boosted T-cell reactivation in the target organ triggers manifestation of autoimmune CNS disease. *Proc Natl Acad Sci U S A* 113: 3323-3328. DOI 10.1073/pnas.1519608113
10. Kernfeld EM, Genga RMJ, Neherin K, Magaletta ME, Xu P, Maehr R (2018) A Single-Cell Transcriptomic Atlas of Thymus Organogenesis Resolves Cell Types and Developmental Maturation. *Immunity* 48: 1258-1270 e1256. DOI 10.1016/j.immuni.2018.04.015
11. Marnik EA, Wang X, Sproule TJ, Park G, Christianson GJ, Lane-Reticker SK, Jain S, Duffy T, Wang H, Carter GW, et al. (2017) Precocious Interleukin 21 Expression in Naive Mice Identifies a Natural Helper Cell Population in Autoimmune Disease. *Cell Rep* 21: 208-221. DOI 10.1016/j.celrep.2017.09.036
12. Blum JS, Wearsch PA, Cresswell P (2013) Pathways of antigen processing. *Annu Rev Immunol* 31: 443-473. DOI 10.1146/annurev-immunol-032712-095910
13. Diebold SS, Cotten M, Koch N, Zenke M (2001) MHC class II presentation of endogenously expressed antigens by transfected dendritic cells. *Gene Ther* 8: 487-493. DOI 10.1038/sj.gt.3301433
14. Ramarathinam SH, Ho BK, Dudek NL, Purcell AW (2021) HLA class II immunopeptidomics reveals that co-inherited HLA-allotypes within an extended haplotype can improve proteome coverage for immunosurveillance. *Proteomics* 21: e2000160. DOI 10.1002/pmic.202000160

Reviewer Reports on the First Revision:

Referees' comments:

Referee #1 (Remarks to the Author):

The underlying hypothesis in this paper, that thymic B cells are required to prevent tolerance against antigens generated during B cell activation (in this case Aqp4), is novel and interesting. The authors added a number of important experiments that addressed critical points raised by the reviewers, including a list of genes whose expression in thymic B cells was dependent on CD40.

Nonetheless, the data do not show that B cells are the ONLY cells that can perform this function. The manuscript tends to skirt the issue that there is an obvious contribution of mTEC to tolerance of the P41-specific CD4 T cell repertoire, albeit a minor one. This over-interpretation of some of the data needs to be corrected. I suggest making the following edits to allow for a more conservative and precise interpretation of this experiments, while not detracting from the main point—that thymic B cells are playing an important role as tolerizing antigen presenting cells.

Line 106

Hematopoietic cells are responsible for the negative selection of AQP4-specific T cells

Change to

Hematopoietic cells are responsible for the drive negative selection of AQP4-specific T cells

Line 119-121

Notably, Aqp4^{-/-} → wild-type chimeras raised a sizeable fraction of P41/I-Ab⁺ T cells (Fig. 1a), suggesting that hematopoietic cells were involved in the negative selection of AQP4-specific T cells.

Add the additional sentence...

Notably, Aqp4^{-/-} → wild-type chimeras raised a sizeable fraction of P41/I-Ab⁺ T cells (Fig. 1a), suggesting that hematopoietic cells were involved in the negative selection of AQP4-specific T cells. We note that this population is not as large as that in Aqp4^{-/-} → Aqp4^{-/-} bone marrow chimeras, suggesting that both hematopoietic and non-hematopoietic cells contribute to negative selection.

Line 139-141

Similarly, elevated numbers of AQP4-specific T cells were detected in B cell-conditional AQP4-deficient mice but not in Aqp4dTEC mice (Fig. 2a).

Change to

Similarly, elevated numbers of AQP4-specific T cells were detected in B cell-conditional AQP4-deficient mice but not in Aqp4dTEC mice (Fig. 2a). AQP4-specific T cell numbers were slightly elevated in Aqp4 Δ TEC mice, but not to as great a degree as in Aqp4 Δ B mice

Line 141-143

Upon immunization with P41, AQP4-specific T cells were expanded both in Aqp4^{-/-} and in Aqp4 Δ B mice but failed to expand in Aqp4 Δ TEC mice (Fig. 2b).

Change to

Upon immunization with P41, AQP4-specific T cells were expanded in both Aqp4^{-/-} and in Aqp4 Δ B mice, but to a lesser extent in Aqp4 Δ TEC mice (Fig. 2b).

Line 147-149

Indeed, the fraction of Foxp3⁺ Treg cells was higher among P41/I-Ab⁺ T cells in Aqp4 Δ TEC than in Aqp4 Δ B mice and resembled the fraction of Foxp3⁺ in wild- type AQP4-specific T cells (Fig. 2c and Extended Data Fig. 3a). However, about 50 percent of ...

Add

Indeed, the fraction of Foxp3⁺ Treg cells was higher among P41/I-Ab⁺ T cells in Aqp4 Δ TEC than in Aqp4 Δ B mice and resembled the fraction of Foxp3⁺ in wild- type AQP4-specific T cells (Fig. 2c and Extended Data Fig. 3a). This suggests that B cell expression of AQP4 favors the development of self-antigen specific Treg cells. Furthermore, about 50 percent of...

Line 284-285

A failure of this mechanism should therefore allow for AQP4-specific T–B interactions since AQP4-specific Tfh cells would not be purged from the T cell repertoire.

Change to

A failure of this mechanism should might therefore allow for AQP4-specific T–B interactions since AQP4-specific Tfh cells would not be purged from the T cell repertoire as effectively by mTEC alone.

Line 294-295

these data indicate that AQP4 has a very specific status as a paradigmatic autoantigen specifically tolerized by thymic B cells.

Change to

these data indicate that AQP4 has a very specific status as a paradigmatic autoantigen specifically tolerized by that requires thymic B cells for complete tolerance.

Other points:

1) A recent paper by Sagan et al. (PMID: 37463205) compared the role of mTEC and AIRE in generating central tolerance to two Aqp4-specific CD4 T cell populations. A similar epitope identified in the Sagan paper (amino acid 202-218) and the epitope studied here (amino acids 205-215) both showed tolerance is primarily independent of Aqp4 expression in thymic epithelial cells. What that study further noted is that a different Aqp4 epitope (amino acid 133-149) was dependent on thymic epithelial cells for tolerance. While this paper is cited (citation 31), it would be good to specifically discuss its findings, and note that one epitope of Aqp4 is more dependent on epithelial cells for tolerance, and the other tends to be more dependent on B cells.

2) The authors, on more than one occasion, referred to the CD40-dependent gene list as containing “a number of disease-associated autoantigens besides AQP4”. These should be designated and references provided, as I could not discern which autoantigens they were referring to.

Referee #2 (Remarks to the Author):

the authors have substantially improved the manuscript and addressed most of my suggestions to my satisfaction.

Referee #3 (Remarks to the Author):

The authors have appropriately addressed my points #1, 6, 7, 8, 9, and 10.

#2

Even though not statistically significant ($P=0.0858$), there are a marked difference in the number of P41/IAb tetramer+ cells between KO->WT BMC mice and KO->KO BMC mice (Fig. 1a). Indeed, the lowest number in KO->KO BMC group is more than 2 times greater than the highest number in KO->WT BMC group. These data strongly suggest a significant contribution of radioresistant, non-hematopoietic cells in deletion of AQP4-specific T cells. Along with the argument in #3, the authors' interpretation that B cells are the main source of AQP4 and antigen-presenting cells for inducing tolerance to AQP4 in the thymus does not seem reasonable.

#3

The data in Fig. 1e indicate that the anti-AQP4 antibody used in this study is AQP4-specific and some of the AQP4+ cells are CD19+ B cells. At the same time, however, these data clearly show that AQP4+ CD19- (non-B) cells are abundant within the thymus. In WT thymus, the majority of AQP4-expressing cells (red in the image) appear to be non-hematopoietic cells with a stromal cell-like shape, and in addition, the majority of CD19+ B cells (green in the image) are AQP4-negative. This is a very important point for the argument of this paper. As pointed out in my previous review comment, the authors should perform the same experiments using B-cell-specific AQP4-cKO mice.

#4

I recommended the authors to examine whether deletion of P41-specific CD4SP thymocytes is affected in B-cell-deficient mice or CD40-KO mice. However, the authors showed the gene expression (including

AQP4) in thymic B cells from CD40-KO mice (Fig. 3i-j). That is not an answer to my argument. My point here is whether and how thymic B cells contribute to deletion of AQP4-specific CD4SP thymocytes. I found that Extended Data Fig. 5d shows that the number of AQP4-specific CD4 T cells after immunization is comparable between AQP4-KO and CD40-KO mice, and based on this data, the authors described that CD40-licensed thymic B cells are essential for deletion of AQP4-specific T cells. However, a careful look at the data shows that the number of P41/IAb tetramer+ cells in AQP4-KO and CD40-KO mice is around 10^2 , far below that in AQP4-KO mice shown in Fig. 2b. These data are inconsistent. Again, to clarify the role of thymic B cells in thymic deletion, the authors should examine P41-specific CD4SP thymocytes in naïve WT and CD40-KO mice (as for AQP4-KO mice in Fig. 4a).

#5

I appreciate that the authors provided the new data with an improved protocol. However, the concerns I noted in my previous peer-review comment remain unaddressed. Why do the delta-B BMC mice have so many AQP4-TCR transgenic CD8SP cells (Fig. 4d)?

The authors present new FCM data for TCRb vs CD5 in DP thymocytes, suggesting that agonist-mediated deletion occurs at the stage of DP thymocytes (line 270-273). However, Extended Data Figure 7b shows that CD5 expression in TCR retrogenic DP thymocytes with WT B cells is extremely lower than in polyclonal DP thymocytes (Extended Data Figure 7c shows 70% reduction, quite unlikely). Furthermore, in the presence of WT B cells, retrogenic TCRb is barely expressed in DP thymocytes. These data suggest that not only agonist-mediated deletion but also DP cell differentiation may be impaired. Given the very small number of B cells – at most 450 cells per thymus (as shown in Fig. 4f), it is extremely unnatural for such large differences in the status of DP thymocytes to occur. I do not think the authors' interpretation is valid.

Author Rebuttals to First Revision:

Dear Editor

Thank you for allowing us to re-revise our manuscript "B cells control autoimmunity against AQP4 by negative selection of antigen-specific thymocytes". We appreciate the reviewers' additional comments. As discussed, we performed additional experiments to further address the critique of Reviewer #3. In response to the remaining points of Reviewer #1, we toned our interpretation as suggested.

Please find a detailed point-by-point response in this letter:

Referee #1 (Remarks to the Author):

The underlying hypothesis in this paper, that thymic B cells are required to prevent tolerance against antigens generated during B cell activation (in this case Aqp4), is novel and interesting. The authors added a number of important experiments that addressed critical points raised by the reviewers, including a list of genes whose expression in thymic B cells was dependent on CD40.

Nonetheless, the data do not show that B cells are the ONLY cells that can perform this function. The manuscript tends to skirt the issue that there is an obvious contribution of mTEC to tolerance of the P41-specific CD4 T cell repertoire, albeit a minor one. This over-interpretation of some of the data needs to be corrected. I suggest making the following edits to allow for a more conservative and precise interpretation of this experiments, while not detracting from the main point—that thymic B cells are playing an important role as tolerizing antigen presenting cells.

Thank you for these comments. We appreciate the reviewer's point of view and agree with his/her interpretation. We are also grateful for the detailed suggestions made for improving the text. See below.

Line 106

Hematopoietic cells are responsible for the negative selection of AQP4-specific T cells
Change to
Hematopoietic cells are responsible for the drive negative selection of AQP4-specific T cells

Done.

Line 119-121

Notably, Aqp4^{-/-} wild-type chimeras raised a sizeable fraction of P41/I-Ab⁺ T cells (Fig. 1a), suggesting that hematopoietic cells were involved in the negative selection of AQP4-specific T cells.

Add the additional sentence...: Notably, Aqp4^{-/-} wild-type chimeras raised a sizeable fraction of P41/I-Ab⁺ T cells (Fig. 1a), suggesting that hematopoietic cells were involved in the negative selection of AQP4-specific T cells. We note that this population is not as large as that in Aqp4^{-/-} Aqp4^{-/-} bone marrow chimeras, suggesting that both hematopoietic and non-hematopoietic cells contribute to negative selection.

Done.

Line 139-141

Similarly, elevated numbers of AQP4-specific T cells were detected in B cell-conditional AQP4 deficient mice but not in Aqp4dTEC mice (Fig. 2a).

Change to Similarly, elevated numbers of AQP4-specific T cells were detected in B cell-conditional AQP4deficient mice but not in Aqp4dTEC mice (Fig. 2a). AQP4-specific T cell numbers were slightly elevated in Aqp4 TEC mice, but not to as great a degree as in Aqp4 B mice

Done.

Line 141-143

Upon immunization with P41, AQP4-specific T cells were expanded both in Aqp4^{-/-} and in Aqp4 B mice but failed to expand in Aqp4 TEC mice (Fig. 2b).

Change to Upon immunization with P41, AQP4-specific T cells were expanded in both Aqp4^{-/-} and in Aqp4 B mice, but to a lesser extent in Aqp4 TEC mice (Fig. 2b).

Done.

Line 147-149

Indeed, the fraction of Foxp3⁺ Treg cells was higher among P41/I-Ab⁺ T cells in Aqp4 TEC than in Aqp4 B mice and resembled the fraction of Foxp3⁺ in wild-type AQP4-specific T cells (Fig. 2c and Extended Data Fig. 3a). However, about 50 percent of ...

Add Indeed, the fraction of Foxp3⁺ Treg cells was higher among P41/I-Ab⁺ T cells in Aqp4 TEC than in Aqp4 B mice and resembled the fraction of Foxp3⁺ in wild-type AQP4-specific T cells (Fig. 2c and Extended Data Fig. 3a). This suggests that B cell expression of AQP4 favors the development of self-antigen specific Treg cells. Furthermore, about 50 percent of...

Done.

Line 284-285

A failure of this mechanism should therefore allow for AQP4-specific T-B interactions since AQP4-specific Tfh cells would not be purged from the T cell repertoire.

Change to A failure of this mechanism should might therefore allow for AQP4-specific T-B interactions since AQP4-specific Tfh cells would not be purged from the T cell repertoire as effectively by mTEC alone.

Done.

Line 294-295

these data indicate that AQP4 has a very specific status as a paradigmatic autoantigen specifically tolerized by thymic B cells.

Change to these data indicate that AQP4 has a very specific status as a paradigmatic autoantigen specifically tolerized by that requires thymic B cells for complete tolerance.

Done.

Other points:

1) A recent paper by Sagan et al. (PMID: 37463205) compared the role of mTEC and AIRE in generating central tolerance to two Aqp4-specific CD4 T cell populations. A similar epitope identified in the Sagan paper (amino acid 202-218) and the epitope studied here (amino acids 205-215) both showed tolerance is primarily independent of Aqp4 expression in thymic epithelial cells. What that study further noted is that a different Aqp4 epitope (amino acid 133-149) was dependent on thymic epithelial cells for tolerance. While this paper is cited (citation 31), it would be good to specifically discuss its findings, and note that one epitope of Aqp4 is more dependent on epithelial cells for tolerance, and the other tends to be more dependent on B cells.

Thank you for this comment. We have added in the revised discussion that our data support the notion that B cells are key contributors to negative selection, specifically for the AQP4(201-220) epitope. However, we would rather not like to further discuss the AQP4(133-149) epitope since we do not believe that this is a naturally processed epitope. We have extensively worked with the AQP4 protein and are, at the moment, one of the few labs that can express and purify the full-length mouse AQP4 protein. When we use the full-length AQP4 protein to immunize (non-tolerant) *Aqp4*^{-/-} mice and recall with overlapping peptides (20-mers overlapping by 15 AA covering the entire AQP4 sequence), we cannot recall a response to AQP4(133-149) (see also our previous paper (Vogel et al., 2017)). In fact, the AQP4(133-149) epitope was predicted in silico based on optimal binding to the I-Ab complex (Sagan et al., 2016). Therefore, we are unsure about the pathophysiologic relevance of AQP4(133-149). For these reasons, we would rather not embark on a detailed discussion of this problem in the current context (which will not benefit from such a discussion at this point).

2) The authors, on more than one occasion, referred to the CD40-dependent gene list as containing “a number of disease-associated autoantigens besides AQP4”. These should be designated and references provided, as I could not discern which autoantigens they were referring to.

We apologize for not having provided those references in the first place. In the revised discussion, we have added references and extended the discussion as to the selection of autoantigens that are likely tolerized by B cells in a CD40-dependent manner (page 18 of the revised manuscript).

Referee #3 (Remarks to the Author):

The authors have appropriately addressed my points #1, 6, 7, 8, 9, and 10.

Thank you for appreciating our responses to these prior points of critique.

#2

Even though not statistically significant ($P=0.0858$), there are a marked difference in the number of P41/IAb tetramer+ cells between KO->WT BMC mice and KO->KO BMC mice (Fig. 1a). Indeed, the lowest number in KO->KO BMC group is more than 2 times greater than the highest number in KO->WT BMC group. These data strongly suggest a significant contribution of radioresistant, non-hematopoietic cells in deletion of AQP4-specific T cells. Along with the argument in #3, the authors' interpretation that B cells are the main

source of AQP4 and antigen presenting cells for inducing tolerance to AQP4 in the thymus does not seem reasonable.

Our point is that thymic B cells, by presenting their endogenous AQP4, make a major contribution to the negative selection of AQP4-specific T cells. We provide a series of experiments to support that claim. In particular, genetic ablation of *Aqp4* in TECs largely fails to rescue the fraction and absolute number of AQP4-specific (tetramer-binding) to the level observed in *Aqp4*^{-/-} mice, while genetic ablation of *Aqp4* in B cells largely (albeit partially) rescues AQP4-specific T cells. Having said that, we cannot (and do not) exclude that TECs also contribute to the shaping of the AQP4-specific T cell repertoire. For instance, when TECs are deficient in *Aqp4* (in the presence of AQP4-sufficient B cells, i.e. in *Aqp4*^{ΔTEC} mice), the fraction of AQP4-specific Treg cells is increased (as compared to wild type), suggesting that the deletion vs diversion into the Treg cell compartment of AQP4-specific thymocytes is modulated by TECs.

We appreciate this point and have reported this (already during the first round of revision). In summary, while we stand by our claim that thymic B cells are key negative selectors of AQP4-specific T cells, we toned our statements according to the above-mentioned aspects (that were also raised by reviewer #1).

#3

The data in Fig. 1e indicate that the anti-AQP4 antibody used in this study is AQP4-specific and some of the AQP4+ cells are CD19+ B cells. At the same time, however, these data clearly show that AQP4+ CD19- (non-B) cells are abundant within the thymus. In WT thymus, the majority of AQP4-expressing cells (red in the image) appear to be non-hematopoietic cells with a stromal cell-like shape, and in addition, the majority of CD19+ B cells (green in the image) are AQP4negative. This is a very important point for the argument of this paper. As pointed out in my previous review comment, the authors should perform the same experiments using B-cell-specific AQP4-cKO mice.

As suggested, we now also performed triple staining (AQP4, CD19, and EpCAM) on thymus tissue of naive *Aqp4*^{ΔB} mice in addition to the stainings on wild-type and *Aqp4*^{-/-} mice that we had already presented in the first round of revision (Fig. 1e).

In the thymus of naive *Aqp4*^{ΔB} mice, AQP4⁺CD19⁺ B cells were absent as expected. As in wild-type mice (and as also indicated by the re-analyzed single-cell data sets (see Extended Data Fig. 2), there is also some overlap of EpCAM and AQP4 in *Aqp4*^{ΔB} mice. However, many AQP4⁺ cells in the thymus that are not B cells do not have a stromal cell-like shape but are likely thymocytes. In summary, the architecture of AQP4 expression in the EpCAM⁺ compartment in the thymus of wild-type and *Aqp4*^{ΔB} mice is similar. We have now included these data in the revised Fig. 1e.

#4

I recommended the authors to examine whether deletion of P41-specific CD4SP thymocytes is affected in B-cell-deficient mice or CD40-KO mice. However, the authors showed the gene expression (including AQP4) in thymic B cells from CD40-KO mice (Fig. 3i-j). That is not an answer to my argument. My point here is whether and how thymic B cells contribute to deletion of AQP4-specific CD4SP thymocytes.

We appreciate this recommendation and have decided to perform this experiment: We measured the frequency and absolute number of AQP4-specific (i.e. tetramer-binding) CD4⁺ SP thymocytes in unmanipulated B cell-deficient mice (*Mb1-Cre*^{KI/KI}) and in *Cd40*^{-/-} mice in addition to the data that we had already presented in our R1-revised Fig. 4a and b (i.e. AQP4-specific CD4⁺ SP thymocytes in wild-type, *Aqp4*^{-/-}, *Aqp4*^{ΔTEC}, and *Aqp4*^{ΔB} mice. Indeed, B-deficient mice and *Cd40*^{-/-} mice showed frequencies and

absolute numbers of AQP4-specific CD4⁺ SP thymocytes that were similar to *Aqp4*^{ΔB} mice, supporting the idea that thymic B cells (licensed by CD40-activation) were important negative selectors of AQP4-specific thymocytes. These new data are now integrated in the revised Fig. 4a and b and described in the revised manuscript (pp. 11 and 12).

I found that Extended Data Fig. 5d shows that the number of AQP4-specific CD4 T cells after immunization is comparable between AQP4-KO and CD40-KO mice, and based on this data, the authors described that CD40-licensed thymic B cells are essential for deletion of AQP4-specific T cells. However, a careful look at the data shows that the number of P41/IAb tetramer+ cells in AQP4-KO and CD40-KO mice is around 10², far below that in AQP4-KO mice shown in Fig. 2b. These data are inconsistent. Again, to clarify the role of thymic B cells in thymic deletion, the authors should examine P41-specific CD4SP thymocytes in naïve WT and CD40-KO mice (as for AQP4-KO mice in Fig. 4a).

We feel that our statement – based on the data shown in Extended Data Fig. 5d of the R1 version of our manuscript – is valid. In fact, the expansion of AQP4-specific CD4⁺ T cells upon immunization with P41 was similar in CD40-deficient mice as in AQP4-deficient mice, suggesting that CD40-licensed thymic B cells might control the AQP4-specific precursor frequency. A certain variability of the absolute numbers of AQP4-specific T cells is due to the variability of expansion observed after immunization (as compared to the data in Fig. 2). Yet, we appreciate this referee's concern and thus performed the experiment as outlined above: In that new experiment, we measured the AQP4-specific CD4⁺ SP in the thymus of unmanipulated (no immunization) B cell-deficient and *Cd40*^{-/-} mice in comparison to wild-type mice and global *Aqp4*^{-/-} mice as well as *Aqp4*^{ΔB} mice, illustrating that CD40-licensed thymic B cells substantially contribute to the precursor frequency of AQP4-specific T cells in the thymus. These data are now included in the revised Fig. 4a and b.

#5

I appreciate that the authors provided the new data with an improved protocol. However, the concerns I noted in my previous peer-review comment remain unaddressed. Why do the delta-B BMC mice have so many AQP4-TCR transgenic CD8SP cells (Fig. 4d)?

Thank you for this comment. It is actually not uncommon that a fraction of MHC class II-restricted TCRs express CD8 (instead of CD4) in a retrogenic setting. We have seen this for quite a few MHC class-II restricted TCRs (personal communication Ludger Klein). And also in the literature, this phenomenon has been reported (Alli et al., 2008; Holst et al., 2006). Clearly, the majority of the thymocytes expressing clone 6 in our mixed retrogenic bone marrow approach are sorted into the CD4 compartment in the absence (but not in the presence) of AQP4-sufficient B cells.

The authors present new FCM data for TCRb vs CD5 in DP thymocytes, suggesting that agonist-mediated deletion occurs at the stage of DP thymocytes (line 270-273). However, Extended Data Figure 7b shows that CD5 expression in TCR retrogenic DP thymocytes with WT B cells is extremely lower than in polyclonal DP thymocytes (Extended Data Figure 7c shows 70% reduction, quite unlikely). Furthermore, in the presence of WT B cells, retrogenic TCRb is barely expressed in DP thymocytes. These data suggest that not only agonist-mediated deletion but also DP cell differentiation may be impaired. Given the very small number of B cells – at most 450 cells per thymus (as shown in Fig. 4f), it is extremely unnatural for such large differences in the status of DP thymocytes to occur. I do not think the authors' interpretation is valid.

We clearly noticed differences in CD5 expression in the DP compartment depending on the presence or absence of AQP4-sufficient B cells (as the only independent variable). Having said that, we have not undertaken an in-depth analysis of the development and fate of the thymic DP compartment. While we feel that our conclusions are solid, we will not argue that the system that we use is artificial and that (in particular) the absolute number of recovered B cells may be underestimated due to B cell frailty during the preparatory process. Therefore, we have toned our statements as to the interpretation of the mixed bone marrow retrogenic mice in the revised version of the manuscript, taking into account these points.

Finally, we would like to thank the reviewers again for the time and effort they spent with our manuscript, which clearly helped us to improve this study.

References

- Alli, R., P. Nguyen, and T.L. Geiger. 2008. Retrogenic modeling of experimental allergic encephalomyelitis associates T cell frequency but not TCR functional affinity with pathogenicity. *Journal of immunology* 181:136-145.
- Holst, J., K.M. Vignali, A.R. Burton, and D.A. Vignali. 2006. Rapid analysis of T-cell selection in vivo using T cell-receptor retrogenic mice. *Nat Methods* 3:191-197.
- Sagan, S.A., R.C. Winger, A. Cruz-Herranz, P.A. Nelson, S. Hagberg, C.N. Miller, C.M. Spencer, P.P. Ho, J.L. Bennett, M. Levy, M.H. Levin, A.S. Verkman, L. Steinman, A.J. Green, M.S. Anderson, R.A. Sobel, and S.S. Zamvil. 2016. Tolerance checkpoint bypass permits emergence of pathogenic T cells to neuromyelitis optica autoantigen aquaporin-4. *Proc Natl Acad Sci U S A* 113:14781-14786.
- Vogel, A.L., B. Knier, K. Lammens, S.R. Kalluri, T. Kuhlmann, J.L. Bennett, and T. Korn. 2017. Deletional tolerance prevents AQP4-directed autoimmunity in mice. *Eur J Immunol* 47:458-469.

Reviewer Reports on the Second Revision:

Referees' comments:

Referee #3 (Remarks to the Author):

#3

Additional data on the detection specificity of anti-AQP4 antibody, which I pointed out in my first review comment, was provided (Fig 1e). As I expected, the authors' data show that only a fraction of AQP4+ cells in the thymus are B cells, while the majority of the AQP4+ cells are not B cells. The authors commented that AQP4+ non-B cells are likely thymocytes. However, those AQP4+ cells appear to be stromal cells (not necessarily TECs), and there is no evidence that thymocytes express AQP4. In AQP4-deltaB mice, even though the architecture of AQP4-expressing stroma in the thymus is similar to that in WT mice, the loss of AQP4 expression in a fraction of B cells results in impaired negative selection of AQP4-specific CD4 T cells (as shown in other Figures). This is a surprising results and an issue to be tested by future studies.

#4

I appreciate that the authors provided new data. It shows that thymic B cells contribute to deletion of AQP4-specific CD4SP thymocytes.

#5

Yes, it has been known that some TCRs can bind both MHC-I and MHC-II and lead to differentiation into CD4 T or CD8 T cells (Eshima et al, J Immunol 2006). That is not my point. I am pointing out, as I stated in my first comment, whether the experiments with TCR retrogenic mice (Fig 4 and Ext Data Fig 7) were properly established. And as I wrote in my second comment, the concerns I noted in my first comment remain unaddressed. In my first comment, I pointed that in those TCR retrogenic mice, some stresses under the experimental conditions may have caused reduction of DP thymocytes and then affect the frequency of CD4SP thymocytes. To prove that the TCR retrogenic mice are not stressed and have normal thymocyte development, the authors should show data on total thymocyte numbers (as pointed out in my first comment).

The suspicion that the TCR retrogenic experiments have not been properly established and the frequency of CD4SP thymocytes may be incorrectly assessed is also related to the data that very low numbers of thymic B cells caused massive deletion of AQP4-specific TCR retrogenic thymocytes. The authors commented that the absolute number of B cells may have been underestimated due to the fragility of B cells during cell preparation, but is there any evidence of such a thing? I do not agree that the authors' conclusions are solid.

Author Rebuttals to Second Revision:

Referee #3 (Remarks to the Author):

#3

Additional data on the detection specificity of anti-AQP4 antibody, which I pointed out in my first review comment, was provided (Fig 1e). As I expected, the authors' data show that only a fraction of AQP4+ cells in the thymus are B cells, while the majority of the AQP4+ cells are not B cells. The authors commented that AQP4+ non-B cells are likely thymocytes. However, those AQP4+ cells appear to be stromal cells (not necessarily TECs), and there is no evidence that thymocytes express AQP4. In AQP4-deltaB mice, even though the architecture of AQP4-expressing stroma in the thymus is similar to that in WT mice, the loss of AQP4 expression in a fraction of B cells results in impaired negative selection of AQP4-specific CD4 T cells (as shown in other Figures). This is a surprising results and an issue to be tested by future studies.

With all due respect, we have extensively tested the hypothesis that B cells are key negative selectors for AQP4-specific thymocytes in the current manuscript. In fact, we provide a genetic loss-of-function model where we ablate AQP4-expression selectively in B cells (and by that means eliminate tolerance induction against AQP4 in *Aqp4^{ΔB}* mice). In addition, we created a model in which only B cells (but no other cell types) are able to express AQP4 and show that tolerisation against AQP4 is intact in this model. We show that CD40 (plus IL-21) stimulated B cells (as well as directly ex vivo isolated thymic B cells) can present their endogenous AQP4 in the context of MHC class II and stimulate an AQP4-specific TCR. Finally, while genetic ablation of AQP4 in B cells eliminates tolerance induction against AQP4, genetic ablation of AQP4 in TECs does not, suggesting that mTECs fail to be the only thymic APC subset that purges the repertoire of AQP4-specific TCRs.

All these data make the case that B cells are relevant thymic APCs to ablate AQP4-specific thymocytes. Yet, as we have shown in Fig. 1 of our manuscript, B cells are not the only cell type that expresses AQP4 in the thymus. TECs express it (see also Extended Data Fig. 2). Also, naive T cells have been shown to express AQP4 (Ayasoufi et al., 2018). In order to illustrate that thymocytes also express AQP4 and thus present a source for the non-B cell AQP4-signal in the microphotographs of Fig. 1, we sorted DN, DP, CD4+SP, and CD8+SP thymocytes from the thymus of wild-type and *Aqp4^{ΔB}* mice and performed qPCR for *Aqp4*. We provide these data here as a Figure to this letter (Fig. L1):

Figure L1. AQP4 expression in thymocytes. Thymic B cells (IgM⁺IgD⁻) and subsets of thymocytes, i.e. CD4⁻CD8⁻ (double negative, DN), CD4⁺CD8⁺ (double positive, DP), CD4⁺ single positive (SP), and CD8⁺ SP, were sorted from the thymus of wild-type mice (WT), global *Aqp4*^{-/-} mice, and B cell-conditional AQP4-deficient mice (*Aqp4*^{AB}). RNA was prepared from these highly pure subsets, transcribed into cDNA, and analyzed for *Aqp4* expression (in relation to GAPDH) by quantitative PCR. Expression values were all normalized to the expression of *Aqp4* in astrocytes (external calibrator, set to 1.0). The individual symbols indicate biological replicates (n=2 mice). While no *Aqp4* signal was detectable (n.d.) in global *Aqp4*^{-/-} thymocytes as expected (negative control), all thymocyte subsets isolated from wild-type or *Aqp4*^{AB} mice showed *Aqp4* expression in a similar range.

#4

I appreciate that the authors provided new data. It shows that thymic B cells contribute to deletion of AQP4-specific CD4SP thymocytes.

Thank you. This statement is consistent with our interpretation of the data.

#5

Yes, it has been known that some TCRs can bind both MHC-I and MHC-II and lead to differentiation into CD4 T or CD8 T cells (Eshima et al, J Immunol 2006). That is not my point. I am pointing out, as I stated in my first comment, whether the experiments with TCR retrogenic mice (Fig 4 and Ext Data Fig 7) were properly established. And as I wrote in my second comment, the concerns I noted in my first comment remain unaddressed. In my first comment, I pointed that in those TCR retrogenic mice, some stresses under the experimental conditions may have caused reduction of DP thymocytes and then affect the frequency of CD4SP thymocytes. To prove that the TCR retrogenic mice are not stressed and have normal thymocyte development, the authors should show data on total thymocyte numbers (as pointed out in my first comment).

While this referee concedes that some TCRs can actually be sorted into both the CD4⁺ and CD8⁺ compartment in a retrogenic setting, which we had pointed out before, they now raise the issue that the retrogenic thymocytes might be "stressed" under our specific experimental conditions (Fig. 4 of the revised manuscript). We agree that a retrogenic setting is an inherently artificial situation. However, the point we are making here is based on the comparison of situation A (retrogenic model with wild-type B cells present) and situation B (identical retrogenic model with the only difference to situation A being that now AQP4-deficient B cells are present). Therefore, we are very confident that whatever the "delta" in the retrogenic CD4⁺SP compartment might be, this must have been influenced by the only experimental parameter that was changed between conditions A and B, i.e., the deficiency of AQP4 in B cells.

The suspicion that the TCR retrogenic experiments have not been properly established and the frequency of CD4SP thymocytes may be incorrectly assessed is also related to the data that very low numbers of thymic B cells caused massive deletion of AQP4-specific TCR retrogenic thymocytes. The authors commented that the absolute number of B cells may have been underestimated due to the fragility of B cells during cell preparation, but is there any evidence of such a thing? I do not agree that the authors' conclusions are solid.

It is difficult to put forward a scientific argument that can tackle this point. As pointed out above, we have used proper controls in our retrogenic setting to support our claims. A low number of any APC in the thymus may not *per se* be an argument for its irrelevance. Within the framework of promiscuous gene expression

of TECs, very low numbers of specific mTECs can actually delete an entire monoclonal TCR repertoire. Yet, it is by now a well-established concept that mTECs are very relevant for the negative selection of thymocytes that recognize tissue-restricted antigens. Here, we show by multiple approaches that B cells have a major contribution to the ablation of AQP4-specific T cells in the thymus.

References

Ayasoufi, K., N. Kohei, M. Nicosia, R. Fan, G.W. Farr, P.R. McGuirk, M.F. Pelletier, R.L. Fairchild, and A. Valujskikh. 2018. Aquaporin 4 blockade improves survival of murine heart allografts subjected to prolonged cold ischemia. *Am J Transplant* 18:1238-1246.